# Truncated *FGFR2* is a clinically actionable oncogene in multiple cancers

Daniel Zingg[1,2,22], Jinhyuk Bhin[1,2,3,22], Julia Yemelyanenko[1,2,22], Sjors M. Kas[1,2,22], Frank Rolfs[1,2,4], Catrin Lutz[1,2], Jessica K. Lee[5], Sjoerd Klarenbeek[6], Ian M. Silverman[7], Stefano Annunziato[1,2], Chang S. Chan[8,9], Sander R. Piersma[4], Timo Eijkman[1,2], Madelon Badoux[1,2], Ewa Gogola[1,2], Bjørn Siteur[10], Justin Sprengers[10], Bim de Klein[1,2], Richard R. de Goeij-de Haas[4], Gregory M. Riedlinger[9,11], Hua Ke[8,9], Russell Madison[5], Anne Paulien Drenth[1,2], Eline van der Burg[1,2], Eva Schut[1,2], Linda Henneman[1,2,10], Martine H. van Miltenburg[1,2], Natalie Proost[10], Huiling Zhen[12], Ellen Wientjens[1,2], Roebi de Bruijn[1,2,3], Julian R. de Ruiter[1,2,3], Ute Boon[1,2], Renske de Korte-Grimmerink[10], Bastiaan van Gerwen[10], Luis Féliz[13], Ghassan K. Abou-Alfa[14,15], Jeffrey S. Ross[5,16], Marieke van de Ven[10], Sven Rottenberg[1,17,18], Edwin Cuppen[2,19,20], Anne Vaslin Chessex[21], Siraj M. Ali[5], Timothy C. Burn[7], Connie R. Jimenez[4], Shridar Ganesan[8,9 ✉], Lodewyk F. A. Wessels[2,3 ✉] & Jos Jonkers[1,2 ✉]

Somatic hotspot mutations and structural amplifications and fusions that affect fibroblast growth factor receptor 2 (encoded by *FGFR2*) occur in multiple types of cancer[1]. However, clinical responses to FGFR inhibitors have remained variable[1–9], emphasizing the need to better understand which *FGFR2* alterations are oncogenic and therapeutically targetable. Here we apply transposon-based screening[10,11] and tumour modelling in mice[12,13], and find that the truncation of exon 18 (E18) of *Fgfr2* is a potent driver mutation. Human oncogenomic datasets revealed a diverse set of *FGFR2* alterations, including rearrangements, E1–E17 partial amplifications, and E18 nonsense and frameshift mutations, each causing the transcription of E18-truncated *FGFR2* (*FGFR2^{ΔE18}*). Functional in vitro and in vivo examination of a compendium of *FGFR2^{ΔE18}* and full-length variants pinpointed *FGFR2*-E18 truncation as single-driver alteration in cancer. By contrast, the oncogenic competence of *FGFR2* full-length amplifications depended on a distinct landscape of cooperating driver genes. This suggests that genomic alterations that generate stable *FGFR2^{ΔE18}* variants are actionable therapeutic targets, which we confirmed in preclinical mouse and human tumour models, and in a clinical trial. We propose that cancers containing any *FGFR2* variant with a truncated E18 should be considered for FGFR-targeted therapies.

FGFR2 is a receptor tyrosine kinase (RTK) that consists of an extracellular ligand-binding domain, intracellular tyrosine kinase domains and a carboxy (C)-terminal tail relevant for receptor activity fine-tuning[14]. In human cancers, *FGFR2* can be affected by hotspot mutations and structural variants, namely fusions and amplifications[1], some of which produce truncated FGFR2 isoforms[15–19]. *FGFR2* structural variants have been considered to be oncogenic and actionable due to the resulting overexpression and increased stabilization of the receptor[2,20–22]. However, in patients with cancer with such structural variants, ATP-competitive small-molecule inhibitors targeting FGFRs have produced inconsistent clinical benefits[1–9]. A better understanding of the determinants defining the oncogenicity and clinical actionability of *FGFR2* structural variants is therefore critical for precise matching of cancer patients to FGFR-targeted therapies.

## A *SB* screen identified *Fgfr2^{ΔE18}*

*Sleeping Beauty* (*SB*) transposon-based insertional mutagenesis screening has revealed potential tumour drivers in mice through transcriptional activation and/or truncation of target genes, and identified *Fgfr2*

[1]Division of Molecular Pathology, Netherlands Cancer Institute, Amsterdam, The Netherlands. [2]Oncode Institute, Utrecht, The Netherlands. [3]Division of Molecular Carcinogenesis, Netherlands Cancer Institute, Amsterdam, The Netherlands. [4]OncoProteomics Laboratory, Department of Medical Oncology, Cancer Center Amsterdam, Amsterdam UMC, Vrije Universiteit Amsterdam, Amsterdam, The Netherlands. [5]Foundation Medicine, Cambridge, MA, USA. [6]Experimental Animal Pathology, Netherlands Cancer Institute, Amsterdam, The Netherlands. [7]Incyte Research Institute, Wilmington, DE, USA. [8]Department of Medicine, Division of Medical Oncology, Rutgers Cancer Institute of New Jersey, New Brunswick, NJ, USA. [9]Department of Medicine and Pharmacology, Rutgers University, Piscataway, NJ, USA. [10]Mouse Clinic for Cancer and Aging, Netherlands Cancer Institute, Amsterdam, The Netherlands. [11]Department of Pathology, Rutgers Cancer Institute of New Jersey, New Brunswick, NJ, USA. [12]Incyte, Wilmington, DE, USA. [13]Incyte Biosciences International, Morges, Switzerland. [14]Department of Medicine, Memorial Sloan Kettering Cancer Center, New York, NY, USA. [15]Department of Medicine, Weill Medical College at Cornell University, New York, NY, USA. [16]Upstate University Hospital, Upstate Medical University, Syracuse, NY, USA. [17]Institute of Animal Pathology, Vetsuisse Faculty, University of Bern, Bern, Switzerland. [18]Bern Center for Precision Medicine, University of Bern, Bern, Switzerland. [19]Hartwig Medical Foundation, Amsterdam, The Netherlands. [20]Center for Molecular Medicine, University Medical Center Utrecht, Utrecht, The Netherlands. [21]Debiopharm International, Lausanne, Switzerland. [22]These authors contributed equally: Daniel Zingg, Jinhyuk Bhin, Julia Yemelyanenko, Sjors M. Kas. ✉e-mail: ganesash@cinj.rutgers.edu; l.wessels@nki.nl; j.jonkers@nki.nl

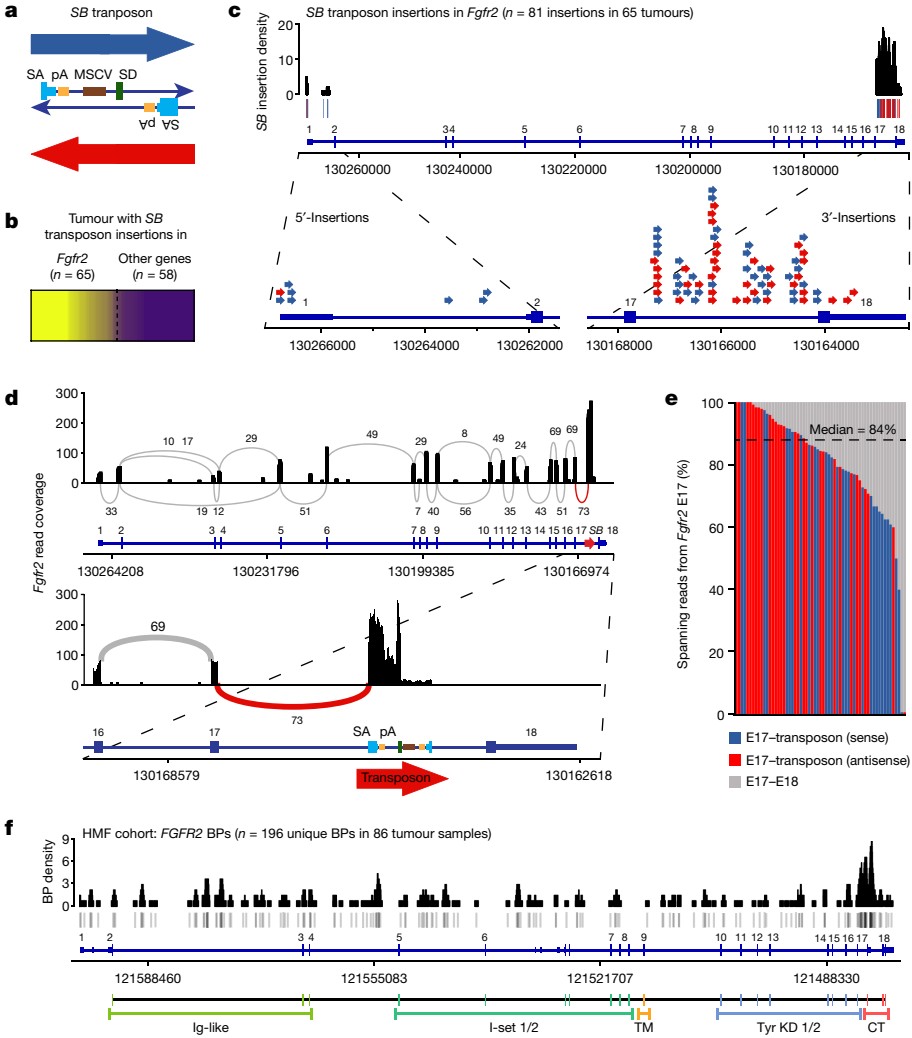

**Fig. 1 | *SB* transposon screen and WGS analysis identifies recurrent *FGFR2* E18 truncation. a**, Schematic of the *SB* transposon, which encodes splice acceptors (SA) followed by polyadenylation (pA) signals on both strands and a murine stem cell virus (MSCV) promoter followed by a splice donor (SD) on the plus strand. **b**, Mammary tumours with *SB* transposon insertions in *Fgfr2* as identified in an insertional mutagenesis screen[10]. The relative clonality of *SB* insertions in *Fgfr2* is shown by a colour gradient (yellow to purple, clonality of 1 to 0). *SB* insertions in *Fgfr2* were called for tumours with a *Fgfr2* relative insertion clonality of ≥0.25. **c**, The *SB* transposon insertions found in *Fgfr2* (chromosome 7). The *SB* insertion density was calculated using a 500 bp sliding window. The blue bars/arrows show sense *SB* insertions; the red bars/arrows show antisense *SB* insertions. **d**, Sashimi plot showing *Fgfr2* read coverage and junction reads plotted as arcs with the indicated junction read counts of a tumour with an I17 antisense *SB* insertion. **e**, The ratio of spanning reads from *Fgfr2*-E17 to E18 versus *SB* transposon in tumours with I17 *SB* insertions. *n* = 31 (sense) and *n* = 30 (antisense). **f**, BPs generating *FGFR2* (chromosome 10) genomic REs identified in 86 out of 2,112 analysed WGS profiles from the HMF cohort on metastatic solid tumours[23]. *n* = 266 (total) and *n* = 196 (unique) BPs. BP density was calculated using a 500 bp sliding window. The grey bars show BPs. Corresponding protein domains are indicated. CT, C terminus; TM, transmembrane; Tyr KD, tyrosine kinase domain.

as a top candidate driver in mammary tumorigenesis[10,11] (Fig. 1a,b). Mapping of the *SB* insertions in *Fgfr2* showed strong enrichment for insertions in intron 17 (I17; Fig. 1c and Extended Data Fig. 1a). Analysis of RNA sequencing (RNA-seq) data of *SB* tumours revealed that *Fgfr2*-I17 insertions enforce splicing of *Fgfr2*-E17 into the transposon. This led to *Fgfr2* transcripts that lacked E17–E18 spanning reads, and expression of *Fgfr2*^ΔE18 was confirmed by reverse transcription with quantitative PCR (RT–qPCR; Fig. 1d,e and Extended Data Fig. 1b–d). Tumours with *SB* insertions in the 5′ region of *Fgfr2* either contained a second *SB* insertion in I17 or contained rearrangements (REs) in *Fgfr2*-I17, producing gene fusions[11] and therefore also expressing *Fgfr2*^ΔE18 (Extended Data Fig. 1d). E18 of both mouse and human *FGFR2* encodes the C terminus of this RTK[14]. We observed an overall upregulation of *Fgfr2* transcripts in tumours with *SB* insertions (Extended Data Fig. 1e), suggesting a loss of regulatory elements that are presumably encoded by the *Fgfr2*

3′-untranslated region (3′-UTR)[21] and/or positive oncogenic selection of C-terminally truncated FGFR2.

## *FGFR2*^ΔE18 variants in human cancer

To assess whether genomic alterations producing *FGFR2*^ΔE18 occur in human cancers, we first analysed whole-genome sequencing (WGS) data of metastatic solid tumours from the Hartwig Medical Foundation (HMF)[23]. Examination of structural variants affecting *FGFR2* in 2,112 HMF WGS profiles revealed a significant enrichment of RE breakpoints (BPs) in I17 (Fig. 1f and Extended Data Fig. 1f), coinciding with reported *FGFR2* fusion BPs[20–22]. Recurring chromosomal REs, such as breakage–fusion–bridge cycles, can produce focal *FGFR2* amplifications (*FGFR2*^amp)[16,17,24], which we observed in a fraction of tumours with *FGFR2* REs (Extended Data Fig. 1g). Some *FGFR2*-I17 REs implicated

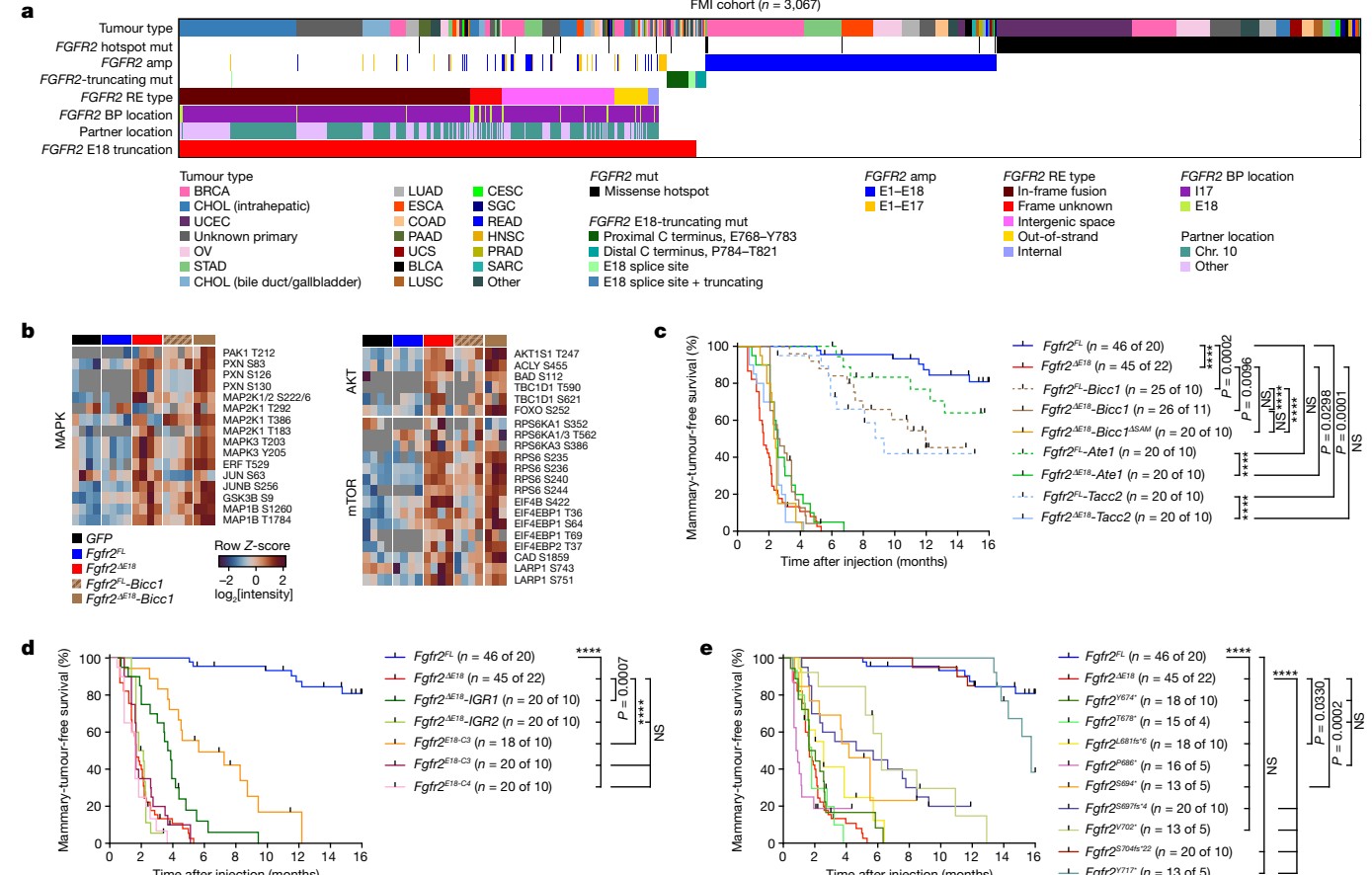

**Fig. 2 | Human *FGFR2* E18-truncating alterations are oncogenic drivers in mice. a**, Analysis of 3,067 samples (1.23% incidence) containing *FGFR2*-I17/E18 in-frame fusions (*n* = 757, 0.30% incidence), frame unknown REs (*n* = 82, 0.03% incidence), intergenic space REs (*n* = 291, 0.12% incidence), out-of-strand REs (*n* = 88, 0.04% incidence), internal REs (*n* = 29, 0.01% incidence), *FGFR2*-E18 splice-site mutations (mut; *n* = 21, 0.01% incidence), E18-truncating nonsense and frameshift mutations (proximal, *n* = 59, 0.02% incidence; distal, *n* = 23, 0.01% incidence), *FGFR2*-E1–E17 partial amplifications (amp; *n* = 73, 0.03% incidence), E1–E18 full-length amplifications (*n* = 838, 0.34% incidence), and/or *FGFR2* missense hotspot mutations affecting Ser252, Cys382, Asn549 or Lys659 (*n* = 978, 0.39% incidence) found in 249,570 pan-cancer diagnostic panel-seq profiles from FMI. BLCA, bladder urothelial carcinoma; BRCA, breast invasive carcinoma; CESC, cervical squamous cell carcinoma and endocervical adenocarcinoma; CHOL, cholangiocarcinoma; chr, chromosome; COAD, colon adenocarcinoma; ESCA, oesophageal carcinoma; HNSC, head and neck squamous cell carcinoma; LUAD, lung adenocarcinoma; LUSC, lung squamous cell carcinoma; OV, ovarian serous cystadenocarcinoma; PAAD, pancreatic adenocarcinoma; PRAD, prostate adenocarcinoma; READ, rectum

adenocarcinoma; SARC, sarcoma; SGC, salivary gland carcinoma; STAD, stomach adenocarcinoma; UCEC, uterine corpus endometrial carcinoma; UCS, uterus carcinosarcoma. **b**, Global phosphoproteomic analysis of NMuMG cells expressing *GFP* or the indicated *Fgfr2* variants. Groups were compared in a pairwise manner using the robust kinase activity inference (RoKAI) tool, including two-tailed hypothesis testing on *Z*-scores and false-discovery rate (FDR) multiple-testing correction using the Benjamini–Hochberg method. Group-comparison fold change (FC) values of −1.5 ≥ FC ≥ 1.5 and *P* < 0.05 were considered. The heatmaps show phosphosites subselected from the RoKAI output and grouped into the indicated signalling pathways guided by RoKAI as colour-coded row *Z*-scores calculated from log₂-transformed intensity values. **c**–**e**, Kaplan–Meier curves showing mammary-tumour-free survival of female mice intraductally injected with lentiviruses encoding the indicated *Fgfr2* variants. Cohort counts (*n*) are injected mammary glands (MGs) per number of mice. The *Fgfr2^FL^* and *Fgfr2^ΔE18^* curves in **c** are duplicated in **d** and **e**. *P* values were calculated using log-rank (Mantel–Cox) tests; \*\*\*\*P < 0.0001; NS, not significant (*P* ≥ 0.05).

canonical in-frame fusions with 3′-partner genes, but we also found non-canonical REs in which the reading frame of the partner gene was undeterminable (frame unknown), the partner gene was out of strand, the partner sequence was derived from intergenic space or the REs occurred internally in *FGFR2* (Extended Data Fig. 1h and Supplementary Table 1).

We next analysed oncogenomic data from Foundation Medicine (FMI) derived from 249,570 targeted tumour sequencing assays for the occurrence of *FGFR2* alterations. Across cancers, we identified 1,367 samples with alterations potentially producing *FGFR2^ΔE18^* (0.55% incidence). These were mutually exclusive to samples with *FGFR2^amp^* (*n* = 838, 0.34% incidence) or *FGFR2* missense hotspot mutations (*FGFR2^hotspot^*; *n* = 978, 0.39% incidence; Fig. 2a and Extended Data Fig. 2a). Alterations potentially perturbing E18 were made up of 55.4% *FGFR2*-I17/E18

in-frame fusions. The remaining 44.6% were classified as variants of unknown significance and comprised *FGFR2*-I17/E18 frame-unknown, out-of-strand, intergenic space and internal REs, some of which coincided with focal amplifications (Fig. 2a, Extended Data Fig. 2b and Supplementary Table 2). Across the *FGFR2* REs, intrachromosomal REs and known fusion partner genes (for example, *BICC1*, *TACC2* and *ATE1*)[20–22] were enriched (Extended Data Fig. 2c,d). *FGFR2* RE partner genes encoded 337 unique proteins. Among these, the ability to self-interact was enriched as compared to the human proteome (Extended Data Fig. 2e–g). Nevertheless, 42.8% of all REs made use of a partner without evident self-interaction ability (Extended Data Fig. 2h). Other variants of unknown significance were *FGFR2*-E1–E17 partial amplifications, E18 splice-acceptor-site mutations and protein-truncating mutations significantly enriched in the coding sequence of *FGFR2*-E18 (Fig. 2a

and Extended Data Fig. 2i–k). The identified *FGFR2*<sup>ΔE18</sup> variants were most frequent in cholangiocarcinoma, but we also found considerable frequencies of in-frame fusions and especially structural variants of unknown significance in gastroesophageal and breast cancer (Extended Data Fig. 2l).

## Expression of *FGFR2*<sup>ΔE18</sup> in human cancer

To validate expression of *FGFR2*<sup>ΔE18</sup> variants, we analysed RNA-seq profiles matched to the HMF WGS samples. In the majority of the cases in which RNA-seq data was available, the predicted *FGFR2* RE types were robustly expressed; REs with intergenic space produced *FGFR2* transcripts terminating in intergenic region (IGR) pseudoexons encoding splice acceptors, a coding sequence and stop codons (Extended Data Figs. 1h and 3a–c). We also observed splicing to an alternative *FGFR2*-E18, termed C3 (Extended Data Fig. 3b–e), which is located in I17 and encodes a single isoleucine followed by a 3′-UTR. Two more *FGFR2* isoforms make use of an alternative E18. The encoded C termini either overlap with the proximal part of the canonical FGFR2 C terminus (C2) or are different to it (C4)[16–18]. Thus, splicing to E18-C3 or E18-C4 generates *FGFR2* isoforms that encode dysfunctional C termini resembling E18 truncation (Extended Data Fig. 2k). In a few cases with IGR REs, we found *FGFR2* in-frame fusions at the RNA level. Reconstruction of derivate chromosomes revealed complex *FGFR2* REs with several BPs that ultimately yielded in-frame fusions with protein-coding genes (Extended Data Fig. 3d,e).

Next, we performed hybrid-capture RNA-seq analysis of two tumour samples, which were diagnosed by FMI to contain structural variants of unknown significance in *FGFR2*. One sample contained an *FGFR2*-I17 RE with intergenic space and the other contained an *FGFR2*<sup>amp</sup> involving E1–E17 only. RNA-seq profiling revealed a *FGFR2* in-frame fusion in the first tumour, whereas the second tumour showed high *FGFR2*-E1–E17 expression with splicing to E18-C3 (Extended Data Fig. 4a,b). Comprehensive analysis of The Cancer Genome Atlas (TCGA) RNA-seq data identified tumours expressing *FGFR2* in-frame fusions as well as non-canonical REs (Extended Data Fig. 4c–e). We found a few tumours containing *FGFR2*-I17 REs and concomitantly using E18-C3. However, a larger fraction of tumours used *FGFR2*-E18-C3 or *FGFR2*-E18-C4 in a mutually exclusive manner (Extended Data Fig. 4d–g and Supplementary Table 3). Taken together, we demonstrated that human tumours express diverse *FGFR2*<sup>ΔE18</sup> transcripts derived from a variety of genomic alterations and alternative splicing events.

## E18 loss is key to *FGFR2* oncogenicity

Previous research showed in vitro transforming abilities of C-terminally truncated FGFR2 isoforms[17–19,25–27]. Our in vivo screening data and analyses of human oncogenomic datasets similarly suggested that exclusion of E18 is a critical determinant to render *FGFR2* REs oncogenic. To test this, we introduced mouse *Fgfr2*<sup>ΔE18</sup> variants into mouse mammary epithelial cells. These were *Fgfr2*<sup>ΔE18</sup> alone or fused to *Ate1*, *Bicc1*, *Tacc2*, two of the IGRs found in TCGA (Extended Data Fig. 4f,g), or the human E18-C2, E18-C3 or E18-C4 sequences, as well as *Fgfr2* bearing E18 nonsense and frameshift mutations. The corresponding controls were full-length (FL) *Fgfr2* (representing *FGFR2*<sup>amp</sup>), *Fgfr2*<sup>FL</sup> fusions, *Fgfr2*<sup>hotspot</sup> variants and kinase-domain-dead *Fgfr2*<sup>K422R</sup> variants (Extended Data Figs. 2k and 5a and Supplementary Table 4). Mass-spectrometry-based expression proteomics and phosphoproteomics revealed that overexpressed *Fgfr2*<sup>ΔE18</sup> and *Fgfr2*<sup>ΔE18</sup>-*Bicc1* both induced FGFR2 signalling resulting in the activation of the MAPK and PI3K–AKT–mTOR pathways (Fig. 2b and Extended Data Fig. 5b–f). This depended on a functional FGFR2 kinase domain, whereas the BICC1–SAM oligomerization domain[28] was dispensable for *Fgfr2*<sup>ΔE18</sup>-*Bicc1* activity (Extended Data Fig. 5g,h). Comparably, all of the tested *Fgfr2*<sup>ΔE18</sup> variants, including proximal E18-truncating mutations and hotspot *Fgfr2*<sup>C287R</sup>, promoted colony formation in a 3D soft agar assay (Extended Data Fig. 6a). By contrast, overexpression of *Fgfr2*<sup>FL</sup>, its fusion variants that retain E18, and distal E18-truncating mutations and the remaining *Fgfr2*<sup>hotspot</sup> variants had limited potential to promote FGFR2 signalling or soft agar colonies (Fig. 2b and Extended Data Figs. 5 and 6a).

Next, we evaluated the in vivo oncogenicity of *Fgfr2* variants using somatic delivery to mouse mammary glands through intraductal injection of lentiviruses[12,13]. Lineage tracing using lentiviral *Fgfr2-P2A-cre* constructs and *mT/mG* female mice showed comparable mammary epithelial transduction rates and FGFR2 expression levels across the *Fgfr2* variants tested. However, only *Fgfr2*<sup>ΔE18</sup> variants drove clonal expansion of the mammary epithelium, which depended on the FGFR2 kinase domain but not on the BICC1–SAM oligomerization domain (Extended Data Fig. 6b,c). To assess *Fgfr2*<sup>ΔE18</sup> oncogenicity in mammary tumour models representative of different breast cancer subtypes—including invasive lobular carcinoma, a hallmark of which is E-cadherin loss[29]—we intraductally delivered *Fgfr2* variants to wild-type (WT) or *Wap-cre;Cdh1*<sup>F/F</sup> mice. *Fgfr2*<sup>ΔE18</sup> variants rapidly induced mammary tumours regardless of *Cdh1* mutation status (Fig. 2c,d and Extended Data Fig. 6d–g), and progressive truncation of *Fgfr2*-E18 gradually decreased tumour onset (Fig. 2e and Extended Data Figs. 6h and 7a,b). By contrast, mammary glands injected with *Fgfr2*<sup>FL</sup> variants displayed no or slow tumorigenesis in WT and *Wap-cre;Cdh1*<sup>F/F</sup> mice (Fig. 2c–e and Extended Data Figs. 6d–h and 7a,b). *Fgfr2*<sup>hotspot</sup> variants were also non-tumorigenic, except for *Fgfr2*<sup>C287R</sup>, which drove marked mammary tumour formation (Extended Data Fig. 7c,d). Furthermore, we generated genetically engineered mouse models (GEMMs) bearing Cre-inducible *Fgfr2-IRES-Luc* alleles (Extended Data Fig. 7e–i), in which *Wap-cre*-mediated induction of *Fgfr2*<sup>FL</sup>-IRES-Luc had comparably little effect on mammary tumorigenesis. However, induction of *Fgfr2*<sup>ΔE18</sup>-IRES-Luc led to increased mammary gland bioluminescence, which coincided with rapid and multifocal tumour formation in *Wap-cre;Cdh1*<sup>F/+</sup>;*Fgfr2*<sup>ΔE18</sup>-IRES-Luc and *Wap-cre;Cdh1*<sup>F/F</sup>;*Fgfr2*<sup>ΔE18</sup>-IRES-Luc females (Extended Data Fig. 7j–m). Histopathological evaluation of the mammary glands of *Fgfr2*<sup>FL</sup> somatic models and GEMMs revealed mostly healthy tissue or low-grade lesions. By contrast, the majority of *Fgfr2*<sup>ΔE18</sup> glands contained FGFR2-positive high-grade adenocarcinomas or E-cadherin-negative invasive lobular carcinomas or sarcomatoid tumours (Extended Data Fig. 8). Proteomic analyses of tumours induced by *Fgfr2* variants demonstrated consistent expression and phosphorylation of FGFR2 variants along with downstream signalling activities, which were distinct from the phosphoproteome of FGFR2-independent *K14-cre;Brca1*<sup>F/F</sup>;*Trp53*<sup>F/F</sup>;*(Mdr1a/b*<sup>−/−</sup>) tumours (Extended Data Fig. 9a–c). Notably, MAPK and AKT–mTOR signalling pathways were particularly active in tumours driven by *Fgfr2*<sup>ΔE18</sup> variants (Extended Data Fig. 9d,e). Together, these data establish that E18 truncation of *Fgfr2* is a bona fide tumour-driver alteration and the loss of the C terminus is a key determinant of FGFR2 oncogenicity.

## *FGFR2* oncogenicity depends on co-drivers

Compared with *Fgfr2*<sup>ΔE18</sup>, our in vivo modelling efforts showed limited oncogenic competences of *Fgfr2*<sup>FL</sup> and *Fgfr2*<sup>hotspot</sup> variants in WT and *Cdh1*-deficient mammary glands. Yet, besides *FGFR2*<sup>ΔE18</sup>, *FGFR2*<sup>amp</sup> and *FGFR2*<sup>hotspot</sup> made up considerable fractions of human *FGFR2* alterations. The oncogenic ability of specific *FGFR2* alterations might be affected by the tissue of origin as well as the mutational context. To examine possible cooperation between *FGFR2* variants and other genes, we analysed driver gene alterations diagnosed by FMI oncogenomic profiling and their incidence in *FGFR2*-altered cancers (Extended Data Fig. 10a and Supplementary Table 2). *FGFR2*<sup>ΔE18</sup>, *FGFR2*<sup>amp</sup> and *FGFR2*<sup>hotspot</sup> showed co-occurrences and mutual exclusivities with distinct sets of driver alterations (Extended Data Fig. 10b,c). The proportions of *FGFR2*<sup>ΔE18</sup>, *FGFR2*<sup>amp</sup> and *FGFR2*<sup>hotspot</sup> varied across cancer entities, suggesting differential selections of *FGFR2* aberrations and concurrent driver

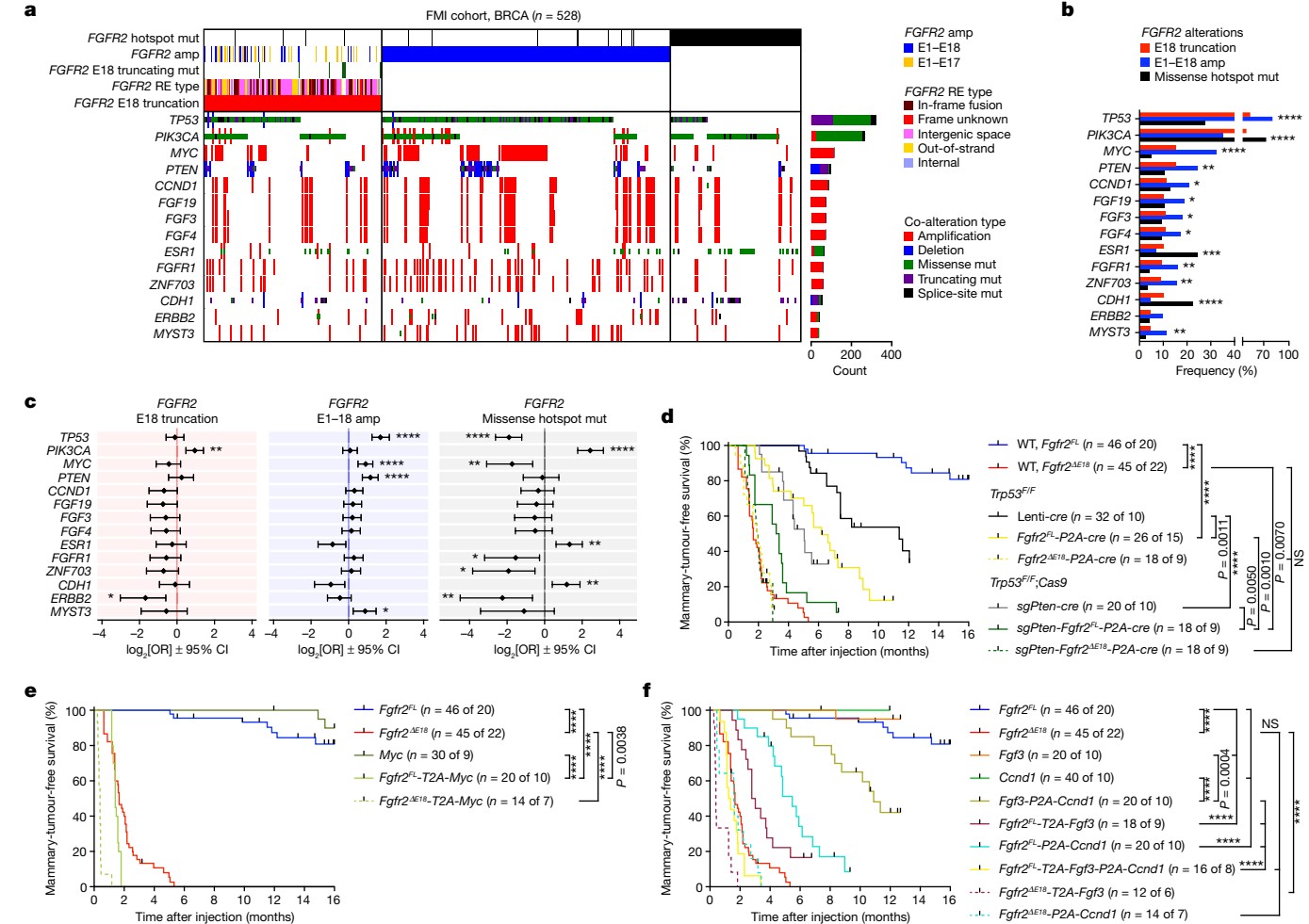

**Fig. 3 | The oncogenic competence of *FGFR2* alterations depends on co-occurring drivers. a**, Analysis of 528 breast cancer samples classified as either *FGFR2* E18-truncated (*n* = 157, 29.7% of total, 0.70% incidence), E1–E18 amplified (*n* = 256, 48.5% of total, 1.14% incidence) or missense hotspot mutant (*n* = 115, 21.8% of total, 0.51% incidence), and the top co-enriched tumour driver alterations found in 22,380 breast cancer profiles from FMI. **b**, Enrichment of the top tumour driver co-alterations in the indicated *FGFR2* alteration categories in the FMI breast cancer cohort. **c**, The odds ratios (ORs) of the top tumour driver co-alterations in the indicated *FGFR2* alteration categories (E18 truncation, *n* = 157; E1–E18 amplification, *n* = 256; missense hotspot mutation, *n* = 115) versus *FGFR2* WT samples (*n* = 22,307) of the FMI breast cancer cohort. Data are represented as log₂-transformed OR ± 95%

confidence interval (CI). Co-occurrence, OR > 1; mutual exclusivity, OR < 1. *P* values were calculated using one-tailed proportion *Z*-tests (**b**) or two-tailed Fisher's exact tests (**c**) with FDR multiple-testing corrections using the Benjamini–Hochberg method (**b** and **c**). Sample sizes and statistical details for **b** and **c** are shown in Supplementary Table 2. **d**–**f**, Kaplan–Meier analysis of the mammary-tumour-free survival of *Trp53*^F/F^ and *Trp53*^F/F^;*Rosa26-Cas9* (**d**) or WT (**e**,**f**) female mice that were intraductally injected with lentiviruses encoding the indicated variants. Cohort counts (*n*) represent injected mammary glands (MGs) per number of mice. The *Fgfr2*^FL^ and *Fgfr2*^ΔE18^ curves in **d**–**f** are duplicates from Fig. 2c. *P* values were calculated using log-rank (Mantel–Cox) tests. *\**P* < 0.05; \*\**P* < 0.01; \*\*\**P* < 0.001; \*\*\*\**P* < 0.0001.

alterations among tissues of origin (Extended Data Fig. 10d). We evaluated driver-gene enrichments among the three *FGFR2* alteration categories in a tumour-type-specific manner. In breast cancers with *FGFR2*^amp^, *TP53* driver mutations, *MYC* amplifications, *PTEN* loss-of-function alterations, and *CCND1* and *FGF3/4/19* co-amplifications were significantly more enriched compared with the other classes of *FGFR2* aberration (Fig. 3a,b). Accordingly, *FGFR2*^amp^ showed co-occurrence with *TP53*, *PTEN* and *MYC* alterations in breast cancer (Fig. 3c). In several other cancer types, we also observed enrichments of *TP53* and *MYC* driver alterations in *FGFR2*-amplified cases (Extended Data Fig. 10e). By contrast, *FGFR2*^ΔE18^ and *FGFR2*^hotspot^ samples did not co-occur with these drivers (Fig. 3b,c and Extended Data Fig. 10e). This suggested that the oncogenic competence of full-length *FGFR2*^amp^ depends on specific cooperating driver genes.

We therefore combined lentiviral *Fgfr2*^FL^ with *cre* to delete floxed *Trp53* (*Trp53*^f^) alleles and a single-guide RNA against *Pten* (sgPten) to disrupt the endogenous *Pten* locus. Intraductal delivery of

*Fgfr2*^FL^-P2A-*cre* or sg*Pten*-*Fgfr2*^FL^-P2A-*cre* lentiviruses into mammary glands of *Trp53*^F/F^ or *Trp53*^F/F^;*Cas9* mice, respectively, significantly increased *Fgfr2*^FL^ tumorigenicity. *Fgfr2*^FL^ became nearly as oncogenic as *Fgfr2*^ΔE18^ when *Trp53* and *Pten* were concomitantly lost, whereas *Fgfr2*^ΔE18^ oncogenicity was unaffected by the loss of *Trp53* and/or *Pten* (Fig. 3d and Extended Data Fig. 11a). Similarly, combinations of *Fgfr2*^FL^ with *Myc*, *Fgf3* and/or *Ccnd1* cDNAs into single lentiviral constructs cooperatively shortened tumour onset after intraductal delivery, with the latencies of the *Fgfr2*^FL^-T2A-*Myc* and *Fgfr2*^FL^-T2A-*Fgf3*-P2A-*Ccnd1* combinations matching *Fgfr2*^ΔE18^ single-driver latency. Notably, *Myc*, *Fgf3* and *Ccnd1* alone were effectively non-tumorigenic (Fig. 3e,f and Extended Data Fig. 11b,c). Evaluation of mammary glands containing *Fgfr2*^FL^ and co-driver alterations confirmed targeting or expression of the driver combinations and revealed high-grade tumours comparable to *Fgfr2*^ΔE18^-driven lesions (Extended Data Fig. 11d,e). Thus, *Fgfr2*^FL^ oncogenicity relied on a cooperative oncogenomic network, whereas *Fgfr2*^ΔE18^ acted as a context-independent oncogene.

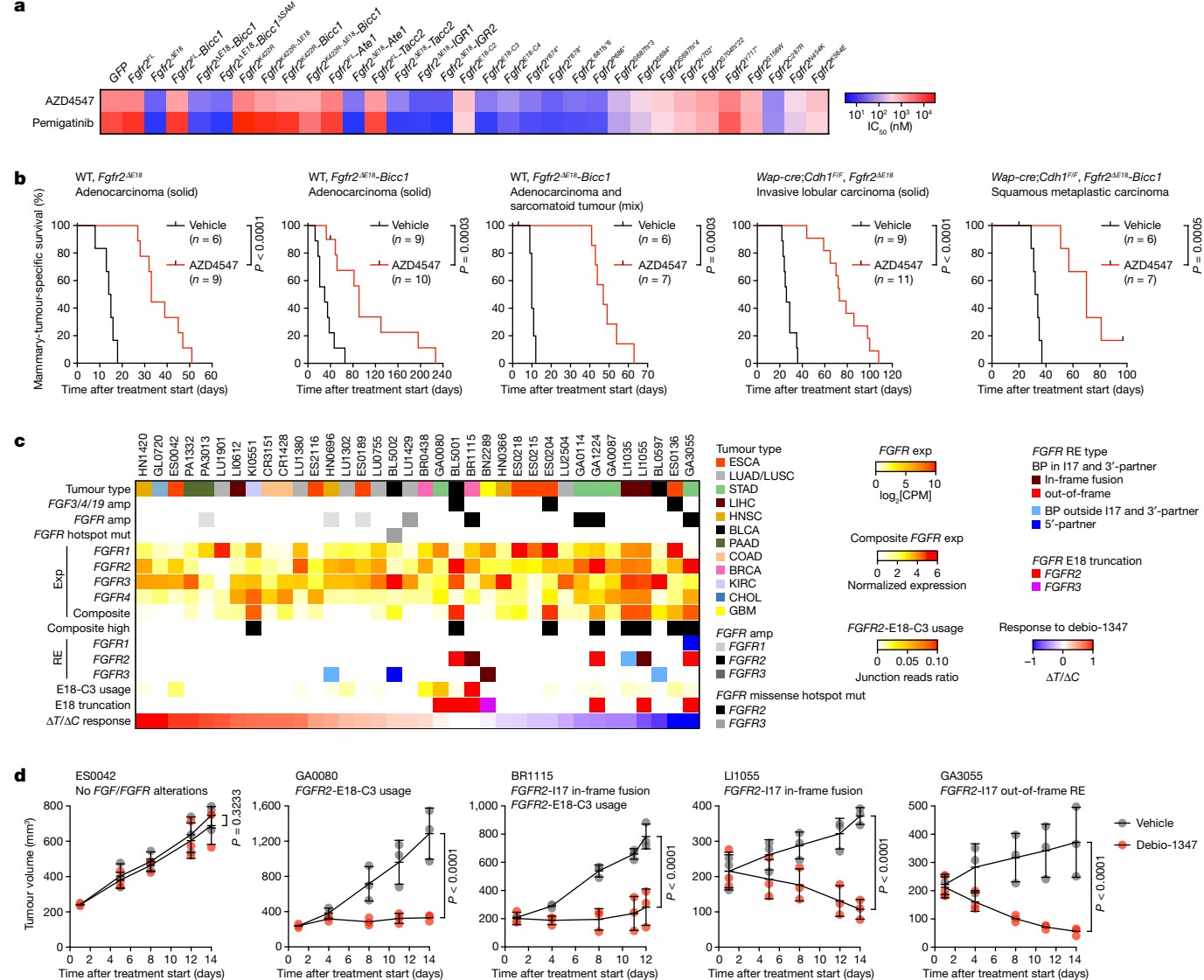

**Fig. 4 | Human and mouse *FGFR2* alteration cancer models are sensitive to FGFRi. a**, Half-maximum inhibitory concentration (IC$_{50}$) value quantifications of 2D-grown NMuMG cells expressing *GFP* or the indicated *Fgfr2* variants and treated with AZD4547 or pemigatinib for 4 days. Data are the mean of 5 independent experiments (*GFP*, *Fgfr2^FL^*, *Fgfr2^ΔE18^*) or 1 experiment (other *Fgfr2* variants). **b**, Kaplan–Meier analysis of mammary-tumour-specific survival of female syngeneic WT mice bearing mammary fat pad transplants derived from the indicated tumour donors and treated daily orally with vehicle or 12.5 mg per kg AZD4547 using an intermittent dosing regimen. *P* values were calculated using log-rank (Mantel–Cox) tests. **c**, Collection of PDX models (*n* = 36) rank-ordered according to debio-1347 *ΔT/ΔC* response ratios. *FGF/FGFR* copy number alteration and mutation data and RNA-seq profiles to analyse *FGF/FGFR*

expression (exp) were obtained from CrownBio-HuPrime, and had been generated from non-treated PDXs. Composite *FGFR* expression was defined as high if normalized expression > 3. *FGFR2*-E18-C3 use and *FGFR* RE types were identified in RNA-seq profiles. GBM, glioblastoma multiforme; KIRC, kidney renal clear cell carcinoma; LIHC, liver hepatocellular carcinoma. **d**, Growth curves of the indicated PDXs engrafted in female BALB/c nude mice and treated daily orally with vehicle or debio-1347 (BR1115 and LI1050, 60 mg per kg; ES0042, GA0080 and GA3055, 80 mg per kg). *n* = 3 mice per PDX model and treatment group. Data are mean ± s.d. *P* values were calculated using one-tailed two-way analysis of variance with FDR multiple-testing corrections using the two-stage step-up method from Benjamini, Krieger and Yekutieli.

## *FGFR2^ΔE18^* tumours are sensitive to FGFRi

We next tested whether different *Fgfr2* variants were sensitive to the clinical FGFR inhibitors (FGFRi) AZD4547, pemigatinib, BGJ398 and debio-1347. Expression of *Fgfr2^ΔE18^* variants and *Fgfr2^C287R^* rendered mouse mammary epithelial cells highly sensitive to FGFR2 inhibition (Fig. 4a and Extended Data Fig. 12a,b). As a consequence, FGFRi suppressed both *Fgfr2^ΔE18^*-variant-induced signalling and soft agar clonogenicity (Extended Data Figs. 5g,h and 6a). By contrast, cells expressing *Fgfr2^FL^* and the remaining *Fgfr2^hotspot^* variants were less sensitive to FGFRi (Fig. 4a and Extended Data Fig. 12a,b). We also orthotopically transplanted tumours driven by *Fgfr2^ΔE18^* variants and treated the recipient

mice with AZD4547, which significantly suppressed tumour growth (Fig. 4b and Extended Data Fig. 12c,d).

To further investigate the connection between distinct *FGFR2* alterations and the FGFRi response in human tumour models, we analysed human cancer cell line pharmacogenomic datasets[30–32]. We evaluated the association between dose–responses to the FGFRi AZD4547 and PD173074 and genomic and transcriptomic features that potentially affect FGFR signalling, that is, *FGF3/4/19* amplification, *FGFR1–4* mutation, amplification, RE and expression, and use of *FGFR2*-E18-C3 (Extended Data Fig. 13a,b and Supplementary Table 5). Among these, *FGFR2/3* missense hotspot mutations, *FGFR2* expression and composite *FGFR* expression—a biomarker of FGFRi response[33]—modestly

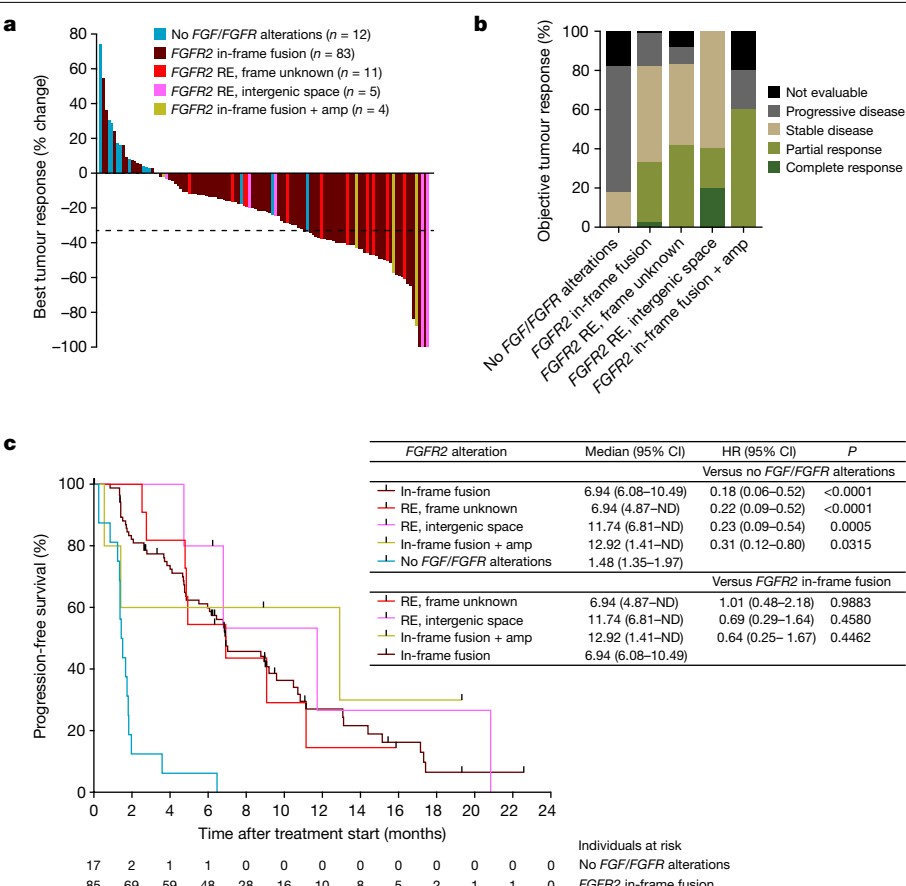

**Fig. 5 | Patients with cholangiocarcinoma respond to FGFR-targeted therapy irrespective of *FGFR2* RE type. a**, Centrally assessed best percentage change from the baseline in target lesion size of 115 (92%) of 125 individual patients with cholangiocarcinoma treated with pemigatinib, who had post-baseline scans. Data are from the FIGHT-202 study[9] and the coloured bars indicate *FGFR2*-I17/E18 RE types and *FGFR2* amplification status as diagnosed by FoundationOne. **b**, Objective tumour responses observed in the FIGHT-202 study assessed according to the Response Evaluation Criteria in Solid Tumours v.1.1 (RECIST 1.1) and grouped according to *FGFR2* RE types/amplification status. No *FGF/FGFR* alterations (*n* = 17), *FGFR2* in-frame fusion (*n* = 85), frame unknown RE (*n* = 12), intergenic space RE (*n* = 5), in-frame fusion + amplification (*n* = 5). 'Not evaluable' indicates that the patient was not evaluable for response using RECIST. **c**, Kaplan–Meier analysis of the progression-free survival of patients with cholangiocarcinoma treated with pemigatinib from the FIGHT-202 study and grouped according to *FGFR2* RE types/amplification status. Data are median ± 95% CI for each cohort, and log-rank hazard ratios (HR) ± 95% CI for the indicated comparisons are shown. *P* values were calculated using log-rank (Mantel–Cox) tests. ND, not defined.

correlated with FGFRi sensitivity (Extended Data Fig. 13c–f), whereas cell lines with concurrent *FGFR2*[amp] and E18 truncations through I17 REs and/or E18-C3 use exhibited high sensitivity to FGFRi (Extended Data Fig. 13a,g). Cell lines expressing E18-truncating *FGFR3* REs were less sensitive to FGFRi compared with cells with *FGFR2*[ΔE18] (Extended Data Fig. 13b,g). On the basis of these correlates, we obtained the human SUM52PE, MFM-223, SNU-16, KATO-III and NCI-H716 cancer cell lines, each expressing amplified *FGFR2* variants[15–18,34,35]. These cells highly expressed full-length *FGFR2*[E18-C1] but also E18-truncated transcripts, namely *FGFR2*[E18-C3] and *FGFR2*-I17 REs, and were sensitive to FGFRi (Extended Data Fig. 14a–c). To functionally dissect the dependence on *FGFR2*[E18-C1] versus E18-truncated transcripts, we used small interfering RNAs (siRNAs) targeting either shared or unique *FGFR2* exons (Extended Data Fig. 14d). Silencing of all *FGFR2* transcripts suppressed the growth of cell lines with *FGFR2*[amp] (Extended Data Fig. 14e,f). Regardless of expression prevalence (Extended Data Fig. 14c), the growth of these cell lines could also be inhibited by specific silencing of E18-truncated *FGFR2* RE or E18-C3 transcripts. By contrast, siRNAs specifically targeting *FGFR2*[E18-C1] only marginally suppressed cell line growth (Extended Data Fig. 14e,f). Importantly, in KATO-III cells mainly expressing *FGFR2*[E18-C3] (Extended Data Fig. 14c), overexpression of *FGFR2*[E18-C3]

fully rescued silencing of any *FGFR2* transcripts, which depended on a functional FGFR2 kinase domain. However, full-length *FGFR2*[E18-C1] was hardly able to rescue silencing of E18-truncated *FGFR2*[E18-C3] (Extended Data Fig. 14g–i and Supplementary Table 4).

We next screened the CrownBio-HuPrime patient-derived xenograft (PDX) collection for the occurrence of genomic and transcriptomic alterations in the FGFR signalling pathway and enrolled the PDXs into a drug-intervention study using debio-1347 (Fig. 4c and Supplementary Table 6). All PDXs with *FGFR2/3*-I17 in-frame fusions as well as those with noncanonical *FGFR2* REs or E18-C3 use strongly responded to FGFR blockade (Fig. 4d and Extended Data Fig. 15a–c). Among the correlates with debio-1347 treatment response were *FGFR2* and composite *FGFR* expression (Extended Data Fig. 15d–f), but especially truncation of *FGFR2/3*-E18 exhibited substantial correlation with debio-1347-mediated growth inhibition (Extended Data Fig. 15g). Thus, human tumour models express and are dependent on E18-truncated *FGFR2* and *FGFR3* variants and are actionable by FGFR-targeted therapies.

These findings suggest that patients with cancer with any type of *FGFR2*[ΔE18] variant might respond to FGFR2 targeting. We therefore re-examined FIGHT-202 (NCT02924376), a phase II trial of pemigatinib

in patients with advanced cholangiocarcinoma. Patients with fusions or REs in the *FGFR2*-I17/E18 hotspot had an objective response rate of 35.5%, whereas those with other or no *FGF/FGFR* alterations had no response[9]. To determine which classes of *FGFR2* REs benefit from pemigatinib, we stratified individual patients according to their *FGFR2$^{amp}$* status and E18-damaging RE types, namely in-frame fusions, frame unknown REs and REs with IGRs. Patients with *FGFR2* in-frame fusions, frame unknown REs and IGR REs—independently of *FGFR2$^{amp}$* status—showed strong tumour responses to pemigatinib therapy (Fig. 5a), resulting in objective responses or stable disease in 80–100% of patients, irrespective of the diagnosed *FGFR2* RE type (Fig. 5b). As a consequence, although the patient cohort with no *FGF/FGFR* alterations quickly progressed during pemigatinib treatment (Fig. 5a–c), the four patient cohorts with non-amplified and amplified *FGFR2* in-frame fusions and non-canonical REs showed equally prolonged progression-free survival times (Fig. 5c). Taken together, *FGFR2$^{ΔE18}$*, generated by either in-frame fusions or other REs, is a clinically actionable oncogene in patients with cholangiocarcinoma and probably in patients with other types of cancer.

## Discussion

The C-terminal tail of FGFR2 encoded by E18 proposedly moderates RTK signalling[14]. The proximal part of the C terminus includes the RTK internalization motif[27], a tyrosine residue that is relevant for PI3K-pathway attenuation[25,36], and serine residues that bind to ERK1/2 and RSK2 to stimulate receptor endocytosis and suppress MAPK signalling[37,38]. The distal C terminus contains proline-rich motifs that bind to GRB2 and mitigate kinase domain activity[39,40]. We found that especially proximal C-terminal truncation is critical for oncogenic signal transduction, as evidenced by the gradually accelerated tumorigenesis observed for the *Fgfr2$^{E18-C2}$* variant and proximal E18-truncating mutations. Notably, C-terminally truncated isoforms of EGFR, HER2 and other RTKs have also shown elevated transforming activities[41–44]. Thus, the paradigm of C-terminal FGFR2 truncation identified here might be key to the pathogenicity of multiple RTKs.

*FGFR2* amplification and fusion structural variants have been considered to be relevant tumour drivers owing to the consequential receptor overexpression and constitutive dimerization mediated by oligomerization domains in the fusion partners[2,20–22]. We identified that the tumour-driver potential of C-terminally truncated FGFR2 is independent of specific fusion partners, whereas full-length FGFR2 overexpression was marginally tumorigenic in the absence of other driver alterations. The oncogenicity of *FGFR2$^{amp}$* might therefore depend on the ability to generate *FGFR2$^{ΔE18}$* transcripts, for example, through complex REs[16,17,24]. As shown in our study, *FGFR2$^{ΔE18}$* acts as a potent single-driver alteration. By contrast, the oncogenic competence of full-length *FGFR2* relied on co-drivers that may augment canonical FGFR2 signalling[45–48] and thereby phenocopy the strong signalling induction observed for C-terminally truncated FGFR2. In clinical trials, objective responses to FGFRi were scarce in patients with *FGFR2$^{amp}$* tumours[2–4]. Interestingly, tumours with overexpression of E18-truncated *FGFR2$^{E18-C3}$* responded particularly well to FGFR2 targeting[2]. In cohorts of mixed *FGFR* alterations, patients with *FGFR2*-E18-truncating fusions displayed favourable responses over patients with other *FGF/FGFR* alterations[5–9]. Thus, FGFRi efficacy might be dictated by the expression of single-driver *FGFR2$^{ΔE18}$* versus *FGFR2* alterations that depend on oncogenic co-drivers. In *FGFR*-aberrant cancers, *MYC* or *CCND1* amplifications can indeed confer resistance to FGFRi[48–50]. Combination therapies might therefore elevate the response rates in *FGFR2$^{amp}$* tumours, as proposed for FGFRi–CDK4/6i combination therapy in patients with breast cancer with *FGFR2* and *CCND1* amplifications[50].

Our findings have fundamental implications for the selection of patients for FGFR2 targeting therapies. Instead of considering patients on the basis of *FGFR2* mutation, fusion or amplification status alone,

our data suggest that expression of oncogenic *FGFR2* transcripts and co-mutational landscapes should also be considered. Importantly, identifying cancers with structural variants or mutations that result in expression of *FGFR2$^{ΔE18}$* variants will be a highly relevant biomarker for FGFR-targeted therapeutics, and may substantially expand the number of cancer patients who may benefit from such therapy.

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

## Methods

### Mouse models

**GEMMs.** The FVB/NRj *Wap-cre;Cdh1^(F/F)^*, *Trp53^(F/F)^*, *Trp53^(F/F)^;Rosa26-Cas9*, and *Rosa26-mT/mG* mouse strains were maintained at the NKI animal facility and PCR-genotyped as previously described[12,13,51–53]. To generate GEMMs bearing *Fgfr2-IRES-Luc* alleles, mouse *Fgfr2* (NM_201601.2) was isolated from a cDNA clone (MC221076, OriGene) using the primer sequences listed in Supplementary Table 7 amplifying *Fgfr2*-E1–E18 (FL) or *Fgfr2*-E1–E17 (ΔE18) and the sequences were verified and inserted into FseI-PmeI fragments into the *Frt-invCag-IRES-Luc* vector (shuttle vector). This resulted in the *Frt-invCAG-Fgfr2^(FL)^-IRES-Luc* and *Frt-invCAG-Fgfr2^(ΔE18)^-IRES-Luc* alleles. Flp-mediated integration of the shuttle vectors in *Wap-cre;Cdh1^(F/F)^;Col1a1^(frt/+)^* GEMM-derived embryonic stem cell (ESC) clones (FVB/NRj background) and subsequent blastocyst injections of the modified ESCs were performed using the GEMM–ESC methodology[54]. Chimeric animals were mated with *Cdh1^(F/+)^* and *Cdh1^(F/F)^* mice on the FVB/NRj background to generate the experimental cohorts. The *Col1a1^(frt-invCAG-Fgfr2-IRES-Luc/+)^* and WT alleles were detected using standard PCR with an annealing temperature of 58 °C using the primer sequences listed in Supplementary Table 8 generating the following PCR products: *Col1a1^(frt-invCAG-Fgfr2-FL-IRES-Luc)^*, 585 bp; *Col1a1^(frt-invCAG-Fgfr2-ΔE18-IRES-Luc)^*, 420 bp; and WT, 234 bp. Here, *Col1a1^(frt-invCAG-Fgfr2-FL-IRES-Luc)^* and *Col1a1^(frt-invCAG-Fgfr2-ΔE18-IRES-Luc)^* are referred to as *Fgfr2^(FL)^-IRES-Luc* and *Fgfr2^(ΔE18)^-IRES-Luc*, respectively. The GEMM cohorts were monitored weekly and mammary-tumour-free survival was scored (event) when the first palpable tumour was detected, whereas mice that did not develop any mammary tumours were censored. Tumour volume was measured in two dimensions using callipers as follows: volume = length × width$^2$ × 0.5.

**Somatic mouse models.** To somatically model *Fgfr2* variants in the mouse mammary gland, 6-week-old FVB/NRj WT, *Wap-cre;Cdh1^(F/F)^*, *Trp53^(F/F)^*, *Trp53^(F/F)^;Rosa26-Cas9* or *Rosa26-mT/mG* female mice were intraductally injected as previously described[12] with lentiviruses encoding *Fgfr2* variants in combination with *cre*, *Myc*, *Ccnd1*, *Fgf3* and/or a previously validated sgRNA targeting E7 of *Pten* (sgPten)[12,13]. In brief, 20 μl of high-titre lentiviruses were injected into the fourth and/or the third mammary glands using a 34G needle. Lentiviral titres ranging from $2 \times 10^8$ to $2 \times 10^9$ transfection units (TU) per ml were used. The somatic model cohorts were monitored twice weekly and mammary-tumour-free survival was scored (event) for each injected mammary gland individually when palpable tumours were detected, whereas mammary glands that did not develop any tumours were censored. Tumour volume was measured in two dimensions using callipers as follows: volume = length × width$^2$ × 0.5.

**AZD4547 intervention study.** To allograft tumours, DMSO-preserved 1 mm$^3$ tumour fragments derived from somatic *Fgfr2* models were orthotopically transplanted into the right mammary fat pad of 8-week-old syngeneic FVB/NRj female mice (Janvier Labs) as previously described[55]. The mice were twice weekly weighed and monitored for mammary tumour development and, as soon as tumours reached a volume of 62.5 mm$^3$ (5 × 5 mm, measured in two dimensions using callipers; volume = length × width$^2$ × 0.5), the mice were randomly allocated to vehicle versus AZD4547 FGFRi treatment arms. The treatments were performed daily through oral gavage using vehicle (1% Tween-80 in demineralized water) or 12.5 mg per kg AZD4547 (AstraZeneca) according to a previously optimized intermittent dosing regimen[55]. Mice were euthanized 1 h after the last dosing.

**General guidelines.** For all mouse models, mammary-tumour-specific survival was scored when a single mammary tumour burden reached a volume of 1,500 mm$^3$, the total mammary tumour burden reached a volume of 2,000 mm$^3$ or the mice suffered from clinical signs of distress, such as respiratory distress, ascites, distended abdomen, rapid weight loss and severe anaemia, caused by primary tumour burden or metastatic disease. Mice that were euthanized due to other circumstances were censored. The maximal permitted disease end points were not exceeded in any of the experiments. Mammary glands were collected and analysed for histological abnormalities. Sample sizes were determined using G*Power software (v.3.1)[56] and were large enough to measure the effect sizes. Tumour measurements and post mortem analyses were performed in a blinded manner. The mouse colony was housed in a certified animal facility under a 12 h–12 h light–dark cycle in a temperature- and humidity-controlled room set to 21 °C and 55% relative humidity. The mice were kept in individually ventilated cages, and food and water were provided ad libitum. All of the animal experiments were approved by the Animal Ethics Committee of the Netherlands Cancer Institute and performed in accordance with institutional, national and European guidelines for Animal Care and Use.

### In vivo bioluminescence imaging

In vivo bioluminescence imaging of luciferase expression was performed as previously described[57] by intraperitoneally injecting 150 mg per kg beetle luciferin (E1601, Promega). Signal intensity was measured on the whole body of the mouse (excluding the head and tail) using an IVIS Spectrum In Vivo Imaging System (124262, PerkinElmer) operated by Living Image Software (v.4.5.2, PerkinElmer) and a size-fixed square. Signal intensity was quantified as flux (photons per second per cm$^2$ per steradian).

### Histology and IHC

Tissues were formalin-fixed and paraffin-embedded (FFPE), sectioned and processed for haematoxylin and eosin (H&E) histochemical and immunohistochemistry (IHC) staining using routine procedures. For IHC staining, antigen retrieval was performed with citrate buffer (CBB999, ScyTek) at pH 6 (FGF3, FGFR2, PTEN) or Tris-EDTA at pH 9 (MYC, Cyclin D1, E-cadherin, P53). Sections were incubated with primary antibodies (Supplementary Table 9) overnight at 4 °C. Primary antibodies were labelled with the EnVision+ HRP Labelled Polymer Anti-Rabbit System (K4003, Dako), visualized with the Liquid DAB+ Substrate Chromogen System (K3468, Dako) and counterstained with haematoxylin. The antibodies used were independently validated by a certified pathologist by evaluation of IHC results in positive and negative biological control FFPE tissues to ensure specificity and sensitivity. Moreover, negative technical controls were performed by omission of the primary antibody in extra sections for a randomly selected subset of the samples. H&E and E-cadherin slides were used to classify mammary tumour lesion types according to the international consensus of mammary pathology[58]. IHC stains were quantitatively analysed by evaluating tumour cell-specific positivity using a histo-scoring system (0, negative; 1, weakly positive; 2, moderately positive; 3, strongly positive) or by calculating a histo (*H*)-score for each tumour defined as follows: *H*-score = 1 × (the percentage of tumour cells with weak staining intensity) + 2 × (the percentage of tumour cells with moderate staining intensity) + 3 × (the percentage of tumour cells with strong staining intensity), resulting in a score between 0 and 300. All slides were reviewed and quantified by a comparative pathologist (S.K.) in a blinded manner. Slides were digitally processed using a Pannoramic 1000 whole-slide scanner (3DHISTECH) and captured using CaseViewer software (v.2.2.1, 3DHISTECH).

### Isolation of MMECs

Primary mouse mammary epithelial cells (MMECs) were isolated from 10-week-old WT, *Fgfr2^(FL)^-IRES-Luc* and *Fgfr2^(ΔE18)^-IRES-Luc* female mice as previously described[53]. In brief, mammary glands were minced and digested with 4 mg ml$^{-1}$ collagenase A (11088793001, Roche) and 25 μg ml$^{-1}$ DNase I (DN25, Sigma-Aldrich) in Dulbecco's modified Eagle medium/nutrient mixture F-12 (DMEM/F-12, 31331, Thermo Fisher Scientific) containing 100 IU ml$^{-1}$ penicillin–streptomycin

(15070, Thermo Fisher Scientific) for 1 h at 37 °C. Digests were passed through a 70 µm cell strainer that was prewetted with PBS containing 10% fetal bovine serum (FBS, S-FBS-EU-015, Serana) and 2 mM EDTA, and cells were cultured in DMEM/F-12 supplemented with 10% FBS, 100 IU ml$^{-1}$ penicillin–streptomycin, 5 ng ml$^{-1}$ epidermal growth factor (EGF, E4127, Sigma-Aldrich), 5 ng ml$^{-1}$ insulin (I0516, Sigma-Aldrich) and 10 µM Y-27632 (M1817, AbMole). Contaminating fibroblasts were removed from MMEC cultures by differential trypsinization. Confluent wells were transduced with adenoviral Ad5CMVCre particles (AdCre, $1 \times 10^9$ TU ml$^{-1}$; University of Iowa Viral Vector Core) in the presence of 8 µg ml$^{-1}$ Polybrene (H9268, Sigma-Aldrich) for 48 h before subjection to subsequent assays. Switching of *Fgfr2-IRES-Luc* alleles was confirmed using 1 mg ml$^{-1}$ beetle luciferin and bioluminescence imaging with an Infinite M Plex plate reader (Tecan) operated with Tecan i-control software (v.3.9.1, Tecan).

### FACS analysis of mammary glands
Mammary glands of *Rosa26-mT/mG* female mice injected with *Fgfr2-P2A-cre* lentiviruses were processed as described for MMEC isolation. Single cells were stained with the fluorescence-activated cell sorting (FACS)-validated BV650-conjugated anti-EPCAM antibody (1:100, 740559, BD Biosciences) in FACS buffer (PBS with 10% FBS and 2 mM EDTA), labelled with the LIVE/DEAD Fixable Violet Dead Cell Stain Kit (405 nm excitation, L34964, Thermo Fisher Scientific), fixed with BD Phosflow Fix Buffer I (557870, BD Biosciences) and permeabilized with BD Phosflow Perm Buffer III (558050, BD Biosciences), each for 30 min at 4 °C. Cells were incubated with primary antibodies (anti-FGFR2, 1:200, 11835, Cell Signaling Technology; anti-GFP, 1:200, ab6673, Abcam) overnight and subsequently with secondary antibodies for 1 h both in FACS buffer at 4 °C. Anti-FGFR2 and anti-GFP antibodies were validated for FACS using NMuMG cells overexpressing GFP or FGFR2 versus control cells negative for these proteins. Details of the antibodies used are provided in Supplementary Table 9. FACS was performed using the BD LSRFortessa Cell Analyzer (BD Biosciences) equipped with the BD FACSDiva Software (v.8.0.2, BD Biosciences) and with 405 nm (450/50, 670/30 pass filters), 488 nm (530/30 pass filters) and 638 nm (670/30 pass filters) lasers to measure 405-Live/Dead, BV650–EPCAM, EGFP–AF488 and FGFR2–AF647, respectively. Data were analysed using FlowJo (v.10.7.1, BD Biosciences).

### Lentiviral vectors and virus production
The SIN.LV.SF, SIN.LV.SF-T2A-Puro, SIN.LV.SF-GFP-T2A-Puro lentiviral vectors and the SIN.LV.SF-Cre (Lenti-Cre), SIN.LV.SF-P2A-Cre and pGIN sgPten–P2A-Cre lentivectors all encoding improved *cre* with mammalian codon usage (derived from pBOB-CAG-iCRE-SD, 12336, Addgene) and the last-mentioned encoding a validated sgRNA targeting E7 of *Pten* (sgPten) were all previously described[10,12,13,59]. Mouse *Fgfr2* (NM_201601.2) and *Myc* (NM_010849.4) were isolated from cDNA clones (*Fgfr2*, MC221076, OriGene; *Myc*, 8861953, Source BioScience) using Q5 High-Fidelity DNA Polymerase (M0491S, New England Biolabs) and primers with AgeI–SalI overhangs amplifying *Fgfr2$^{FL}$* or *Fgfr2$^{\Delta E18}$*, or BamHI–AgeI overhangs amplifying *Myc*. Amplicons were inserted into the SIN.LV.SF, SIN.LV.SF-T2A-Puro and SIN.LV.SF-P2A-Cre vectors, resulting in SIN.LV.SF-Fgfr2$^{FL}$, SIN.LV.SF-Fgfr2$^{\Delta E18}$, SIN.LV.SF-Fgfr2$^{FL}$-T2A-Puro, SIN.LV.SF-Fgfr2$^{\Delta E18}$-T2A-Puro, SIN.LV.SF-Fgfr2$^{FL}$-P2A-Cre, SIN.LV.SF-Fgfr2$^{\Delta E18}$-P2A-Cre, and SIN.LV.SF-Myc. Custom-synthesized gBlocks gene fragments of mouse *Ccnd1* (NM_001379248.1), *Fgf3* (NM_008007.2), *Ate1*-E11–E12 (NM_001271343.1), *Bicc1*-E3–E21 (NM_001347189.1) and *Tacc2*-E15–E21 (NM_206856.4), which were the homologues of human *FGFR2* fusion partner genes (Extended Data Fig. 2h), as well as human full-length *FGFR2$^{E18-C1}$* (NM_000141.4), *FGFR2$^{E18-C2}$* (XM_017015921.2), *FGFR2$^{E18-C3}$* (NM_001144913.1), *FGFR2$^{E18-C4}$* (NM_001144915.2), the IGR1 sequence (identified in TCGA-A8-A08A; Extended Data Fig. 4f) and E18 sequences resulting from frameshift mutations (Extended Data Fig. 2k) were purchased (Integrated DNA

Technologies). gBlocks gene fragments or parts thereof were assembled in the respective lentivectors using PCR amplification of backbone and insert(s) and the In-Fusion HD Cloning Plus kit (638911, Takara Bio) according to the manufacturer's recommendations. Point mutations in *FGFR2*, short deletions/insertions to generate gradual *Fgfr2* E18 truncations and introduction of the IGR2 sequence (identified in TCGA-BH-A203; Extended Data Fig. 4g) to *Fgfr2* were performed using the QuikChange Lightning Site-Directed Mutagenesis Kit (210519, Agilent). In-Fusion and site-directed mutagenesis primers were designed using SnapGene (v.5.2) and QuikChange Primer Design[60], respectively. All lentivectors were verified using Sanger sequencing. Concentrated lentiviral stocks were produced by transient co-transfection of four plasmids in HEK293T cells as previously described[61]. Viral titres were determined using the qPCR Lentivirus Titre Kit (LV900, Applied Biological Materials).

### Cell culture
HEK293T cells (CRL-3216, ATCC) were cultured in Iscove's modified Dulbecco's medium (31980, Thermo Fisher Scientific) containing 10% FBS and 100 IU ml$^{-1}$ penicillin–streptomycin. MCF7 (HTB-22), MDA-MB-134-VI (HTB-23), MDA-MB-231 (HTB-26), NCI-H716 (CCL-251), NMuMG (CRL-1636), KATO-III (HTB-103), SNU-1 (CRL-5971) and SNU-16 (CRL-5974, all ATCC) as well as MFM-223 (98050130, ECACC) and SUM-52PE (HUMANSUM-0003018, BioIVT) cells were cultured in DMEM/F-12 containing 10% FBS and 100 IU ml$^{-1}$ penicillin–streptomycin. All cell lines were previously authenticated by providers. No re-authentication was carried out for this study. To stably express the lentiviral GFP-T2A-Puro or FGFR2-T2A-Puro constructs, NMuMG and KATO-III cells were transduced with lentiviral supernatants at equal TU per ml in the presence of 8 µg ml$^{-1}$ Polybrene for 24 h. Transduced cells were selected with 2 µg ml$^{-1}$ puromycin (A11138, Thermo Fisher Scientific) for 5 days and subsequently grown in DMEM/F-12 containing 10% FBS, 100 IU ml$^{-1}$ penicillin–streptomycin, 1 µg ml$^{-1}$ puromycin and 10 µM Y-27632. Overexpression of lentiviral constructs was verified using RT–qPCR (Supplementary Table 4). All cell lines were cultured in standard incubators at 37 °C with 5% $CO_2$ and routinely tested for mycoplasma contamination using the MycoAlert Mycoplasma Detection Kit (LT07-218, Lonza).

### Gene silencing using siRNA
Human cells were transfected with Silencer Select Negative Control 1 or 2 siRNAs (siCo#1 and #2, 4390844, 4390847, Thermo Fisher Scientific) or Silencer Select siRNAs designed with the GeneAssist Custom siRNA Builder (Thermo Fisher Scientific) to target shared exons (E5, E9, E15) among *FGFR2* isoforms (siFGFR2$^{E5/E9/E15}$), the 3′-UTR of E18-C1 of full-length *FGFR2* (siFGFR2$^{E18-C1}$), the 3′-UTR of E18-C3 of truncated *FGFR2$^{E18-C3}$* (siFGFR2$^{E18-C3}$) or the *FGFR2-COL14A1* fusion (siFGFR2-COL14A1). siFGFR2$^{E5}$ targeted endogenous *FGFR2* transcripts as well as *FGFR2* transcripts derived from lentiviral constructs. All other siRNAs specifically targeted endogenous *FGFR2* transcripts, because the *FGFR2* cDNA sequences used in the lentivectors lacked 3′-UTRs and contained silent mutations in E9 and E15 to prevent the binding of siFGFR2$^{E9/E15}$. A list of the custom-designed siRNA sequences is provided in Supplementary Table 10. siRNA (50 nM) was used in combination with the jetPRIME transfection reagent (114-15, Polyplus Transfection) as previously described[62].

### Drug-response curves
A total of 800 NMuMG or SNU-1 cells; 2,000 MCF7, MDA-MB-231, KATO-III, SNU-16, or SUM52PE cells; 3,000 MFM-223 or NCI-H716 cells; or 4,000 MDA-MB-134-VI cells per well were seeded in 96-well plates using DMEM/F-12 supplemented with penicillin–streptomycin and 10% FBS for human cell lines or 3% FBS for NMuMG. After 24 h, cells were treated with FGFRi for 4 days using vehicle (DMSO), AZD4547 (Astra-Zeneca), or pemigatinib (HY-109099), BGJ398 (HY-13311) or debio-1347

(HY-19957, all MedChemExpress) with a range of 0.1 nM to 100 μM. Usage of AZD4547, pemigatinib, BGJ398 and debio-1347 was previously described[63–66]. Cell viability was assayed using CellTiter-Blue Reagent (G808A, Promega) for 4 h and subsequently measuring fluorescence on the Infinite M Plex plate reader operated using the Tecan i-control software. Drug-response curves were modelled using [inhibitor] versus response with variable slope (four parameters) and least-squares regression in Prism (v.9.3.1, GraphPad Software).

## 2D colony-formation assay
A total of 5,000 KATO-III or MCF7 cells per well were seeded in six-well plates and, after 24 h, were treated with vehicle, 100 nM AZD4547 or 100 nM pemigatinib, or transfected with 50 nM siRNAs and cultured for 7 days. For KATO-III cells, six-well plates were precoated with laminin using RAC-11P cells[67] as previously described[53]. Cells were stained with crystal violet as previously described[53] and the plates were imaged using the GelCount colony counter (Oxford Optronix).

## 96-well cell growth assay
A total of 800 SNU-1 cells; 1,500 MCF7, MFM-223, KATO-III, SNU-16 or SUM52PE cells; or 3,000 NCI-H716 cells per well were seeded in 96-well plates and, after 24 h, were treated with vehicle or 100 nM AZD4547, pemigatinib, BGJ398 or debio-1347, or transfected with 50 nM siRNAs. Cell density was assayed over 8 days on sister plates using the CellTiter-Blue Reagent for 4 h and the Infinite M Plex plate reader operated using the Tecan i-control software.

## 3D soft agar colony-formation assay
Six- or twelve-well plates were precoated with 2 ml or 1 ml of 0.6% low-gelling-temperature agarose (A9414, Sigma-Aldrich) by diluting 3% agarose solution (in PBS) in DMEM/F-12 supplemented with 3% FBS and penicillin–streptomycin. The bottom layer was solidified at 4 °C for 30 min. NMuMG cells were passed through a 70 μm cell strainer and 20,000 or 10,000 single cells per well of the six- or twelve-well plates, respectively, were suspended in 2 ml or 1 ml of 0.35% low-gelling-temperature agarose in DMEM/F-12 supplemented with 3% FBS, penicillin–streptomycin and vehicle, 100 nM AZD4547 or 100 nM pemigatinib, and plated on top. The top layer was solidified at 4 °C for 30 min before transferring the plates to the incubator. The plates were imaged after 15 days using the GelCount colony counter and anchorage-independent growth was quantified using the integrated GelCount colony counting platform (v.1.1.2, Oxford Optronix).

## FACS analysis of cells
Cultured NMuMG cells were collected with 2 mM EDTA and passed through a 70 μm cell strainer prewetted with FACS buffer. Single cells were labelled with the LIVE/DEAD Fixable Violet Dead Cell Stain Kit and fixed, permeabilized, incubated with primary and secondary antibodies, and analysed as described for FACS analysis of mammary glands.

## RNA isolation and RT–qPCR
RNA from frozen mammary tumour pieces was isolated as previously described[10,68]. Cultured cells were lysed (72 h after siRNA transfection in case of human cell lines) in buffer RLY (BIO-52079, Bioline) containing 1% 2-mercaptoethanol. Total RNA extraction and DNase treatment of samples was performed using the ISOLATE II RNA Mini Kit (BIO-52072, Bioline) according to manufacturer's guidelines. Purified RNA was quantified using the DS-11 Series Spectrophotometer/Fluorometer (DeNovix) and subjected to reverse transcriptase reaction using the Tetro cDNA Synthesis Kit (BIO-65042, Bioline) with oligo (dT)$_{18}$ primers (tumour pieces) or random hexamer primers (cells). qPCR was performed using the SensiFAST SYBR Hi-ROX Kit (BIO-92005, Bioline) and the QuantStudio 6 Flex Real-Time PCR System (4485691, Thermo Fisher Scientific) operated with the QuantStudio Real-Time PCR Software (v.1.7.2, Thermo Fisher Scientific). Primers used were designed

using Primer-BLAST[69] and a list of which is provided in Supplementary Table 11. Relative quantified cDNA was normalized using either mouse *Hprt* (tumour pieces) or *Usf1* (cells) or human *USF1* as the housekeeping transcript.

## Protein isolation and western blotting
NMuMG cells were cultured in DMEM/F-12 starvation medium (0% FBS) for 48 h and treated with vehicle or 100 nM AZD4547 for 3 h. Cells were lysed in previously described RIPA buffer[53] containing Halt protease and phosphatase inhibitor cocktail (78440, Thermo Fisher Scientific). Protein concentrations were determined using the BCA Protein Assay Kit (23227, Thermo Fisher Scientific) and by measuring absorbance using the Infinite M Plex plate reader operated with the Tecan i-control software. Equal amounts of protein and the BlueEye Prestained Protein Marker (PS-104, Jena Bioscience) were separated on NuPAGE 4–12% Bis-Tris Mini Protein Gels (NP0323, NP0329, Thermo Fisher Scientific) and transferred overnight at 4 °C onto nitrocellulose membranes (88018, Thermo Fisher Scientific) in previously described transfer buffer[53]. The membranes were stained with Ponceau S solution (ab270042, Abcam) and imaged using Fusion FX (Vilber), blocked in 5% bovine serum albumin (BSA, A8022, Sigma-Aldrich) in PBS-T (0.05% Tween-20) and incubated with primary antibodies in 5% BSA in PBS-T overnight at 4 °C. The membranes were washed with PBS-T and incubated with secondary antibodies in 5% BSA in PBS-T for 1 h at room temperature. A list of the antibodies used (all validated for western blotting by the manufacturers) is provided in Supplementary Table 9. The membranes were washed in PBS-T and developed using SuperSignal West Pico PLUS Chemiluminescent Substrate or Femto Maximum Sensitivity Substrate (34580, 34095, Thermo Fisher Scientific). The membranes were imaged using Fusion FX operated with the Fusion FX7 Edge imaging system (v.18.05, Vilber), post-imaging processed with Photoshop 2022 (v.23.2.2, Adobe) using input levels and output levels, and band-intensities were measured with mean grey value in Fiji (v.1.0)[70]. Protein band intensities were normalized to β-actin and phosphoprotein bands were further normalized to corresponding total protein bands and to FGFR2 intensity.

## Proteomics of mouse cells and tumours
**Sample preparation.** Two (global phosphoproteomics) or three (phosphorylated-Tyr immunoprecipitation (p-Tyr IP) proteomics) 15 cm dishes of NMuMG cells expressing *GFP* or *Fgfr2* variants were collected in 3 ml urea lysis buffer[71]. Fresh-frozen samples of *Fgfr2* tumours collected in this study and *K14-cre;Brca^F/F^;Trp53^F/F^* (KB1P) and *KB1P;Mdr1a/b^−/−^* (KB1PM) tumours collected elsewhere[72] were mounted with Milli-Q H$_2$O and processed using a cryotome. Sections were collected to a final wet weight of up to 250 mg in urea lysis buffer (40× wet weight). Lysates were sonicated and cleared by centrifugation as previously described[73]. Protein concentrations were determined using the BCA Protein Assay Kit, and protein phosphorylation integrity was verified using western blotting and the p-Tyr-1000 antibody (8954, Cell Signaling Technology). To create a spectral library for protein expression analysis, for each setting, a ten-band in-gel-digestion experiment was performed and SDS gels were processed as described previously[74]. Per cell lysate sample, 45 μg total protein was loaded. Furthermore, 45 μg total protein of a mouse liver lysate[71] was added. Tumour lysates were prepared in 6 pools consisting of 4–7 individual samples each, and 60 μg total protein was loaded per pool. For global phosphoproteomics and p-Tyr IP experiments, in-solution protein digestion of an equivalent of 500 μg total protein (p-Tyr IP cells, 5 mg; p-Tyr IP tumours, 4 mg) using trypsin and desalting with Oasis HLB 1 cm$^3$ Vac Cartridge (186000383, Waters) was performed as previously described[71,73]. For global phosphoproteomic experiments, phosphopeptide enrichment was performed on the AssayMAP Bravo Platform (Agilent Technologies) using 5 μl Fe(III)-NTA immobilized metal affinity chromatography (IMAC) cartridges (G5496-60085, Agilent Technologies) starting from 200 μg desalted peptides

in 0.1% trifluoroacetic acid and 80% acetonitrile. Phosphopeptides were eluted in 25 µl 5% NH$_4$OH/30% acetonitrile. Phosphopeptide enrichment for KB1P(M) tumours was performed using titanium dioxide beads as previously described[71]. IP of p-Tyr-containing peptides was performed using the PTMScan p-Tyr-1000 Kit (8803, Cell Signaling Technology) as previously described[73].

**MS measurements.** For *Fgfr2* samples, phosphopeptides were separated using the Ultimate 3000 nanoLC–mass spectrometry (MS)/MS system (Thermo Fisher Scientific) equipped with a 50 cm × 75 µm ID Acclaim Pepmap (C18, 1.9 µm) column. After injection, peptides were trapped at 3 µl min$^{-1}$ on a 10 mm × 75 µm ID Acclaim Pepmap trap at 2% buffer B (80% acetonitrile, 0.1% formic acid) and separated at 300 nl min$^{-1}$ in a 10–40% gradient of buffer B over 110 min at 35 °C. Eluting peptides were ionized at a potential of +2 kVa into a Q Exactive HF mass spectrometer (Thermo Fisher Scientific) operated by Tune (v.2.11) and Xcalibur Software (v.4.3.73.11, OPTON-30965, both Thermo Fisher Scientific). Intact masses were measured at *m/z* 350–1,400 at a resolution of 120,000 (at *m/z* 200) in the Orbitrap system using an AGC target value of 3 × 10$^6$ charges and a maxIT of 100 ms. The top 15 peptide signals (charge-states 2$^+$ and higher) were submitted to MS/MS in the higher-energy collision cell (1.4 amu isolation width, 26% normalized collision energy). MS/MS spectra were acquired at a resolution of 15,000 (at *m/z* 200) in the Orbitrap system using an AGC target value of 1 × 10$^6$ charges, a maxIT of 64 ms and an underfill ratio of 0.1%, resulting in an intensity threshold of 1.3 × 10$^5$. Peptide separation for KB1P(M) samples was performed using a 40 cm × 75 µm (inner diameter) fused silica column custom packed with 1.9 µm 120 Å ReproSil Pur C18 aqua (Dr. Maisch). After injection, peptides were trapped at 6 µl min$^{-1}$ on a 10 mm × 100 µm (inner diameter) trap column packed with 5 µm 120 Å ReproSil Pur C18 aqua at 2% buffer B and separated at 300 nl min$^{-1}$ in a gradient of 10–40% buffer B over 90 min. The LC column was maintained at 50 °C using a pencil column heater (Phoenix S&T). Eluting peptides were ionized at a potential of +2 kVa into a Q Exactive HF mass spectrometer operated by Tune and Xcalibur Software. Intact masses were measured at a resolution of 70,000 (at *m/z* 200) in the Orbitrap system using an AGC target value of 3 × 10$^6$ charges. The top 10 peptide signals (charge-states 2$^+$ and higher) were submitted to MS/MS in the higher-energy collision cell (1.6 amu isolation width, 25% normalized collision energy). MS/MS spectra were acquired at a resolution of 17,500 (at *m/z* 200) in the Orbitrap system using an AGC target value of 1 × 10$^6$ charges, a maxIT of 80 ms and an underfill ratio of 0.1%, resulting in an intensity threshold of 1.3 × 10$^4$. For *Fgfr2* and KB1P(M) samples, a dynamic exclusion was applied with a repeat count of 1 and an exclusion time of 30 s. For protein expression experiments, peptides (1 µg total peptides, desalted) were separated and eluted as described for *Fgfr2* phosphopeptides. The data independent acquisition (DIA)-MS method consisted of an MS1 scan from 350 to 1,400 *m/z* at a resolution of 120,000 (AGC target of 3 × 10$^6$ and 60 ms injection time). For MS2, 24 variable-size DIA segments were acquired at 30,000 resolution (AGC target 3 × 10$^6$ and auto for injection time). The DIA-MS method starting at 350 *m/z* included one window of 35 *m/z*, 20 windows of 25 *m/z*, 2 windows of 60 *m/z* and one window of 418 *m/z*, which ended at 1,400 *m/z*. Normalized collision energy was set at 28. The spectra were recorded in centroid mode with a default charge state for MS2 set to 3$^+$ and a first mass of 200 *m/z*. Spectral library data files were acquired with the same acquisition settings as for the phosphoproteomic experiments.

**(Phospho)-peptide quantification and data analysis.** For protein expression experiments, MS/MS spectra derived from data-dependent acquisition (DDA) mode of the in-gel digestion experiment were searched against the Swiss-Prot *Mus musculus* reference proteome (25,374 entries, canonical and isoforms, release 2021_10) using MaxQuant (v.2.0.3.0)[75,76] software with the default settings. Peptide identifications were propagated across samples with the match between runs (MBR) option

enabled. The MaxQuant msms.txt file was used to generate a spectral library using Spectronaut software (v.15.4.210913, Biognosys). Spectra derived from single sample measurements in DIA mode were first analysed library-free in Spectronaut (directDIA) using the Biognosys factory settings to create a second spectral library. For the final search of DIA data in Spectronaut, both libraries were assigned using the default settings using the protein LFQ method set to MaxLFQ, imputation option switched off and an automatic normalization strategy. The Spectronaut report was further processed with R. Single-sample gene set enrichment analysis (ssGSEA) was performed by the GenePattern platform[77] using the ssGSEA module (v.10.0.11)[78] and Hallmark gene sets from MSigDB (v.7.0)[79]. Missing values were imputed with a zero. For phosphoproteome experiments, phosphopeptide identification and quantification was performed as previously described[71] using the Swiss-Prot (*Fgfr2* samples) or the UniProt (KB1P(M) samples) *Mus musculus* reference proteomes (UniProt, 34,331 entries, canonical and isoforms, release 2015_06) and MaxQuant software. Phosphosites with a localization probability of <0.75 (class 1)[80] were discarded. The R package limma (v.3.52.1)[81] was used to perform differential expression analysis on class 1 phosphosite intensity data. For two-group comparisons, phosphosite intensity data were filtered for high data presence in at least one of the groups under comparison (cells, ≥75%; tumours, ≥50%). In the case of data presence in one group and absence in the other (phosphosite on/ off behaviour), only observations with a very high data presence in the 'phosphosite on' group were allowed (cells, 100%; tumours, ≥90%). In these cases, missing values were imputed in the 'phosphosite off' group with a zero. Fold change values were determined using the mean of each treatment group and the antilog value was calculated. If downstream analysis did not allow the presence of duplicated phosphosite amino acid windows, the entry with the lowest *P* value was used. Phosphosite signature enrichment analysis (PTM-SEA)[82] was performed with the GenePattern platform[77] using a seven-amino-acid sequence flanking the phosphosite as an identifier and the mouse kinase/pathway definitions of PTMsigDB (v.1.9.0)[82] with the default settings. When PTM-SEA was performed following a two-group comparison, the rank metric was derived by multiplying the sign of FCs with the −log$_{10}$-transformed *P* values calculated by limma. When PTM-SEA was performed on single samples, duplicated phosphosite amino acid windows were filtered for entries with the highest row-sum of intensities over all of the samples. The samples were ranked using the phosphosite intensities and missing values were imputed with a zero. To assign probable upstream kinases to differentially regulated phosphosites, the robust kinase activity inference (RoKAI) tool[83] was used with the default settings and the UniProt *Mus musculus* reference proteome. RoKAI kinase and kinase target tables were shortlisted (cells, FDR < 0.05, number of substrates ≥ 3; tumours, number of substrates ≥ 2), assigned to significantly changed phosphosites (−1.5 ≥ FC ≥ 1.5, *P* < 0.05) and selected subsets of these phosphosites were visualized.

### Analysis of *SB* transposon insertions in the mutagenesis screen
For the *SB* transposon insertional mutagenesis screen in ref. [10], mapping of *SB* insertions and calculation of insertion clonalities using next-generation sequencing of genomic DNA from *SB*-containing tumours was described in detail[10]. In brief, the relative clonality scores of *SB* insertions were calculated by normalizing each unique ligation score between genomic DNA and a *SB* cassette insertion to the highest ligation score within a given tumour sample. Then, each *SB* insertion was assigned a score between 0 (no insertion) and 1 (fully clonal insertion). Tumours with at least one relative insertion clonality score for *Fgfr2* of ≥0.25 were defined as tumours containing *SB* insertion(s) in *Fgfr2* (*n* = 65 tumours; total, *n* = 123 tumours).

### Analysis of RNA-seq data from *SB* tumours
Published RNA-seq data generated from tumours of the *SB* transposon insertional mutagenesis screen[10] were used to derive *Fgfr2* gene- and

exon-level expression as well as splice junction information. Gene fusions affecting *Fgfr2* in tumours with *SB* insertions were previously identified[11]. To quantify the expression of *SB* transposons in *Fgfr2*, customized fasta and gtf files were constructed for individual tumours by inserting the *SB* transposon sequence at the genomic position and according to its orientation as previously mapped[10]. Sequencing reads were then mapped on the basis of the customized fasta and gtf files using STAR (v.2.7.2)[84]. Splice junctions between *Fgfr2*-E17 and the *SB* transposon were quantified using SJ.out.tab obtained from STAR alignment. To determine the usage of *Fgfr2*-I17-inserted *SB* transposons as splice acceptors, the ratio of junction reads spanning from E17 to the *SB* transposon versus E18 was computed. Integrated Genomics Viewer (IGV, v.1.11.0)[85] was used to generate sashimi plots.

## Analysis of WGS data from the HMF
WGS data on metastatic solid tumours were obtained from the HMF (data access request DR-138) through their Google cloud computing platform and analysed based on their bioinformatics pipeline (https://github.com/hartwigmedical/pipeline5) designed to detect all types of somatic alterations including structural variants and CNAs as previously described[23]. In brief, sequencing reads were mapped against the human reference genome GRCh37 using Burrows–Wheeler Alignment (BWA-MEM, v.0.7.5a)[86]. Somatic structural variants were called with GRIDSS (v.1.8.0)[87] and CNAs and tumour purity were estimated using PURPLE (v.2.43)[88]. Finally, LINX (v.1.9)[88] was performed to annotate events and to construct derivate chromosome structures. On the basis of the PURPLE output, samples containing structural variant BPs within the *FGFR2* genomic region were considered for further structural variant analyses (n = 266 total BPs and 196 unique BPs in 86 tumour samples; Fig. 1f). To annotate structural variants, the location and orientation of both BP sides (*FGFR2* and its partner) were used to determine RE types. Among the RE partners, the gene encoding the longer protein sequence was used as RE classification backbone. The following RE types were defined: (1) in-frame fusion, both BP sides were located in the intronic regions of coding genes and the upstream and downstream exons adjacent to the BP were both in-frame (complete reading frame) or both BP sides were located in the exonic regions of coding genes and the fused sequence was in-frame; (2) frame unknown RE, both BP sides were located in the intronic regions of coding genes and either the upstream or the downstream exon adjacent to the BP was out of frame (incomplete reading frame), or one or both BP sides were located in exonic regions of coding genes and the fused sequence was out of frame. Any of these cases made the reading frame unpredictable (unknown). (3) RE with intergenic space, one BP side located to *FGFR2* and the other BP side located to a non-coding IGR; (4) out-of-strand RE, both BP sides were located in the coding regions of genes. The gene upstream to the BP (*FGFR2*) was supported by a sense-oriented read sequence, whereas the gene downstream to the BP was supported by an antisense-oriented read sequence; (5) internal RE, both BP sides were located within the genomic region of *FGFR2*; (6) unresolved REs, the gene upstream to the BP was supported by antisense-oriented read sequences or the REs contained single breakends. Unresolved REs were excluded, resulting in a refined list of samples containing *FGFR2* REs (n = 93 REs in 55 tumour samples; Extended Data Fig. 1g,h). For the samples with multiple *FGFR2* REs, the relative allele frequency of each RE was computed using the ploidy level inferred by LINX. An I17/E18 RE allele frequency of >15% was used as a threshold to define samples with *FGFR2* REs causing E18 truncations (E18-truncating, n = 20; others, n = 35; Extended Data Fig. 1g,h and Supplementary Table 1). *FGFR2* copy number (CN) gains of >5 were defined as amplifications. Among the samples with *FGFR2* CN segment BPs at I17, samples with E1–E17 CN ($CN_{E1–E17}$) > 5 and $CN_{E1–E17} − CN_{E18}$ > 2 were defined as *FGFR2*-E1–E17 partially amplified. A few samples were expressing an *FGFR2* in-frame fusion gene based on RNA-seq, but showed discordant RE types in WGS. In these cases, in-depth annotation of the WGS data was performed using LINX to infer the plausible structures of derivate chromosomes constructed by complex RE events.

## Analysis of RNA-seq data from the HMF
Raw RNA-seq data on metastatic solid tumours were obtained from the HMF (data access request DR-138). Sequencing reads were mapped to the human reference genome GRCh38 (Gencode v32 CTAT) using STAR (v.2.7.2)[84] with the recommended parameters to subsequently run STAR-Fusion (v.1.8.1)[89]. STAR-Fusion was executed with chimeric alignment information (Chimeric.out.junction) obtained from STAR and GRCh38 Gencode v32 CTAT. Chimeric alignments from STAR and gene fusions from STAR-Fusion were inspected for RNA-seq alignments supporting the REs identified in WGS. For the samples with in-frame fusions identified in WGS, upstream and downstream exon numbers of the fusion gene inferred from STAR-Fusion were matched to the fusion found in WGS. For the samples with other types of REs identified in WGS, chimeric reads spanning the upstream exon and the downstream exon (out-of-frame REs), the downstream intergenic sequence (REs with intergenic space) or the downstream antisense gene sequence (out-of-strand REs) were mined from the 'Chimeric.out.junction' file and matched to the REs found in WGS. Genome coordinates were converted from GRCh37 to GRCh38 using UCSC Lift Genome Annotations (https://genome.ucsc.edu/cgi-bin/hgLiftOver). IGV (v.1.11.0)[85] was used to generate sashimi plots.

## Analysis of RNA-seq data from TCGA
Among the 10,344 TCGA samples[90], we preselected samples potentially expressing $FGFR2^{\Delta E18}$ on the basis of several criteria: the presence of (1) *FGFR2* amplifications or (2) truncating mutations in *FGFR2*-E18, (3) shifts in CN segment values in *FGFR2*-I17, (4) a lack of *FGFR2*-E18 expression, (6) usage of *FGFR2*-E18-C3 or -E18-C4, and/or (6) previously annotated *FGFR2* fusions[91]. *FGFR2* amplification and mutation information was obtained from the cBioPortal[92]. CN segment files for CN break information and exon-level expression data were obtained from the NCI-GDC data portal (https://portal.gdc.cancer.gov/). Among the samples with *FGFR2* CN BPs at I17, samples with $CN_{E1–E17}$ segment values ($\log_2[CN/2]$) > 0.3 (typical GISTIC threshold for amplifications) and $CN_{E1–E17} − CN_{E18}$ > 0.3 were defined as *FGFR2* E1–E17 partially amplified. To select samples with loss of *FGFR2*-E18 expression, E18 expression was normalized to the median expression of E1–E17. Tumour samples showing lower *FGFR2*-E18 expression compared with the minimum expression observed in TCGA normal tissue samples were selected. To evaluate E18-C3 and E18-C4 use, we obtained splice junction read counts from the NCI-GDC data portal. *FGFR2*-E17 to E18-C3 and E18-C4 spanning read counts were divided by total junction reads from *FGFR2*-E17 to calculate E18-C3 and E18-C4 use. Tumour samples showing higher *FGFR2*-E18-C3 or -E18-C4 use compared with the maximum usage observed in TCGA normal tissue samples were selected. In total, the selection process yielded 165 samples for which raw RNA-seq data were downloaded from the NCI-GDC data portal using TCGAbiolinks (v. 2.14.1)[93]. Sequencing reads were mapped to the human reference genome GRCh38 (Gencode v32 CTAT) using STAR (v.2.7.2)[84] with the recommended parameters to subsequently run STAR-Fusion (v.1.8.1)[89]. STAR-Fusion was executed with chimeric alignment information (Chimeric.out.junction) derived from STAR to obtain high-confidence in-frame and out-of-frame (frameshift or fusion with non-coding RNA) gene fusions. STAR-Fusion uses only exon–exon spanning reads to detect gene fusions; we therefore used exon–intron/intron–intron spanning reads from the Chimeric.out.junction file to find non-canonical types of out-of-frame fusions applying several filtering steps. Chimeric spanning reads with *FGFR2* BPs were discarded, if we found (1) multiple chimeric alignments, (2) PCR duplicates and/or (3) mitochondrial/Immunoglobulin/HLA mapping. Out-of-frame REs were defined by either exon–exon spanning reads resulting in frameshift or fusion with non-coding RNA (STAR-Fusion)

or exon–intron/intron–intron spanning reads (STAR chimeric alignments). Intergenic REs were defined by spanning reads between *FGFR2* and an IGR. Out-of-strand REs were defined by spanning reads between *FGFR2* and an antisense partner gene. REs with recurrent BP support were considered (spanning read count > 2). For the samples with multiple *FGFR2* REs, the relative expression of each RE was computed on the basis of the supporting junction read counts. An E17 junction read frequency of >15% was used as the threshold to define samples with *FGFR2^ΔE18* REs. IGV (v.1.11.0)[85] was used to generate sashimi plots.

### Analyses of CCLE, CTRPv2 and GDSC pharmacogenomic datasets

Mutation, CN, gene expression, exon usage ratio and fusion data for cell lines of the Broad Institute Cancer Cell Line Encyclopedia (CCLE) were obtained from the CCLE data portal[30]. *FGFR2/3* missense hotspot mutations were selected in agreement with previous annotations[92,94] and, in the case of *FGFR2*, based on the FMI cohort (Extended Data Fig. 2a). Missense mutations affecting the following amino acids were considered to be hotspots: *FGFR2*, Ser252, Cys382, Asn549, Lys659; *FGFR3*, Arg248, Ser249, Tyr373, Lys650. CN data were obtained as $\log_2[CN/2]$ values, and $\log_2[CN/2] \geq 2$ was considered to be an amplification. *FGFR* fusion data (CCLE_Fusions_unfiltered_20181130.txt) were further cleaned by applying the following filters: (1) FFPM > 0.1, (2) spanning fragment count $\geq 5$ and (3) expression value RPKM $\geq 1$. *FGFR2/3* was considered to be E18-truncated if cell lines contained *FGFR2/3* fusions with I17 BPs or exhibited high *FGFR2*-E18-C3 use ($P < 0.01$ derived from $Z$-score normalization of exon usage ratio) among the samples with robust expression of *FGFR2*. To compute composite expression of FGF receptors, *FGFR1–4* expression was normalized by the geometric mean of each receptor among all of the samples and summed as previously described[33]. Drug-response data for AZD4547 and PD173074 were obtained from the Cancer Therapeutics Response Portal (CTRP) v2 deposited in the PharmacoDB database[31,95] and from the Genomics of Drug Sensitivity in Cancer (GDSC) database[32], respectively. Integrated area under the sigmoid-fit concentration-response curve values were used to evaluate the association between *FGF*/*FGFR* status and drug sensitivity.

### Low-coverage WGS of human cancer cell lines

Genomic DNA from cultured cells was isolated using the ISOLATE II Genomic DNA Kit (BIO-52066, Bioline) according to the manufacturer's guidelines. Low-coverage WGS was performed as previously described[57]. Libraries were sequenced with 65 bp single reads using the HiSeq 2500 System with V4 chemistry (Illumina) operated by the HiSeq Control Software (v.2.2.68, Illumina). Sequencing reads were mapped to the human reference genome GRCh38 using BWA-MEM (v.0.7.5a)[86]. Reads with mapping quality lower than 37 were excluded. The resulting alignments were analysed with QDNAseq (v.1.14.0) using sequence mappability and GC content correction and a bin size of 20,000 bp to generate segmented CN values[96].

### RNA-seq analysis of human cancer cell lines

RNA-seq analysis of cultured cells was performed as previously described[68]. In brief, cells were lysed in Buffer RLT (79216, Qiagen) containing 1% 2-mercaptoethanol. Total RNA extraction was performed using the RNeasy Mini Kit (74104, Qiagen) according to the manufacturer's guidelines. The quality and quantity of RNA was assessed using the 2100 Bioanalyzer system and a Nano chip (Agilent). RNA samples with RIN > 8 were processed for polyA-stranded library preparation using the TruSeq RNA Library Prep Kit v2 (RS-122-2001/2, Illumina) according to the manufacturer's instructions, quality-checked with the 2100 Bioanalyzer system using a 7500 chip, and pooled equimolar into a 10 nM sequencing stock solution. Libraries were sequenced with 100 bp paired-end reads using the HiSeq 2500 System with V4 chemistry and operated by HiSeq Control Software. Sequencing reads were mapped

to the human reference genome GRCh38 (Gencode v32 CTAT) using STAR (v.2.7.2)[84] with the recommended parameters to subsequently run STAR-Fusion (v.1.8.1)[89]. Gene- and exon-level expression read counts were quantified by featureCounts (v.1.6.2)[97] on the basis of gene structures defined in GRCh38. Genes with CPM values greater than 1 in at least 10% of the total number of samples were considered expressed and used for downstream analysis. Read counts for expressed genes were normalized by trimmed mean of $M$-value (TMM) method using edgeR (v.3.26.6)[98,99]. To detect *FGFR2* gene fusions and REs from RNA-seq, we followed the approach as described for TCGA RNA-seq analysis.

### Hybrid-capture RNA-seq analysis of FFPE samples

Total RNA from FFPE samples was isolated using the RNeasy DSP FFPE Kit (73604, Qiagen) according to the manufacturer's guidelines. The quality and quantity of RNA was assessed using the Agilent TapeStation system and High Sensitivity D1000 Reagents (Agilent). A total of 20 ng of fragmented total RNA was used for Illumina-compatible cDNA library preparation. First, total RNA was used for reverse transcription and first-strand cDNA synthesis. After end-repair and adapter ligation, cDNA sequences were selected for enrichment of exonic sequences using biotinylated target specific probes as provided in the TruSeq RNA Exome kit (Illumina). Standard RNA-seq libraries were generated using captured/exome-enriched cDNA. Purified cDNA sequences were amplified using barcoded primers for different samples. Purified libraries were quantified using Qubit Flex Fluorometer (Thermo Fisher Scientific) and sequenced with 2 × 150 bp configurations using the NextSeq 500 or the NovaSeq 6000 Systems (Illumina) operated by NextSeq (v.2.0.2) and NovaSeq (v.1.7.5) control software, respectively. STAR-Fusion (v.1.8.1)[89] and the human reference genome GRCh37 were used for RNA fusion detection with the default parameters. STAR (v.2.7.3a)[84] and RSEM (v.1.3.0)[100] were used for gene and transcript quantification using the default parameters. LeafCutter (v.0.2.9)[101] and STAR-produced bam files were used to examine intron excision counts for splicing variants. IGV (v.1.11.0)[85] was used to generate sashimi plots.

### PDX models

**Model selection and analysis.** PDX models were previously characterized by Crown Bioscience[102] and are described in the HuPrime PDX collection (https://www.crownbio.com/oncology/in-vivo-services/patient-derived-xenograft-pdx-tumor-models). PDX models were selected on the basis of (1) *FGF3/4/19* amplification, (2) *FGFR2/3* missense hotspot mutations, (3) *FGFR1/2/3* amplification, (4) high expression of *FGFR1/2/3/4* and/or (5) expression of an *FGFR* fusion gene. The PDX models KI0551, LI0612 and LU1901 were included as controls, because each contained a *MET* oncogenic amplification potentially rendering tumours resistant to FGFRi[55,103]. CN and mutation data generated by whole-exome sequencing and raw RNA-seq data of the selected PDX models (Fig. 4c) were obtained from the CrownBio-HuPrime data portal. Sequencing data were derived from non-treated PDXs. Sequencing reads were mapped to the human (GRCh38 Gencode v32 CTAT) and mouse (mm10 Gencode M23) reference genomes to filter out mouse-derived reads using Disambiguate (v.2018.05.03-6)[104]. The remaining human reads were analysed as described for the analysis of human cell line RNA-seq data, composite *FGFR1-4* expression was computed as described for the CCLE RNA-seq analysis, and *FGFR2* gene fusions and REs were detected as described for TCGA RNA-seq analysis. We also implemented fusions/REs previously annotated by Crown Biosciences and deposited in CrownBio-HuPrime into our analysis.

**Debio-1347 intervention study.** PDX fragments of 2–3 mm in diameter were injected subcutaneously into the right flank of 8-week-old female BALB/cAnNRj-*Foxn1^nu/nu* mice (HFK Bioscience and Shanghai Laboratory Animal Center), except for BL5001 and BL5002, for which 8-week-old female NOD.CB17-*Prkdc*^scid/NCrHsd mice (Envigo) were used. Mice were twice weekly weighed and monitored for tumour development

and, as soon as tumours reached a volume of 200–250 mm³ (measured in two dimensions using callipers; volume = length × width² × 0.5), the mice were randomly allocated to vehicle versus debio-1347 FGFRi treatment arms. The treatments were performed daily through oral gavage for 12–25 consecutive days using vehicle (1% Kollidon VA64 in demineralized water), 40 mg per kg debio-1347 (Debiopharm) and increased to 60 mg per kg during treatment (BL5001, BL5002), 60 mg per kg debio-1347 (BN2289, BL0597, BR0438, BR1115, CR3151, ES0136, ES0189, ES0204, ES0215, ES0218, ES2116, GA0114, LI0612, LI1035, LI1055, LU0755, PA1332) or 80 mg per kg debio-1347 (CR1428, ES0042, GA0080, GA0087, GA1224, GA3055, GL0720, HN0366, HN0696, HN1420, KI0551, LU1302, LU1380, LU1429, LU1901, LU2504, PA3013). The treatment response was determined by relative treatment-to-control ratios ($\Delta T/\Delta C$). $\Delta T$ and $\Delta C$ are the mean volume difference between last treatment day and initial treatment day of the treated and control groups, respectively. All of the animal procedures were conducted at a Crown Bioscience SPF facility. All of the procedures related to animal handling, care and the treatment in this intervention study were performed according to guidelines approved by the Institutional Animal Care and Use Committee (IACUC) of Crown Bioscience following the guidance of the Association for Assessment and Accreditation of Laboratory Animal Care (AAALAC).

## Analysis of CGP data from FMI

**Hybrid-capture-based CGP.** Comprehensive genomic profiling (CGP) was performed on FFPE tumour tissue or blood samples prospectively collected during routine clinical care. Testing was performed in a Clinical Laboratory Improvement Amendments-certified, College of American Pathologists-accredited, New York State-regulated reference laboratory (Foundation Medicine). Approval for this study, including a waiver of informed consent and a Health Insurance Portability and Accountability Act waiver of authorization, was obtained from the Western Institutional Review Board (protocol 20152817). For 217,017 tumour tissue specimens, DNA (>50 ng) was extracted from FFPE specimens and next-generation sequencing was performed by FoundationOne companion diagnostic testing using hybridization-captured, adapter-ligation-based libraries to high, uniform coverage (>500×) for all coding exons of 315 or 324 cancer-related genes plus selected introns of 28 or 36 genes that are frequently rearranged in cancer, as previously described[105]. A total of 26,289 samples were similarly assayed; DNA was sequenced for 406 genes and selected introns of 31 genes involved in REs, and RNA was sequenced for 265 genes[106]. For 6,264 liquid samples, plasma was isolated from 20 ml of peripheral whole blood and ≥20 ng of circulating tumour DNA was extracted to create adapted sequencing libraries for coding exons of 70 genes before hybrid-capture and sample-multiplexed sequencing[107]. Results were analysed for base substitutions, short insertions and deletions, CN gains or losses, and REs. The companion diagnostic tests included probes against all *FGFR2* exons and *FGFR2*-I17.

**Data analysis.** FMI classifies *FGFR2* REs as fusions if the genomic BP is in the I17/E18 hotspot, if the predicted chimeric protein includes both an N terminus and a C terminus (in strand), and if the gene partner is either a previously described fusion partner (in-frame or frame unknown) or a novel gene partner predicted to be in-frame with *FGFR2*. I17/E18 hotspot out-of-strand REs, any REs with a BP in intergenic space and any REs with a BP in E1–E17 of *FGFR2* were classified as REs. Here we reclassified *FGFR2* REs as described for WGS data analysis. In brief, REs were defined as in-frame fusions if the genomic BP was in I17/E18 and if the frame of the fusion partner was predictable and in-strand. Frame-unknown REs, out-of-strand REs and REs with a BP in intergenic space were classified as non-canonical REs. *FGFR2* amplifications were called if ≥80% of the *FGFR2* targets were at an amplified CN (defined as ≥4 + median ploidy of the sample). Differential CN gains of E18 targets < E1–E17 targets were defined as *FGFR2*-E1–E17 partial amplifications. In samples with *FGFR2*

REs and co-amplification, low-level REs were discarded at a read threshold dependent on the amplification CN. *FGFR2*-E18-truncating nonsense and frameshift mutations were subgrouped into mutations affecting the proximal (E768–Y783) versus the distal (P784–T821) C terminus (encoded by E18) on the basis of the functional classifications of truncating mutations in this study (Fig. 2e and Extended Data Figs. 6a,h, 7a,b and 8a–c). I17/E18 in-frame fusions or non-canonical REs, E1–E17 partial amplifications, E18 splice-site mutations and/or proximal E18-truncating mutations were grouped as *FGFR2*-E18-truncating alterations. The four most common *FGFR2* missense mutations affecting Ser252, Cys382, Asn549 and Lys659 are referred to as hotspots throughout this study (Extended Data Fig. 2a), in agreement with previous annotations[92,94]. To establish the co-driver landscape of *FGFR2*-altered tumours, the top 30 driver genes concurrently altered (amplifications, deletions, and missense, truncating and splice-site mutations) in samples with *FGFR2* alterations (E18 truncations, E1–E18 full-length amplifications and/or missense hotspot mutations) were identified. The samples were grouped according to *FGFR2*-E18-truncating alterations, *FGFR2*-E1–E18 full-length amplifications or *FGFR2* missense hotspot mutations, and proportion $Z$-tests were used to identify co-driver genes significantly enriched in either of the 3 *FGFR2* alteration categories, both in the pan-cancer cohort as well as in the BRCA, CHOL, OV, COAD/READ, ESCA/STAD and LUAD/LUSC cohorts specifically. Fisher's exact tests were used to evaluate co-occurrence (odds ratio > 1) or mutual exclusivity (odds ratio < 1) of the co-driver genes in each of the 3 *FGFR2* alteration categories versus *FGFR2* WT samples in the pan-cancer cohort and in the BRCA cohort specifically.

## Analysis of self-interacting capacity among *FGFR2* RE partners

The SLIPPER algorithm predicts the interaction capacities of proteins[108]. It has been trained with seven different proteome databases (DIP, IntAct, MINT, BioGRID, PDB, MatrixDB and I2D) to establish the SLIPPER Golden Standard Dataset of potentially self-interacting proteins[108]. On the basis of this dataset, the proportion of self-interactors among unique proteins encoded by *FGFR2* RE partner genes identified in the FMI dataset was calculated. The self-interacting ability of *FGFR2* RE partners was also evaluated using the SLIPPER algorithm[108] itself. To identify specific self-interacting domains in *FGFR2* RE partners, domain–domain interaction information was obtained from the 3did[109] and PPIDM[110] databases, and domain enrichment analysis was performed with DAVID bioinformatic resources[111]. The Swiss-Prot *Homo sapiens* proteome (release 2021_04) was used as reference dataset for these analyses.

## Re-examination of FIGHT-202 study

Details on the study design, eligibility criteria, and efficacy and safety findings of FIGHT-202 (NCT02924376), a phase II, open-label, multicentre, global study of pemigatinib in patients with previously treated advanced or metastatic cholangiocarcinoma, with or without *FGF/FGFR* alterations, were previously published[9]. Before entering screening for trial eligibility, the patients were either prescreened for *FGF/FGFR* status using FoundationOne or patients provided a commercial FoundationOne report or an *FGF/FGFR* status report based on local testing, the latter of which required retrospective central confirmation through FoundationOne. In FIGHT-202, *FGFR2* REs were classified (fusions versus REs) on the basis of the FoundationOne report and biomarker definition as described above. In this reanalysis of the FIGHT-202 oncogenomic data, we used the alteration data provided by FMI to classify *FGFR2* amplification status and *FGFR2* REs by frame only. Five patients who were classified as having fusions on their FoundationOne report had *FGFR2* amplifications in conjunction with fusions in the alteration data; four patients classified as having fusions on their FoundationOne report had an unknown frame in the structured data; and two patients who were classified as having REs on their FoundationOne report were classified as in-frame in the alteration data. These discrepancies are due to FoundationOne

reporting rules and ongoing updates to the analysis and annotation pipeline used by FMI from the time of the original report to the time of the generation of the alteration data. Importantly, these changes do not affect the results from the primary efficacy cohort from FIGHT-202 but, rather, provide an alternative classification for this subset analysis.

### Statistics and reproducibility
Data of in vitro and in vivo experiments were analysed using Prism (v.9.3.1, GraphPad Software). Genomic and proteomic data were analysed using R (v.3.6.3–4.1.2). In vitro experiments were independently repeated at least twice, and all attempts at replication were successful. Across these, data on $n \geq 3$ independent replica were collected. No sample size calculations were performed. Sample sizes of mouse cohorts and for ex vivo analyses thereof (FACS analyses, H&E and IHC analyses, proteomics, RNA-seq and RT–qPCR) were based on previous calculations[10] or determined using G*Power software (v.3.1)[56], and were large enough to measure the effect sizes. Data were reproducible across mice or batches analysed, and all attempts at replication were successful. Sample sizes in the FIGHT-202 trial were based on previous calculations[9] and were large enough to measure the effect sizes. The statistical tests and multiple-testing correction models used are described in the corresponding figure legends. $P < 0.05$ was considered to be statistically significant. Except for $P < 0.0001$ and $P \geq 0.05$, exact $P$ values are always shown in the corresponding figure panels or, where indicated, in Supplementary Table 2.

### Reporting summary
Further information on research design is available in the Nature Research Reporting Summary linked to this article.

### Data availability
The low-coverage WGS and RNA-seq data of human cell lines generated in this study are available at the European Nucleotide Archive (ENA) under accession number PRJEB42514. The MS proteomic data and MaxQuant-generated text files generated in this study are available at the ProteomeXchange Consortium through the PRIDE database[112] under accession numbers PXD031711 for *Fgfr2* samples and PXD032007 for KB1P(M) samples. Sequencing data of *SB* tumours were previously published[10] and are available at ENA under accession number PRJEB14134. WGS and RNA-seq data from the HMF were downloaded from their Google cloud computing platform under data-sharing agreement DR-138, and can be obtained through standardized procedures and request forms online (https://www.hartwigmedicalfoundation.nl/en/). CGP data can be obtained from FMI on reasonable request (https://www.foundationmedicine.com/service/genomic-data-solutions). TCGA data[90] can be obtained from the NCI-GDC data portal (https://portal.gdc.cancer.gov/). Data from CCLE, CTRPv2 and GDSC are available through the respective data portals[30–32,95]. Details on PDXs can be obtained from the CrownBio-HuPrime data portal (https://www.crownbio.com/oncology/in-vivo-services/patient-derived-xenograft-pdx-tumor-models). The FIGHT-202 study was previously published[9]. Information on Incyte's clinical trial data sharing policy and instructions for submitting clinical trial data requests are available online (https://www.incyte.com/Portals/0/Assets/Compliance%20and%20Transparency/clinical-trial-data-sharing.pdf?ver=2020-05-21-132838-960). The human reference genome (GRCh38 Gencode v32 CTAT) used for RNA-seq data analysis is available in CTAT Genome Lib data resources (https://data.broadinstitute.org/Trinity/CTAT_RESOURCE_LIB). The SLIPPER list of self-interacting proteins was previously published[108]. Domain–domain interaction information from 3did[109] and PPIDM[110] are available online (https://3did.irbbarcelona.org and http://ppidm.loria.fr, respectively). Reference human and mouse Swiss-Prot proteome information is available in the UniProt database (https://www.uniprot.org). All data generated or analysed during this study are included in this

Article and its Supplementary Information. Source data are provided with this paper.

### Code availability
WGS data from the HMF were analysed using published computer codes (https://github.com/hartwigmedical/pipeline5)[23] including BWA-MEM (v.0.7.5a)[86], GRIDSS (v.1.8.0)[87], PURPLE (v.2.43)[88] and LINX (v.1.9)[88] to detect all types of somatic alterations, including structural variants and CNAs. BWA-MEM (v.0.7.5a)[86], R package QDNAseq (v.1.14.0)[96], R package TCGAbiolinks (v.2.14.1)[93], STAR (v.2.7.2)[84], STAR-Fusion (v.1.8.1)[89], featureCounts (v.1.6.2)[97], R package edgeR (v.3.26.6)[98,99], RSEM (v.1.3.0)[100], R package LeafCutter (v.0.2.9)[101], Disambiguate (v.2018.05.03-6)[104] and R package limma (v.3.52.1)[81] were integrated into custom computer codes to analyse the genomic and proteomic data in this study, which are available at https://doi.org/10.5281/zenodo.6630874 and https://doi.org/10.5281/zenodo.6630632, respectively.

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

**Acknowledgements** We thank D. Zimmerli and I. van der Heijden for experimental support and J. Houthuijzen and A. Moises Da Silva for providing mice; the staff at the NKI animal facility, the animal pathology facility, the flow cytometry facility and the genomics core facility and Rutgers Cancer Institute of New Jersey Biospecimen Repository and Histopathology Service Shared Resource for their technical support; the people from the transgenic and the preclinical intervention units of the Mouse Clinic for Cancer and Ageing (MCCA) at the NKI for their technical support performing the animal experiments; and the staff at Crown Bioscience for their technical support with performing the PDX experiments. This publication and the underlying study have been made possible in part on the basis of the data that the Hartwig Medical Foundation and the Center of Personalised Cancer Treatment (CPCT) have made available to the study as well as the data from the Broad Institute (CCLE and CTRPv2), the Wellcome Sanger Institute (GDSC) and TCGA Research Network. This work was funded by the Dutch Cancer Society (KWF 2017-61169 to J.B., L.F.A.W. and J.J.; KWF 2020-12894 to D.Z., E.C., L.F.A.W. and J.J.), the European Research Council (ERC Synergy project CombatCancer to J.J.; ERC AdG-883877 to S.R.), the Netherlands Organisation for Scientific Research (NWO-Middelgroot project 91116017 to C.R.J.), the VUmc-Cancer Center Amsterdam (infrastructure grant to C.R.J.), the Swiss National Science Foundation (SNF P2ZHP3_175027 to D.Z.; SNF 310030_179360 to S.R.), the German Research Foundation (DFG 319175447 to F.R.), and the US National Cancer Institute (NCI P30CA072720 to C.S.C. and S.G.). This work is part of the Oncode Institute, which is partly financed by the Dutch Cancer Society.

**Author contributions** D.Z., J.B., S.M.K., L.F.A.W. and J.J. conceived the ideas and designed the experiments. D.Z., J.Y., S.M.K., F.R., C.L., S.A. and E.W. performed the laboratory experiments. J.B. and F.R. conceived and performed the bioinformatic analyses. F.R., S.R.P, E.G. and R.R.d.G.-d.H. performed the MS-based proteomics experiments. C.L., T.E., M.B., B.S., J.S., B.d.K., A.P.D., E.v.d.B., E.S., N.P., U.B., R.d.K.-G. and B.v.G. performed the animal experiments. J.K.L., R.M., J.S.R. and S.M.A. provided FMI data. S.K. assessed the histology of mouse tumours and quantified IHC slides. I.M.S., H.Z., G.K.A.-A., L.F. and T.C.B. provided re-evaluation of FIGHT-202. C.S.C., G.M.R. and H.K. performed and analysed hybrid-capture RNA-seq. L.H. and M.H.v.M. generated *Fgfr2* GEMMs. R.d.B. supported RNA-seq analyses. J.R.d.R. mapped the *SB* insertion sites. M.v.d.V. supervised the animal experiments. S.R. and C.R.J. supervised the MS-based proteomics experiments. E.C. supported bioinformatic analyses of HMF data. A.V.C. designed and supervised the PDX experiments. D.Z., S.G., L.F.A.W. and J.J. supervised the bioinformatic analyses and the experiments. D.Z., J.B., S.G., L.F.A.W. and J.J. wrote the manuscript.

**Competing interests** J.K.L. and R.M are employees of Foundation Medicine and have equity in Roche. I.M.S., H.Z., L.F. and T.C.B. are employees of and have equity in Incyte. G.K.A.-A. has performed research for ActaBiologica, Agios, Astra Zeneca, Bayer, Beigene, Berry Genomics, BMS, Casi, Celgene, Exelixis, Genentech/Roche, Halozyme, Incyte, Mabvax, Puma, QED, Sillajen and Yiviva; and has consulted for Agios, Astra Zeneca, Autem, Bayer, Beigene, Berry Genomics, Celgene, CytomX, Debio, Eisai, Eli Lilly, Exelixis, Flatiron, Genentech/Roche, Gilead, Helio, Incyte, Ipsen, Loxo, Merck, MINA, Polaris, QED, Redhill, Silenseed, Sillajen, Sobi, Therabionics, Twoxar, Vector and Yiviva. J.S.R. is an employee of Foundation Medicine, has equity in Roche and is a board member of Celsius Therapeutics. A.V.C. is an employee of Debiopharm International. S.M.A. is an employee of and has equity in EQRX, is a former employee of Foundation Medicine, is a scientific advisory board member of iN Therapeutics and has consulted for Takeda. S.G. has received research funding from M2Gen, has equity in Inspirata and Silagene, and has consulted for Foghorn Therapeutics, Foundation Medicine, Inspirata, Merck, Novartis, Roche and Silagene; his spouse is an employee of and has equity in Merck. The other authors declare no competing interests.

**Additional information**
**Correspondence and requests for materials** should be addressed to Shridar Ganesan, Lodewyk F. A. Wessels or Jos Jonkers.

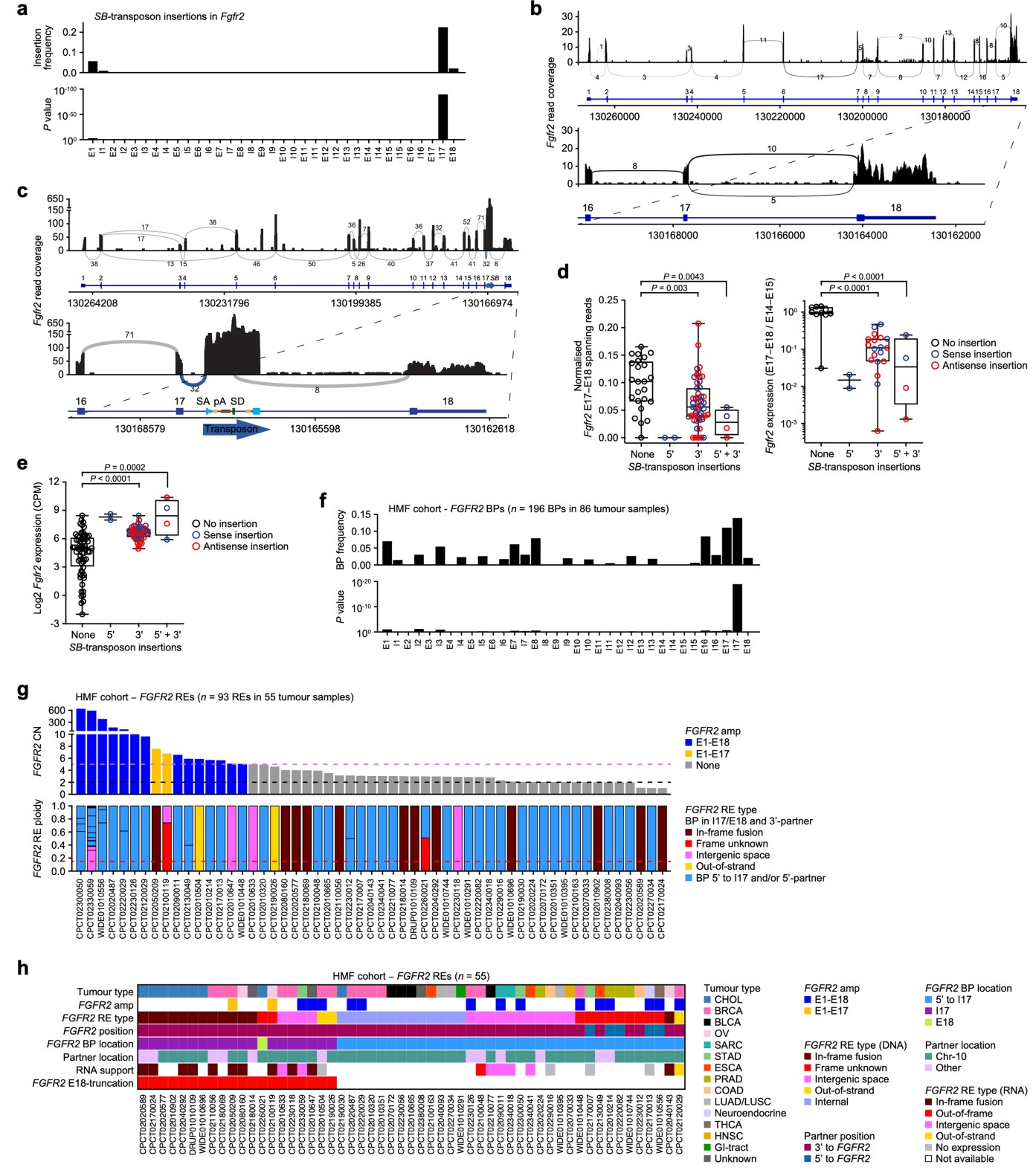

**Extended Data Fig. 1** | See next page for caption.

**Extended Data Fig. 1 | *SB*-insertions in *Fgfr2* and *FGFR2* REs found in the FMI cohort. a**, Normalized frequency (top panel) and enrichment significance (*P* values, bottom panel) of *Sleeping Beauty* (*SB*) transposon insertions (*n* = 81 insertions in 65 tumours) in each *Fgfr2* exon (E) and intron (I) as identified in mammary tumours from a *SB*-transposon in vivo screen[10]. *SB* insertion frequency was normalized by the kilobase of feature (exon/intron) length and total number of *SB*-insertions. **b**, **c**, Sashimi plots showing *Fgfr2* read coverage and junction reads plotted as arcs with indicated junction read counts of tumours with no *SB*-insertion (**b**) and an I17 sense *SB* insertion (**c**) in *Fgfr2*. SA, splice acceptor; SD, splice donor; pA, polyadenylation signal. **d**, Left panel, counts of *Fgfr2*-E17–E18 spanning reads (counts per million, CPM) normalized to *Fgfr2* expression (CPM) in *SB* tumour RNA sequencing (RNA-seq) profiles (none, *n* = 24; 5′-sense, *n* = 2; 3′-sense, *n* = 22; 3′-antisense, *n* = 27; 5′ + 3′-sense, *n* = 2; 5′ + 3′-antisense, *n* = 2); right panel, RT–qPCR to quantify *Fgfr2*-E17–E18 over *Fgfr2*-E14–E15 expression in *SB*-tumours (none, *n* = 10; 5′-sense, *n* = 2; 3′-sense, *n* = 8; 3′-antisense, *n* = 11; 5′ + 3′-sense, *n* = 2; 5′ + 3′-antisense, *n* = 2; individual dots represent mean of 3 independent measurements). **e**, Expression of *Fgfr2* (CPM) in *SB* tumour RNA-seq profiles (none, *n* = 64; 5′-sense, *n* = 2; 3′-sense, *n* = 30; 3′-antisense, *n* = 28; 5′ + 3′-sense, *n* = 2; 5′ + 3′-antisense, *n* = 2). **f**, Normalized frequency (top panel) and enrichment significance (*P* values, bottom panel) of *FGFR2* genomic rearrangement (RE) breakpoints (BPs) identified in 2,112 whole-genome sequencing (WGS) profiles from the Hartwig Medical Foundation (HMF) cohort in each exon/intron. BP frequency was normalized by the kilobase of feature (exon/intron) length and total number of BPs. **g**, *FGFR2* copy numbers (CN, top panel) and RE ploidy frequencies (bottom panel) in samples with *FGFR2* REs. *FGFR2* BPs resulting in unresolved REs were excluded generating a refined list of REs (*n* = 93 REs in 55 tumour samples). Dotted lines, black, normal CN; purple, amplified CN (> 5); red, RE ploidy frequency threshold (> 0.15) to call samples with E18-truncating *FGFR2* REs (E18-truncating, *n* = 20; others, *n* = 35). Amp, amplification. **h**, *FGFR2* RE types found in WGS profiles from HMF. RNA support indicates evidence for *FGFR2* REs in matching RNA-seq profiles. Empty fields, no RNA-seq data available. BLCA, bladder urothelial carcinoma; BRCA, breast invasive carcinoma; CHOL, cholangiocarcinoma; chr, chromosome; COAD, colon adenocarcinoma; ESCA, oesophageal carcinoma; GI, gastro-intestinal; HNSC, head and neck squamous cell carcinoma; LUAD, lung adenocarcinoma; LUSC, lung squamous cell carcinoma; OV, ovarian serous cystadenocarcinoma; PRAD, prostate adenocarcinoma; SARC, sarcoma; STAD, stomach adenocarcinoma; THCA, thyroid carcinoma[90]. Data in **d**, **e** are represented as median (centre line) ± interquartile range (IQR, 25th to 75th percentile, box) and ± full range (minimum to maximum, whiskers). *P* values were calculated with one-tailed binomial tests (**a**, **f**) or one-tailed one-way analysis of variance (ANOVA) and Tukey's multiple-testing corrections (**d**, **e**).

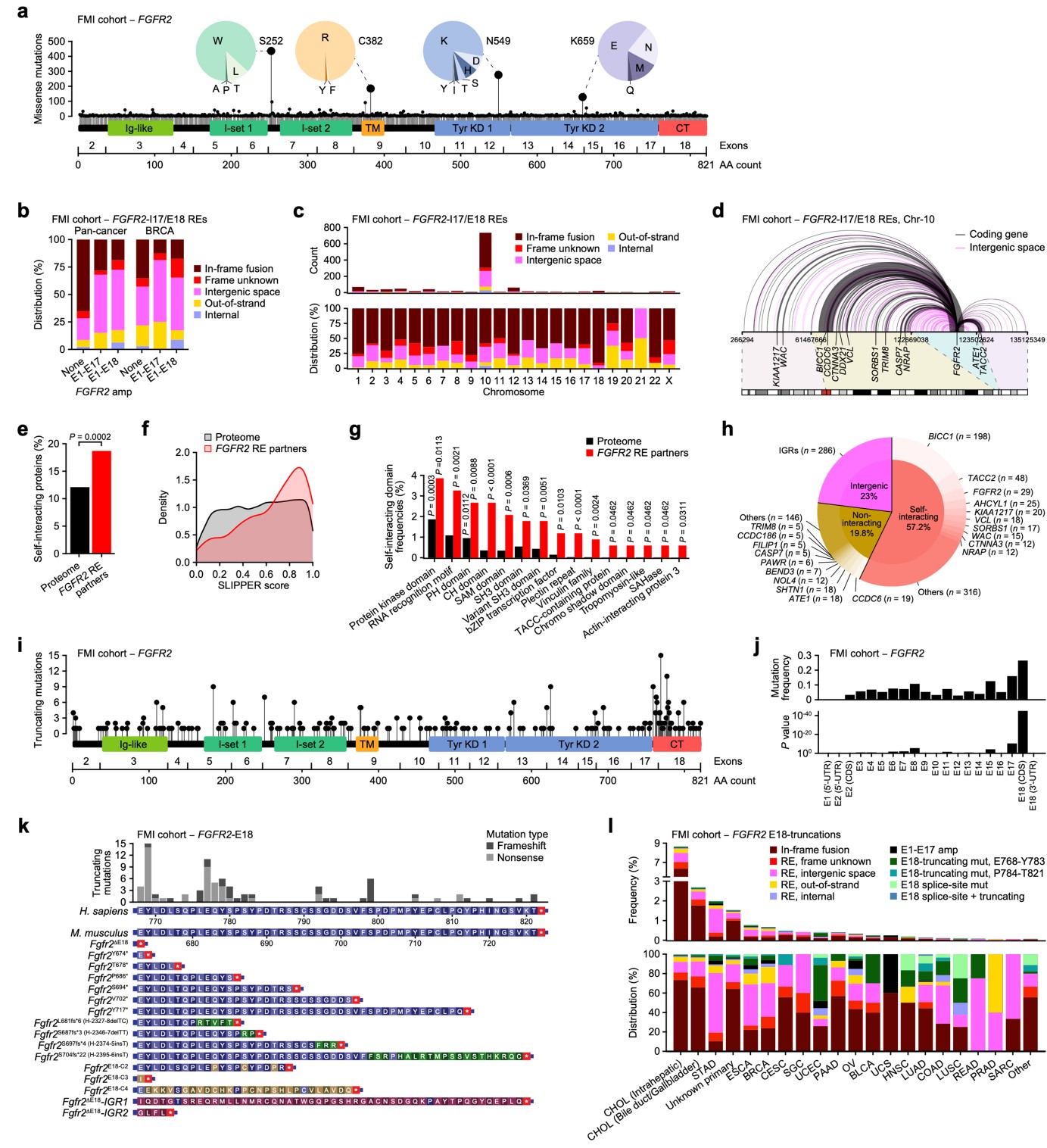

**Extended Data Fig. 2** | See next page for caption.

**Extended Data Fig. 2 | *FGFR2* alterations found in the HMF cohort.**

**a**, Lollipop plot of *FGFR2* missense mutations identified in the Foundation Medicine (FMI) pan-cancer cohort (249,570 diagnostic hybrid-capture panel-seq profiles). The top four recurrent mutations (Ser 252, Cys 382, Asn 549, Lys 659) are referred to as hotspots in this study. **b**, Distribution of *FGFR2*-I17/E18 RE types in FMI samples with *FGFR2* normal CN, E1-E17 amp, and E1-E18 amp. **c**, Total numbers and distributions of *FGFR2*-I17/E18 RE types across chromosomes according to RE partner location. **d**, Linear chr-10 map depicting intrachromosomal *FGFR2*-I17/E18 REs. Thickness of arcs is proportional to the recurrence of the corresponding RE partners. Light/dark grey and red bars denote ideogram and centromere of chr-10. **e**, Percentage of unique proteins with self-interacting capacity among *FGFR2* RE partners (*n* = 337) versus the human proteome (*n* = 20,385). Based on the SLIPPER Golden Standard Dataset of self-interactors[108]. **f**, Distribution of self-interaction scores among *FGFR2* RE partners using the SLIPPER algorithm[108]. **g**, Enrichment of self-interacting protein domains among *FGFR2* RE partners using DAVID[111]. **h**, Recurrence of *FGFR2* RE partners grouped by presence of self-interacting domains. Full list of RE partners is disclosed in Supplementary Table 2. IGRs, intergenic regions.

**i**, **j**, Lollipop plot (**i**) and normalized frequency (top panel) and enrichment significance (*P* values, bottom panel) (**j**) of *FGFR2*-truncating mutations identified in the FMI cohort. Mutation frequency was normalized by the kilobase of feature (exon/intron) length and total number of mutations. AA, amino acid; CDS, coding sequence; CT, C terminus; TM, trans-membrane; UTR, untranslated region. **k**, Distribution of *FGFR2*-E18-truncating mutations identified in the FMI cohort and corresponding cloned mouse *Fgfr2* variants representing most frequent human (H) *FGFR2*-E18 nonsense and frameshift (fs) mutations. C terminus sequences of cloned noncanonical E18-truncated *Fgfr2* (*Fgfr2^{ΔE18}*) variants are also displayed. IGR1 and IGR2 are based on TCGA-A8-A08A and TCGA-BH-A203 in Extended Data Fig. 5f, g. **l**, Frequencies (top panel) and distributions (bottom panel) per tumour type of E18-truncating *FGFR2* alterations found in the FMI cohort. CESC, cervical squamous cell carcinoma and endocervical adenocarcinoma; mut, mutation; PAAD, pancreatic adenocarcinoma; READ, rectum adenocarcinoma; SGC, salivary gland carcinoma; UCEC, uterine corpus endometrial carcinoma; UCS, uterus carcinosarcoma. *P* values were calculated with a one-tailed proportion *z*-test (**e**), one-tailed Fisher's exact tests (**g**), or one-tailed binomial tests (**j**).

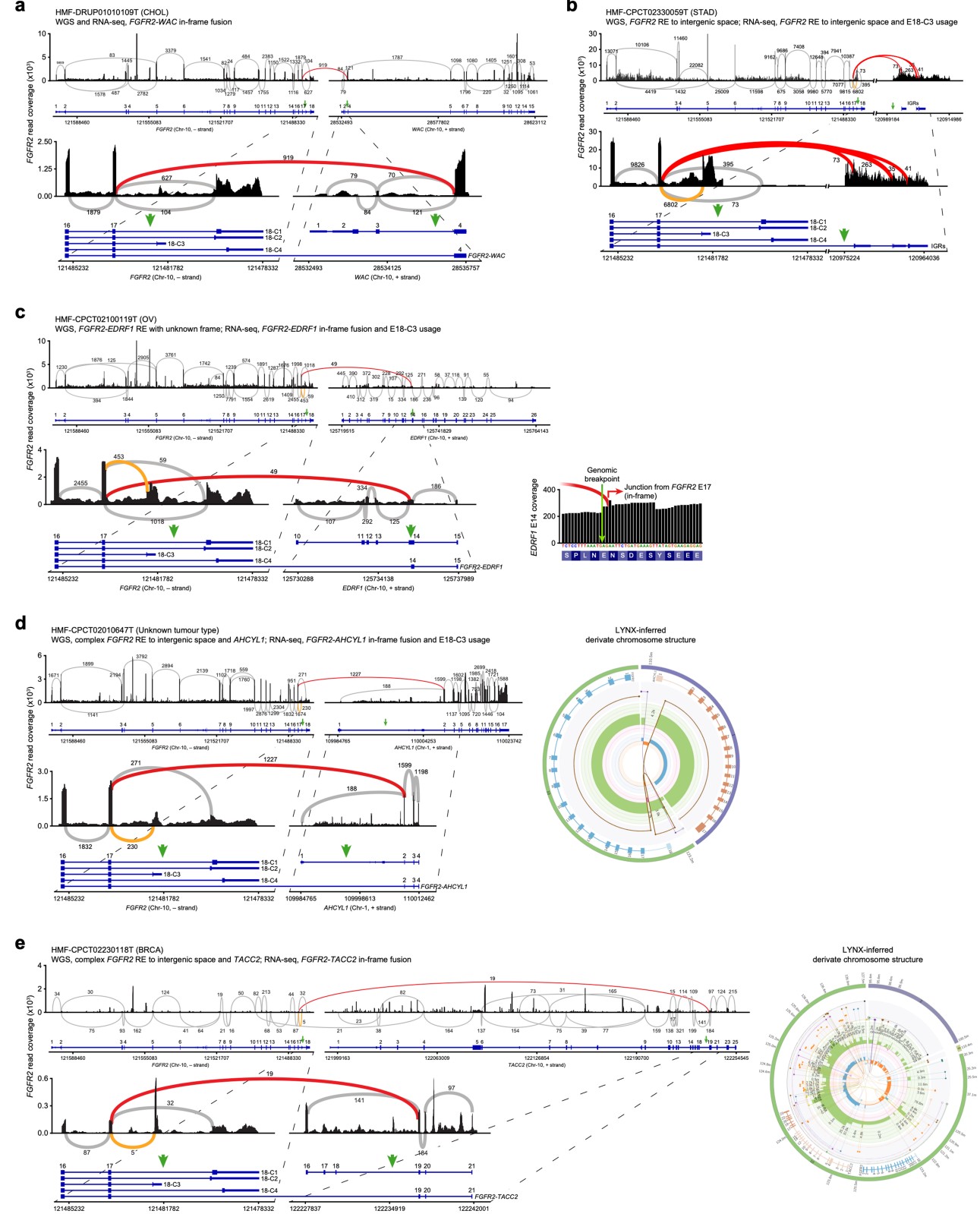

**Extended Data Fig. 3** | See next page for caption.

**Extended Data Fig. 3 | Expression of E18-truncating *FGFR2* variants in HMF samples. a**, Sashimi plot showing *FGFR2* read coverage and junction reads of the HMF sample DRUP01010109T (CHOL). *FGFR2-WAC* in-frame fusion identified with WGS and *FGFR2*-E17 to *WAC*-E4 junction confirmed with RNA-seq. **b**, Sashimi plot showing *FGFR2* read coverage and junction reads of the HMF sample CPCT02330059T (STAD). *FGFR2*-I17 RE to intergenic space identified with WGS and *FGFR2*-E17 to intergenic region (IGR) junctions and *FGFR2*-E18-C3 usage found with RNA-seq. **c**, Sashimi plot showing *FGFR2* read coverage and junction reads of the HMF sample CPCT02100119T (OV). *FGFR2-EDRF1* frame unknown RE identified with WGS and *FGFR2*-E17 to *EDRF1*-E14 in-frame junction and *FGFR2*–E18-C3 usage found with RNA-seq. **d**, Sashimi plot showing *FGFR2* read coverage and junction reads of the HMF sample CPCT02010647T (unknown tumour type). *FGFR2*-I17 RE to intergenic space identified with WGS and discordant *FGFR2-AHCYL1* in-frame fusion with *FGFR2*-E17 to *AHCYL1*-E2 junction and *FGFR2*-E18-C3 usage found with RNA-seq. **e**, Sashimi plot showing *FGFR2* read coverage and junction reads of the HMF sample CPCT02230118T (BRCA). *FGFR2*-I17 RE to intergenic space identified with WGS and discordant *FGFR2-TACC2* in-frame fusion with *FGFR2*-E17 to *TACC2*-E19 junction found with RNA-seq. Reconstructed derivate chromosomes using LINX[88] are displayed for CPCT02010647T (**d**) and CPCT02230118T (**e**) and depict complex *FGFR2* REs involving intergenic space and ultimately resolving to *AHCYL1*-E2 (**d**) and *TACC2*-E19 (**e**). Green arrows indicate BPs identified with WGS. E18-C1, canonical E18 of *FGFR2^FL^*; E18-C2/C3/C4, alternative *FGFR2*-E18.

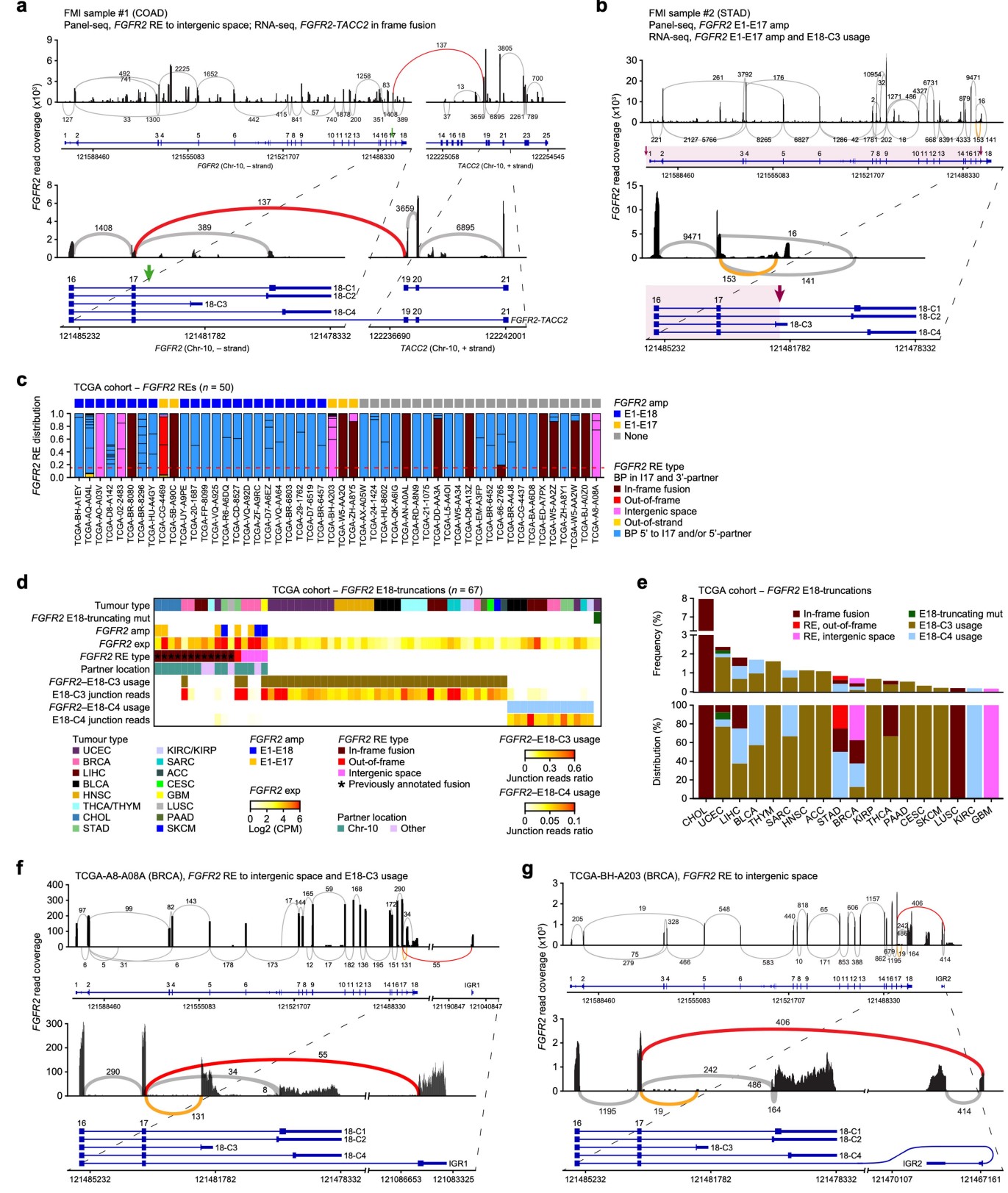

**Extended Data Fig. 4** | See next page for caption.

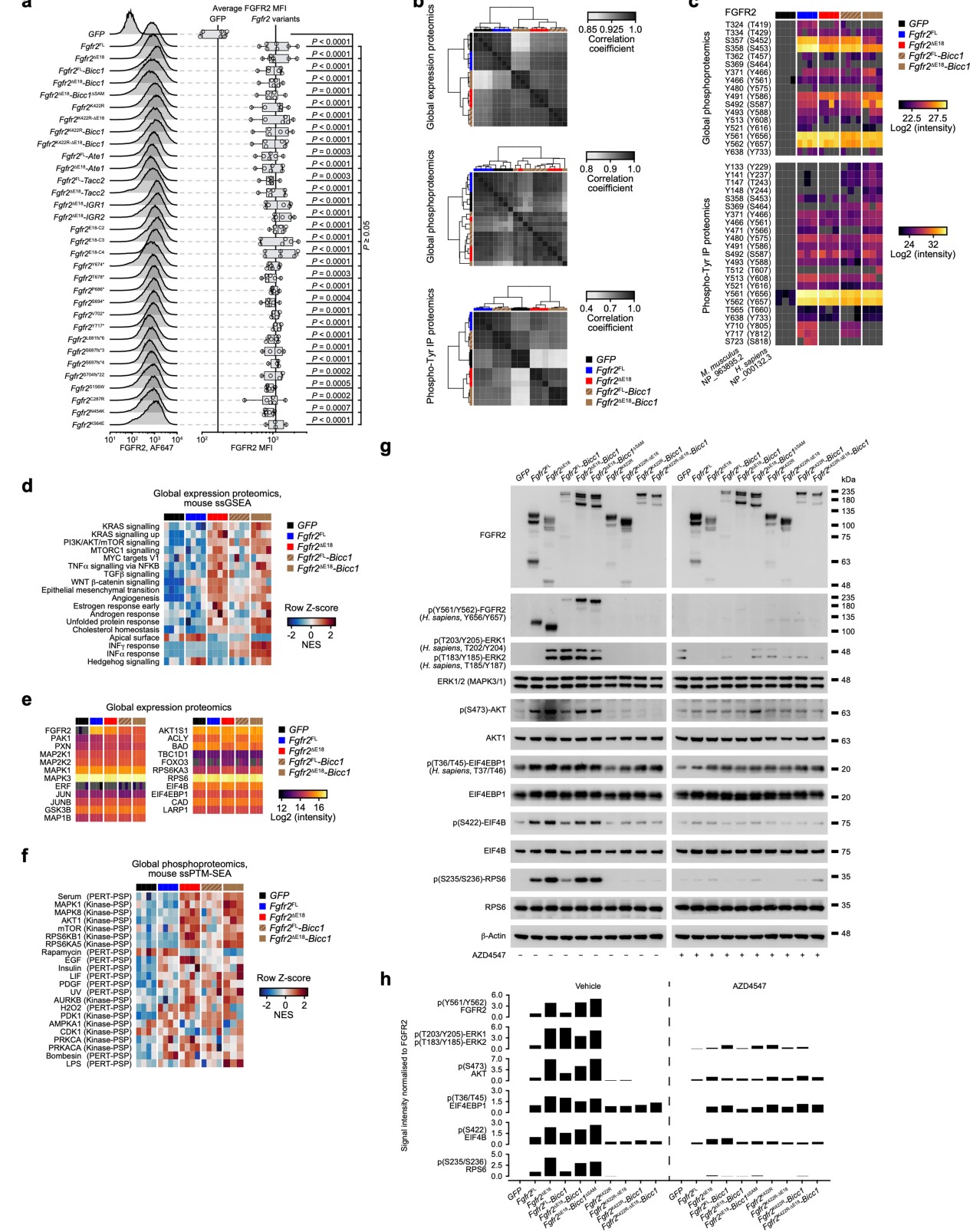

**Extended Data Fig. 5** | See next page for caption.

**Extended Data Fig. 5 | (Phospho)-proteomic analyses of NMuMG cells expressing *Fgfr2* variants. a**, Fluorescence-activated cell sorting (FACS) to analyse FGFR2 mean fluorescence intensity (MFI) in NMuMG cells expressing *GFP* or indicated *Fgfr2* variants. *Fgfr2^FL^*, full-length (FL) *Fgfr2*. *Ate1*, *Bicc1*, and *Tacc2* correspond to the top-recurrent *ATE1*, *BICC1*, and *TACC2* fusion partner genes in Extended Data Fig. 2h. *Bicc1^ΔSAM^* encodes BICC1 lacking its SAM oligomerisation domain. *Fgfr2^K422R^* variants encode tyrosine kinase domain (KD)-dead FGFR2 variants. Truncated or alternative C-termini encoded by *IGR1/IGR2*, E18-C2/C3/C4, *Fgfr2^Y674*^*, *Fgfr2^T678*^*, *Fgfr2^P686*^*, *Fgfr2^S694*^*, *Fgfr2^V702*^*, *Fgfr2^Y717*^*, *Fgfr2^L681fs*6^*, *Fgfr2^S687fs*3^*, *Fgfr2^S697fs*4^*, and *Fgfr2^S704fs*22^* are displayed in Extended Data Fig. 2k. *Fgfr2^S156W^*, *Fgfr2^C287R^*, *Fgfr2^N454K^*, and *Fgfr2^K564E^* correspond to the human *FGFR2^S252W^*, *FGFR2^C382R^*, *FGFR2^N549K^*, and *FGFR2^K659E^* missense hotspot mutations in Extended Data Fig. 2a. Validation of overexpression of *Fgfr2* variants using RT-qPCR is in Supplementary Table 4. Data are represented as median (centre line) ± IQR (25th to 75th percentile, box) and ± full range (minimum to maximum, whiskers) of *GFP*, $n = 6$; *Fgfr2^FL^*, *Fgfr2^ΔE18^*, $n = 7$; *Fgfr2^V702*^*, *Fgfr2^Y717*^*, *Fgfr2^S687fs*3^*, *Fgfr2^S697fs*4^*, *Fgfr2^S704fs*22^*, *Fgfr2^K564E^*, $n = 4$; other *Fgfr2* variants, $n = 6$ independent replica. *P* values were calculated with one-tailed one-way ANOVA and false discovery rate (FDR) multiple-testing correction using the two-stage step-up method from Benjamini, Krieger, and Yekutieli. For FACS gating strategy, see Supplementary Fig. 2a. **b**, Mass spectrometry-based proteomic data showing correlation of NMuMG cells expressing *GFP* or indicated *Fgfr2* variants for global protein expression, global phosphoproteomic analysis after enrichment with IMAC, and phospho-Tyr immunoprecipitation (IP)-enriched samples. Pearson's *R* correlation coefficients are depicted and heatmaps were clustered unsupervised. **c**, Heatmaps visualizing FGFR2 phosphosites identified in (**b**). **d**, Single-sample gene set enrichment analysis (ssGSEA) based on hallmark gene sets from MSigDB[79] and the global protein expression dataset. Significant single-sample normalized enrichment scores (NES) were calculated using GSEA standard settings[77,78]. NES are visualised as colour-coded row *Z*-scores and depicted terms are based on *Fgfr2^ΔE18^* versus *Fgfr2^FL^* two-group comparisons using two-tailed unpaired Student's *t*-tests. Significant terms are shown ($P < 0.05$). **e**, Relative candidate protein expression levels corresponding to MAPK, AKT, and mTOR substrates displayed in Fig. 2b and based on the global protein expression. **f**, Single-sample phosphosite signature enrichment analysis (ssPTM-SEA) based on murine kinase/pathway definitions of PTMsigDB[82] and the global phosphoproteomic dataset. Significant single-sample NES were calculated using gene permutation ($n = 1,000$) and one-tailed permutation testing with FDR multiple-testing correction using the Benjamini-Hochberg method by applying PTM-SEA standard settings[82]. NES are visualised as colour-coded row *Z*-scores and depicted terms are based on *Fgfr2^ΔE18^* versus *GFP*, *Fgfr2^ΔE18^* versus *Fgfr2^FL^*, and/or *Fgfr2^FL^* versus *GFP* two-group comparisons using two-tailed unpaired Student's *t*-tests. Terms significant for either of the three two-group comparisons are shown ($P < 0.05$). **g**, Western blots showing expression and phosphorylation of indicated proteins in NMuMG cells expressing *GFP* or indicated *Fgfr2* variants and treated for 3 h with vehicle or 100 nM AZD4547. β-Actin was run on separate gels as sample processing control, and each blot was stained with Ponceau S to ensure equal loading of total protein. Blots stained with the same antibody were developed and recorded in parallel and subjected to equal post-imaging processing. For gel source data, see Supplementary Fig. 1a–g. **h**, Quantifications of relative phosphoprotein band intensities in (**g**) normalized to β-actin, corresponding total protein, and FGFR2 band intensities. Data in **g**, **h** represent 1 replica of 2 independent experiments.

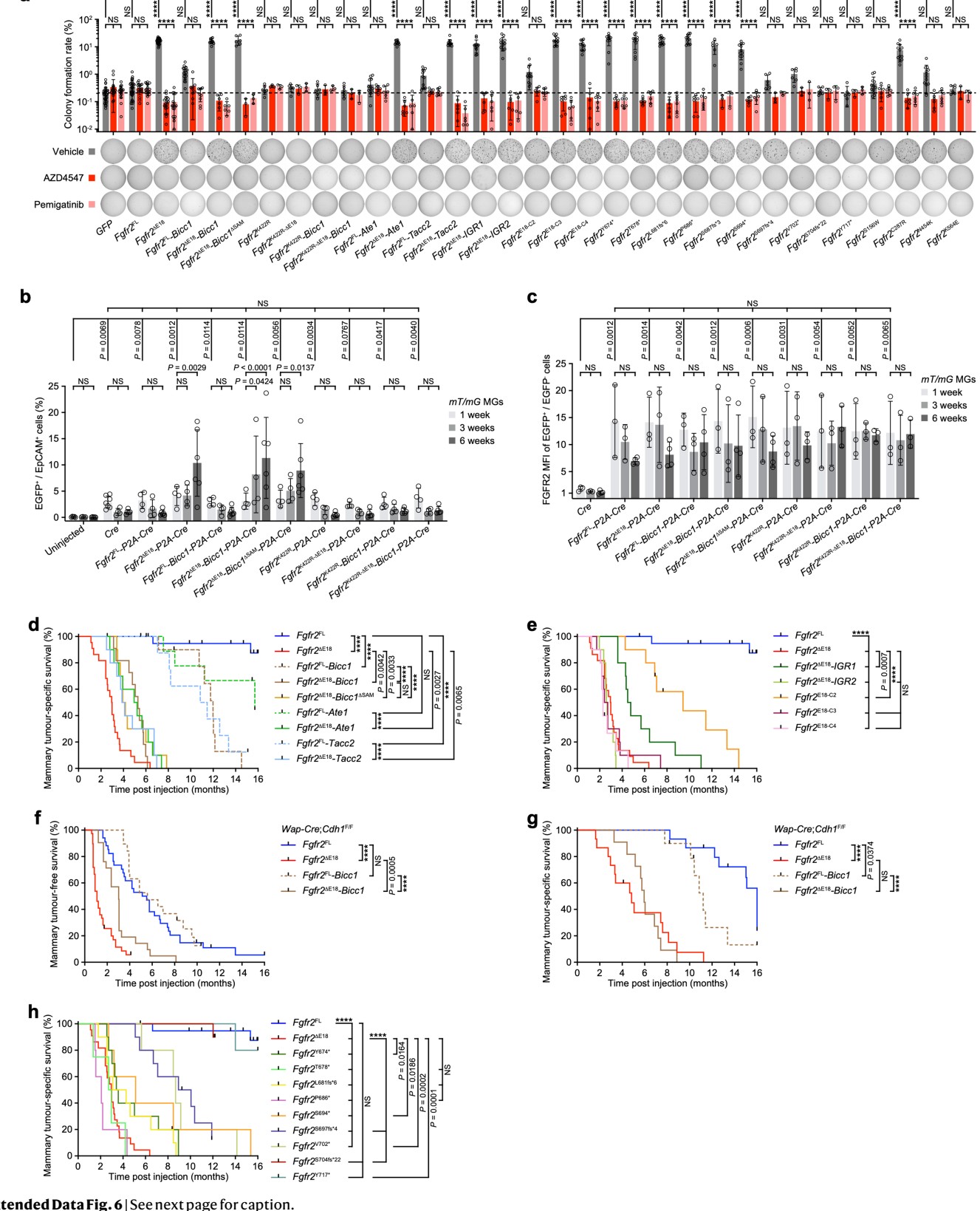

**Extended Data Fig. 6** | See next page for caption.

**Extended Data Fig. 6 | In vitro and in vivo oncogenic capacities of *Fgfr2* variants. a**, Representative images of 12-well plate wells and quantification of 3D soft agar colony formation assay using NMuMG cells expressing *GFP* or indicated *Fgfr2* variants and treated with vehicle, 100 nM AZD4547, or 100 nM pemigatinib for 15 days. Data are represented as mean ± standard deviation (s.d.) of *GFP*, *Fgfr2^FL^*, *Fgfr2^ΔE18^*, vehicle, $n = 33$; AZD4547, pemigatinib, $n = 18$ independent replica from 4 individual experiments. *Fgfr2^FL^-Bicc1*, vehicle, $n = 18$; AZD4547, pemigatinib, $n = 9$ independent replica from 3 individual experiments. *Fgfr2^ΔE18^-Bicc1*, *Fgfr2^FL^-Ate1*, *Fgfr2^ΔE18^-Ate1*, *Fgfr2^FL^-Tacc2*, *Fgfr2^ΔE18^-Tacc2*, *Fgfr2^ΔE18^-IGR1*, *Fgfr2^ΔE18^-IGR2*, *Fgfr2^E18-C2^*, *Fgfr2^E18-C3^*, *Fgfr2^E18-C4^*, *Fgfr2^Y674*^*, *Fgfr2^T678*^*, *Fgfr2^L681fs*6^*, *Fgfr2^P686*^*, *Fgfr2^S694*^*, *Fgfr2^S156W^*, *Fgfr2^C287R^*, *Fgfr2^N454K^*, vehicle, $n = 12$; AZD4547, pemigatinib, $n = 6$ independent replica from 2 individual experiments. *Fgfr2^ΔE18^-Bicc1^ΔSAM^*, *Fgfr2^K422R^*, *Fgfr2^K422R-ΔE18^*, *Fgfr2^K422R^-Bicc1*, *Fgfr2^K422R-ΔE18^-Bicc1*, *Fgfr2^S687fs*3^*, *Fgfr2^S697fs*4^*, *Fgfr2^V702*^*, *Fgfr2^S704fs*22^*, *Fgfr2^Y717*^*, *Fgfr2^K564E^*, $n = 6$; AZD4547, pemigatinib, $n = 3$ independent replica from 1 experiment. **b, c**, FACS to quantify traced EGFP⁺ EpCAM⁺ epithelial cells (**b**) and their FGFR2 MFI (**c**). *Rosa26-mT/mG* female reporter mice were intraductally injected with lentiviruses encoding *Cre* or indicated *Fgfr2-P2A-Cre* variants resulting in Cre-mediated *mT/mG* allele switching, thus cell membrane-localized tdTomato (mT) expression was replaced by membrane-localized EGFP (mG) expression. Mammary glands (MGs) were subjected to FACS analysis at indicated timepoints post injection. Data are represented as mean ± s.d. and each data point represents a MG pool of an individual mouse. Analyses were done in batches of 1–2 mice of each *Fgfr2* variant and one timepoint. In (**b**), 1 week, uninjected MGs, $n = 5$; *Cre*, $n = 7$; other *Fgfr2* variants, $n = 4$; 3 weeks, all groups, $n = 4$; 6 weeks, uninjected MGs, *Cre*, *Fgfr2^K422R^-P2A-Cre*, *Fgfr2^K422R-ΔE18^-P2A-Cre*, *Fgfr2^K422R^-Bicc1-P2A-Cre*, *Fgfr2^K422R-ΔE18^-Bicc1-P2A-Cre*, $n = 5$; other *Fgfr2* variants, $n = 6$ mice. In (**c**), 1 week and 3 weeks, all groups, $n = 3$; 6 weeks, *Cre*, *Fgfr2^FL^-P2A-Cre*, *Fgfr2^ΔE18^-P2A-Cre*, *Fgfr2^FL^-Bicc1-P2A-Cre*, *Fgfr2^ΔE18^-Bicc1-P2A-Cre*, *Fgfr2^ΔE18^-Bicc1^ΔSAM^-P2A-Cre*, $n = 4$; other *Fgfr2* variants, $n = 3$ mice. For FACS gating strategy, see Supplementary Fig. 2b. **d, e**, Kaplan-Meier curves showing mammary tumour-specific survival of female wild-type (WT) mice intraductally injected with lentiviruses encoding indicated *Fgfr2* variants. *Fgfr2^FL^*, $n = 20$; *Fgfr2^ΔE18^*, $n = 22$; *Fgfr2^FL^-Bicc1*, *Fgfr2^ΔE18^-Bicc1^ΔSAM^*, *Fgfr2^FL^-Ate1*, *Fgfr2^ΔE18^-Ate1*, *Fgfr2^FL^-Tacc2*, *Fgfr2^ΔE18^-Tacc2*, *Fgfr2^ΔE18^-IGR1*, *Fgfr2^ΔE18^-IGR2*, *Fgfr2^E18-C2^*, *Fgfr2^E18-C3^*, *Fgfr2^E18-C4^*, $n = 10$; *Fgfr2^ΔE18^-Bicc1*, $n = 11$ mice. *Fgfr2^FL^* and *Fgfr2^ΔE18^* curves in **a** are duplicated in **d**, **h**. **f, g**, Kaplan-Meier curves showing mammary tumour-free (**c**) and -specific (**d**) survival of female *Wap-Cre;Cdh1^F/F^* mice intraductally injected with lentiviruses encoding indicated *Fgfr2* variants. *Fgfr2^FL^*, $n = 34$ of 15; *Fgfr2^ΔE18^*, $n = 39$ of 15; *Fgfr2^FL^-Bicc1*, $n = 19$ of 10; *Fgfr2^ΔE18^-Bicc1*, $n = 21$ injected MGs of 11 mice. **h**, Kaplan-Meier curves showing mammary tumour-specific survival of female wild-type (WT) mice intraductally injected with lentiviruses encoding indicated *Fgfr2* variants. *Fgfr2^Y674*^*, *Fgfr2^L681fs*6^*, *Fgfr2^S697fs*4^*, *Fgfr2^S704fs*22^*, $n = 10$; *Fgfr2^T678*^*, $n = 4$; *Fgfr2^P686*^*, *Fgfr2^S694*^*, *Fgfr2^V702*^*, *Fgfr2^Y717*^*, $n = 5$ mice. *P* values were calculated with one-tailed two-way ANOVA and FDR multiple-testing corrections using the two-stage step-up method from Benjamini, Krieger, and Yekutieli (**a, c**), one-tailed Kruskal-Wallis tests and Dunn's multiple-testing corrections (**b**), or log rank (Mantel-Cox) tests (**b–h**). ****$P < 0.0001$; NS, not significant ($P \geq 0.05$).

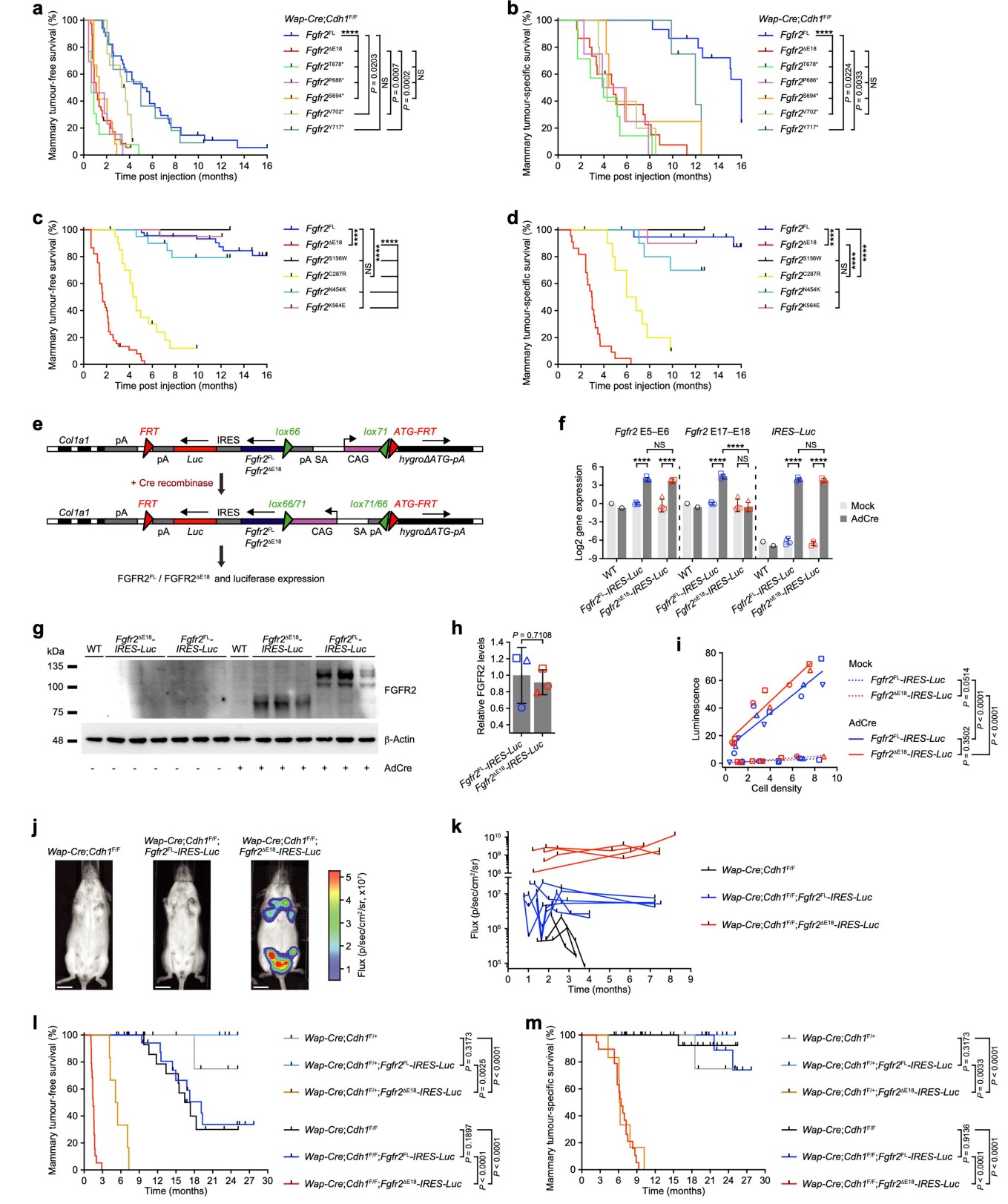

**Extended Data Fig. 7 |** See next page for caption.

**Extended Data Fig. 7 | In vivo oncogenic capacities of *Fgfr2* variants in somatic models and GEMMs. a**, **b**, Kaplan-Meier curves showing mammary tumour-free (**a**) and -specific (**b**) survival of female *Wap-Cre;Cdh1^{F/F}* mice intraductally injected with lentiviruses encoding indicated *Fgfr2* variants. *Fgfr2^{T678*}*, $n = 13$ of 7; *Fgfr2^{P686*}*, $n = 13$ of 4; *Fgfr2^{S694*}*, $n = 12$ of 4; *Fgfr2^{V702*}*, $n = 12$ of 5; *Fgfr2^{Y717*}*, $n = 11$ injected MGs of 4 mice. *Fgfr2^{FL}* and *Fgfr2^{ΔE18}* curves in **a**, **b** are duplicates from Extended Data Fig. 6f, g. **c**, **d**, Kaplan-Meier curves showing mammary tumour-free (**c**) and -specific (**d**) survival of female WT mice intraductally injected with lentiviruses encoding indicated *Fgfr2* variants. *Fgfr2^{S156W}*, *Fgfr2^{C287R}*, *Fgfr2^{N454K}*, *Fgfr2^{K564E}*, $n = 20$ injected MGs of 10 mice. *Fgfr2^{FL}* and *Fgfr2^{ΔE18}* curves in **c**, **d** are duplicates from Fig. 2c and Extended Data Fig. 6d. **e**, Schematic representation of the engineered *Fgfr2^{FL}* and *Fgfr2^{ΔE18}* alleles. *Frt-invCAG-Fgfr2^{FL}-IRES-Luc* and *Frt-invCAG-Fgfr2^{ΔE18}-IRES-Luc* were inserted into the *Col1a1* locus using the genetically engineered mouse model – embryonic stem cell (GEMM-ESC) methodology[54]. Cre activity inverts the *CAG* promoter resulting in coherent FGFR2 and luciferase (Luc) expression. IRES, internal ribosome entry site. **f**, RT-qPCR quantification of *Fgfr2* and *Luc* expression in mouse mammary epithelial cells (MMECs) isolated from pooled MGs of 10-week-old WT control, *Fgfr2^{FL}-IRES-Luc*, and *Fgfr2^{ΔE18}-IRES-Luc* female mice and mock-treated or treated with adenoviral Ad5CMVCre (AdCre) to switch *Fgfr2* alleles in vitro. Data are represented as mean ± s.d. of WT, $n = 1$; *Fgfr2^{FL}-IRES-Luc*, *Fgfr2^{ΔE18}-IRES-Luc*, $n = 4$ MMEC cultures each from MG pools of individual mice. **g**, **h**, Western blot showing FGFR2 expression of mock- or AdCre-treated MMEC cultures (**g**) and quantification of relative FGFR2 intensities normalized to β-actin (**h**). β-Actin was run on a separate gel as sample processing control, and membranes were stained with Ponceau S to ensure equal loading of total protein. For gel source data, see Supplementary Fig. 1h, i. In **h**, data are represented as mean ± s.d. of WT, $n = 1$; *Fgfr2^{FL}-IRES-Luc*, *Fgfr2^{ΔE18}-IRES-Luc*, $n = 3$ MMEC cultures each from MG pools of individial mice. **i**, Luciferase activity measured using luciferin and bioluminescence imaging on mock- or AdCre-treated MMEC cultures. Data are represented as simple linear regressions across *Fgfr2^{FL}-IRES-Luc*, $n = 4$; *Fgfr2^{ΔE18}-IRES-Luc*, $n = 3$ MMEC cultures (each from MG pools of individual mice) at indicated cell densities. **j**, Representative in vivo bioluminescence images showing luciferase activity following luciferin administration measured as photon flux in 10-week-old *Wap-Cre;Cdh1^{F/F}*, *Wap-Cre;Cdh1^{F/F};Fgfr2^{FL}-IRES-Luc*, and *Wap-Cre;Cdh1^{F/F};Fgfr2^{ΔE18}-IRES-Luc* female mice. Scale bars, 1 cm. **k**, Quantification of luciferase activity using recurrent bioluminescence imaging in indicated GEMMs. *Wap-Cre;Cdh1^{F/F}* female mice show background luminescence. *Wap-Cre;Cdh1^{F/F}*, $n = 3$; *Wap-Cre;Cdh1^{F/F};Fgfr2^{FL}-IRES-Luc*, $n = 6$; *Wap-Cre;Cdh1^{F/F};Fgfr2^{ΔE18}-IRES-Luc*, $n = 4$ mice. **l**, **m**, Kaplan-Meier curves showing mammary tumour-free (**l**) and -specific (**m**) survival of indicated GEMMs. *Wap-Cre;Cdh1^{F/+}*, $n = 12$; *Wap-Cre;Cdh1^{F/+};Fgfr2^{FL}-IRES-Luc*, $n = 5$; *Wap-Cre;Cdh1^{F/+};Fgfr2^{ΔE18}-IRES-Luc*, $n = 6$; *Wap-Cre;Cdh1^{F/F}*, $n = 16$; *Wap-Cre;Cdh1^{F/F};Fgfr2^{FL}-IRES-Luc*, *Wap-Cre; Cdh1^{F/F};Fgfr2^{ΔE18}-IRES-Luc*, $n = 19$ mice. *P* values were calculated with log rank (Mantel-Cox) tests (**a**–**d**, **l**, **m**), one-tailed two-way ANOVA and FDR multiple-testing corrections using the two-stage step-up method from Benjamini, Krieger, and Yekutieli (**f**), a two-tailed unpaired Student's *t*-test (**h**), or one-way analysis of covariance (ANCOVA) to compare linear regression slopes (**i**). ****$P < 0.0001$.

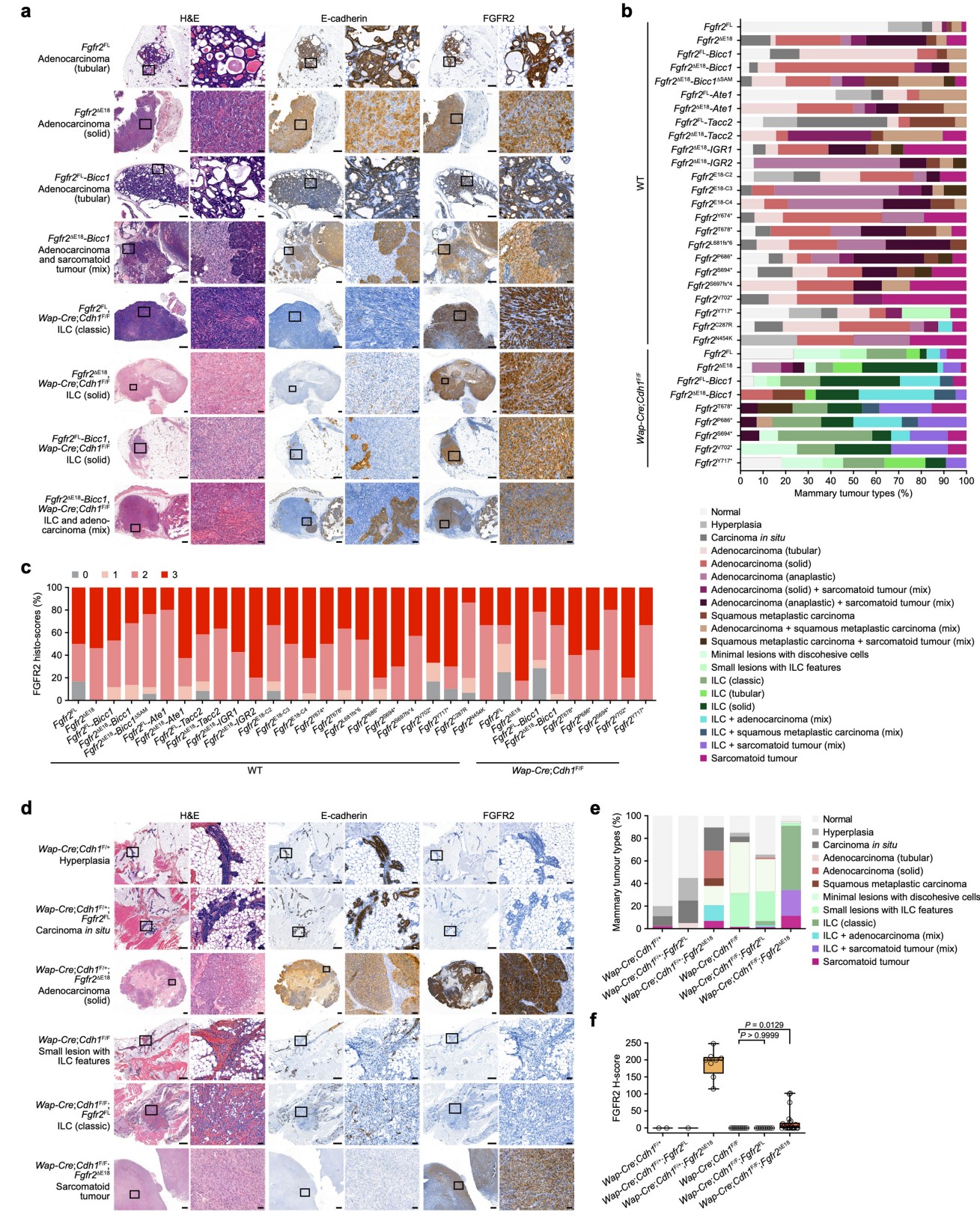

**Extended Data Fig. 8** | See next page for caption.

**Extended Data Fig. 8 | Mammary tumour types observed in *Fgfr2* mouse models. a**, Representative hematoxylin and eosin (H&E) histochemistry and FGFR2 and E-cadherin immunohistochemistry (IHC) stains on mammary tissue sections from indicated *Fgfr2* somatic mouse models. Per MG one tissue section was stained and quantified for each of the indicated stains acquired in multiple independent randomized batches across all *Fgfr2* variants. Numbers of stained and quantified MGs are in (**b**, **c**). ILC, invasive lobular carcinoma. **b**, Mammary tumour type classifications of *Fgfr2* somatic mouse models based on H&Es and E-cadherin IHC stains. WT, *Fgfr2$^{FL}$*, $n$ = 46 of 20; *Fgfr2$^{\Delta E18}$*, $n$ = 45 of 22; *Fgfr2$^{FL}$-Bicc1*, $n$ = 23 of 10; *Fgfr2$^{\Delta E18}$-Bicc1*, $n$ = 26 of 11; *Fgfr2$^{\Delta E18}$-Bicc1$^{\Delta SAM}$*, *Fgfr2$^{\Delta E18}$-Ate1*, *Fgfr2$^{FL}$-Tacc2*, *Fgfr2$^{E18-C3}$*, $n$ = 20 of 10; *Fgfr2$^{FL}$-Ate1*, *Fgfr2$^{\Delta E18}$-Tacc2*, *Fgfr2$^{E18-C4}$*, $n$ = 19 of 10; *Fgfr2$^{\Delta E18}$-IGR1*, $n$ = 18 of 9; *Fgfr2$^{\Delta E18}$-IGR2*, *Fgfr2$^{E18-C2}$*, $n$ = 17 of 10; *Fgfr2$^{Y674*}$*, *Fgfr2$^{C287R}$*, $n$ = 16 of 8; *Fgfr2$^{T678*}$*, $n$ = 15 of 4; *Fgfr2$^{L681lfs*6}$*, $n$ = 14 of 7; *Fgfr2$^{P686*}$*, $n$ = 16 of 5; *Fgfr2$^{S694*}$*, $n$ = 13 of 5; *Fgfr2$^{S697fs*4}$*, $n$ = 8 of 4; *Fgfr2$^{V702*}$*, $n$ = 8 of 3; *Fgfr2$^{Y717*}$*, $n$ = 14 of 5; *Fgfr2$^{N454K}$*, $n$ = 4 injected MGs of 2 mice. *Wap-Cre;Cdh1$^{F/F}$*, *Fgfr2$^{FL}$*, $n$ = 34 of 15; *Fgfr2$^{\Delta E18}$*, $n$ = 39 of 15; *Fgfr2$^{FL}$-Bicc1*, $n$ = 17 of 9; *Fgfr2$^{\Delta E18}$-Bicc1*, $n$ = 21 of 11; *Fgfr2$^{T678*}$*, $n$ = 13 of 7; *Fgfr2$^{P686*}$*, $n$ = 14 of 4; *Fgfr2$^{S694*}$*, $n$ = 12 of 4; *Fgfr2$^{V702*}$*, $n$ = 12 of 5; *Fgfr2$^{Y717*}$*, $n$ = 11 injected MGs of 4 mice. **c**, Histo-scoring of FGFR2 IHC stains on mammary tumours from *Fgfr2* somatic mouse models. WT, *Fgfr2$^{FL}$*, $n$ = 6 of 5; *Fgfr2$^{\Delta E18}$*, $n$ = 39 of 22; *Fgfr2$^{FL}$-Bicc1*, $n$ = 17 of 10; *Fgfr2$^{\Delta E18}$-Bicc1*, $n$ = 22 of 11; *Fgfr2$^{\Delta E18}$-Bicc1$^{\Delta SAM}$*, $n$ = 17 of 9; *Fgfr2$^{FL}$-Ate1*, $n$ = 5 of 5; *Fgfr2$^{\Delta E18}$-Ate1*, *Fgfr2$^{E18-C4}$*, $n$ = 16 of 9; *Fgfr2$^{FL}$-Tacc2*, $n$ = 12 of 8; *Fgfr2$^{\Delta E18}$-Tacc2*, $n$ = 11 of 8; *Fgfr2$^{\Delta E18}$-IGR1*, *Fgfr2$^{E18-C3}$*, $n$ = 14 of 9; *Fgfr2$^{\Delta E18}$-IGR2*, $n$ = 15 of 10; *Fgfr2$^{E18-C2}$*, $n$ = 12

of 9; *Fgfr2$^{Y674*}$*, $n$ = 14 of 8; *Fgfr2$^{T678*}$*, $n$ = 11 of 4; *Fgfr2$^{L681lfs*6}$*, $n$ = 13 of 8; *Fgfr2$^{P686*}$*, *Fgfr2$^{S694*}$*, $n$ = 10 of 5; *Fgfr2$^{S697fs*4}$*, $n$ = 7 of 4; *Fgfr2$^{V702*}$*, $n$ = 6 of 3; *Fgfr2$^{Y717*}$*, $n$ = 10 of 4; *Fgfr2$^{C287R}$*, $n$ = 15 of 8; *Fgfr2$^{N454K}$*, $n$ = 3 tumours of 2 mice. *Wap-Cre;Cdh1$^{F/F}$*, *Fgfr2$^{FL}$*, $n$ = 12 of 8; *Fgfr2$^{\Delta E18}$*, $n$ = 23 of 9; *Fgfr2$^{FL}$-Bicc1*, $n$ = 14 of 7; *Fgfr2$^{\Delta E18}$-Bicc1*, $n$ = 18 of 10; *Fgfr2$^{T678*}$*, *Fgfr2$^{V702*}$*, $n$ = 10 of 5; *Fgfr2$^{P686*}$*, $n$ = 9 of 4; *Fgfr2$^{S694*}$*, $n$ = 10 of 4; *Fgfr2$^{Y717*}$*, $n$ = 6 tumours of 3 mice. **d**, Representative H&E histochemistry and E-cadherin and FGFR2 IHC stains on mammary tissue sections from indicated GEMMs. Per MG one tissue section was stained and quantified for each of the indicated stains acquired in two independent randomized batches across all genotypes. Numbers of stained and quantified MGs are in (**e**, **f**). **e**, Mammary tumour type classifications of GEMMs based on H&Es and E-cadherin IHC stains. *Wap-Cre;Cdh1$^{F/+}$*, $n$ = 45 of 12; *Wap-Cre;Cdh1$^{F/+}$;Fgfr2$^{FL}$*, $n$ = 20 of 5; *Wap-Cre; Cdh1$^{F/+}$;Fgfr2$^{\Delta E18}$*, $n$ = 29 of 6; *Wap-Cre;Cdh1$^{F/F}$*, $n$ = 60 of 16; *Wap-Cre;Cdh1$^{F/F}$; Fgfr2$^{FL}$*, $n$ = 73 of 19; *Wap-Cre;Cdh1$^{F/F}$;Fgfr2$^{\Delta E18}$*, $n$ = 79 MGs of 19 mice. **f**, Histo (*H*)-score quantifications of FGFR2 IHC stains on mammary tumours from GEMMs. *Wap-Cre;Cdh1$^{F/+}$*, $n$ = 2 of 2; *Wap-Cre;Cdh1$^{F/+}$;Fgfr2$^{FL}$*, $n$ = 1 of 1; *Wap-Cre;Cdh1$^{F/+}$;F gfr2$^{\Delta E18}$*, $n$ = 8 of 5; *Wap-Cre;Cdh1$^{F/F}$*, $n$ = 9 of 8; *Wap-Cre;Cdh1$^{F/F}$;Fgfr2$^{FL}$*, $n$ = 8 of 6; *Wap-Cre;Cdh1$^{F/F}$;Fgfr2$^{\Delta E18}$*, $n$ = 24 tumours of 19 mice. Data are represented as median (centre line) ± IQR (25$^{th}$ to 75$^{th}$ percentile, box) and ± full range (minimum to maximum, whiskers) and *P* values were calculated with one-tailed Kruskal-Wallis tests and Dunn's multiple-testing corrections. Scale bars, overview, 500 μm; inset, 50 μm.

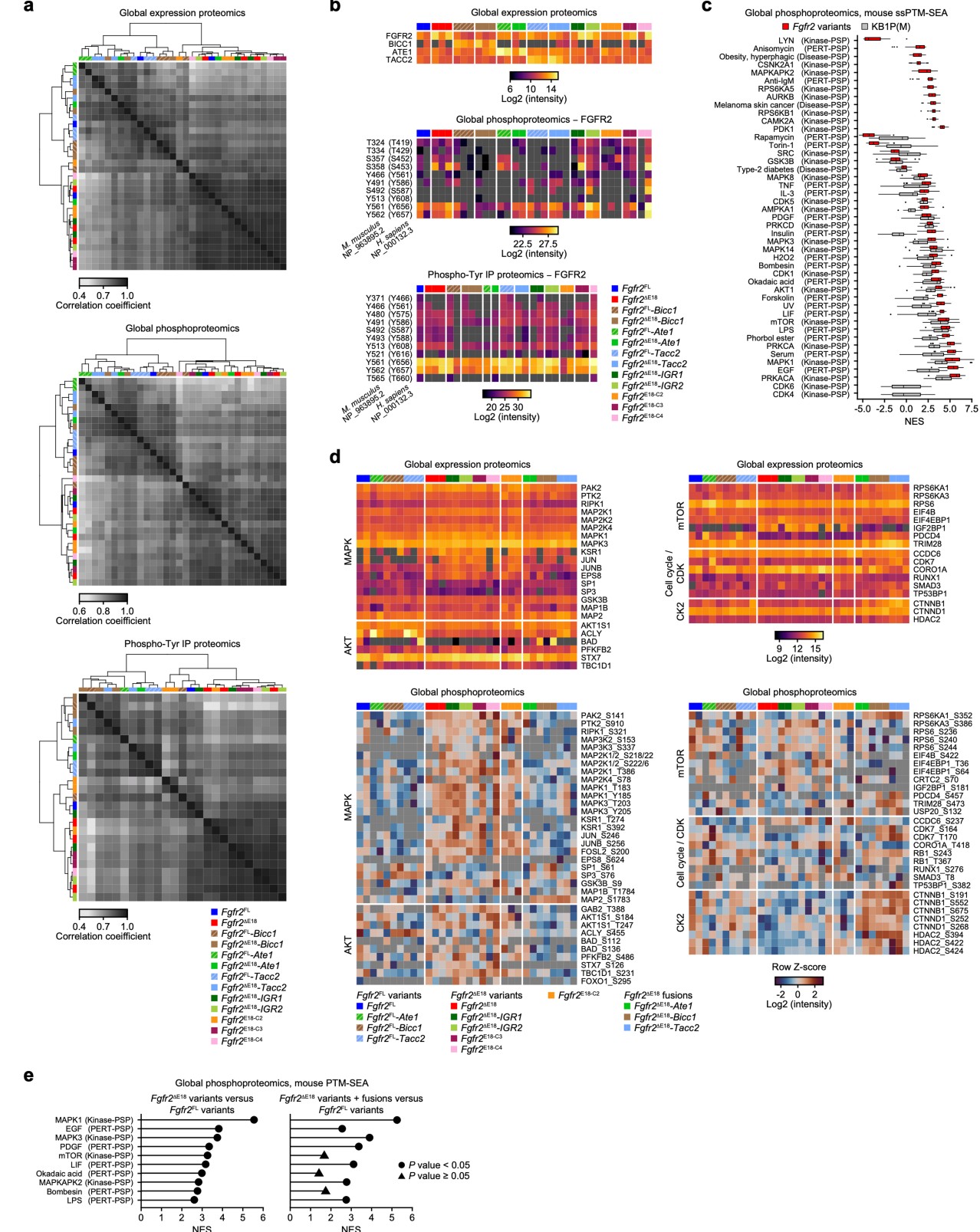

**Extended Data Fig. 9** | See next page for caption.

**Extended Data Fig. 9 | (Phospho)-proteomic analyses of tumours from _Fgfr2_ somatic mouse models. a**, Mass spectrometry-based proteomic data showing correlation of indicated _Fgfr2_ somatic mouse models for global protein expression, global phosphoproteomic analysis after enrichment with IMAC, and phospho-Tyr IP-enriched mammary tumours. Pearson's _R_ correlation coefficients are depicted and heatmaps were clustered unsupervised. **b**, Relative protein expression of FGFR2 and its fusion partners BICC1, ATE1, and TACC2 next to FGFR2 phosphosites identified in datasets from **a**. Heatmaps colour-code relative intensities of protein expression and phosphorylation. **c**, ssPTM-SEA based on murine kinase/pathway definitions of PTMsigDB and the global phosphoproteomic dataset in **a** as well as phosphoproteomic data generated from mammary tumours from _K14-Cre_; _Brca_^F/F;_Trp_53^F/F (KB1P) and _KB1P_;_Mdr1a/b_^−/− (KB1PM) GEMMs. Each boxplot represents one ssPTM-SEA term and shows NES of individual _Fgfr2_ variant tumours or KB1P(M) tumours. ssPTM-SEA terms enriched in the _Fgfr2_ variant and/or the KB1P(M) tumour cohorts are shown. Significant single-sample NES were calculated using gene permutation (_n_ = 1,000) and permutation-derived _P_ values by applying PTM-SEA standard settings[82]. No further statistical selections were applied. Boxplots are represented as median (centre line) ± IQR (25th to 75th percentile, box) and IQR ± 1.5 x IQR (whiskers). _Fgfr2_ variants, _n_ = 32; KB1P, _n_ = 14; KB1PM, _n_ = 10 tumours. **d**, Relative candidate protein expression (top panels) and phosphorylation (bottom panels) levels of MAPK, AKT, mTOR, cell cycle / CDK, and CK2 substrates identified in **a**. For the phosphoproteomic analysis, samples were grouped into _Fgfr2_^FL variants, _Fgfr2_^ΔE18 variants, and _Fgfr2_^ΔE18 fusion variants and compared pairwise using the robust kinase activity inference (RoKAI) tool at default settings[83] including two-tailed hypothesis testing on _Z_-scores and FDR multiple-testing correction using the Benjamini-Hochberg method. Group comparison fold change (FC) values of −1.5 ≥ FC ≥ 1.5 and _P_ < 0.05 were considered. The RoKAI output was used to manually curate phosphosites of interest, and phosphosites were manually grouped into indicated signalling pathways guided by RoKAI. The heatmaps depict relative expression intensities (top panels) and Z-scores of phosphosite intensities calculated per row from log$_2$-transformed intensity values (bottom panels). **e**, PTM-SEA based on murine kinase/pathway definitions of PTMsigDB and performed with global phosphoproteomic data and limma-based two-group comparisons of _Fgfr2_^ΔE18 variants versus _Fgfr2_^FL variants groups (left panel) and _Fgfr2_^ΔE18 variants including fusions versus _Fgfr2_^FL variants groups (right panel). Significant NES were calculated by using gene permutation (_n_ = 1,000) and one-tailed permutation testing without multiple-testing correction by applying PTM-SEA standard settings[82]. Lollipops show NES of terms significantly enriched in either of the two comparisons (_P_ < 0.05).

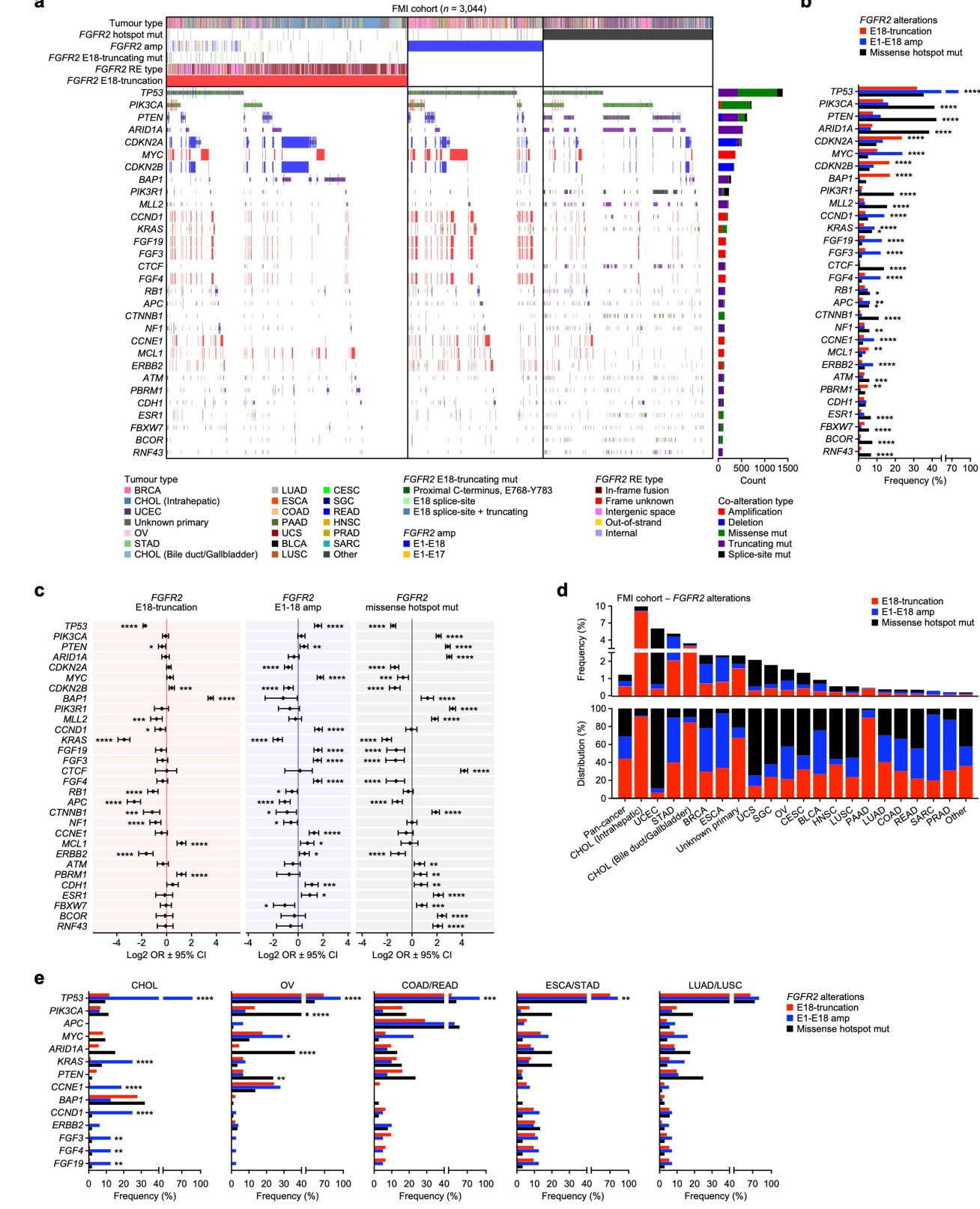

**Extended Data Fig. 10** | See next page for caption.

**Extended Data Fig. 10 | Top driver genes co-occurring in samples with *FGFR2* alterations. a**, 3,044 samples classified as either *FGFR2*-E18-truncated ($n = 1,344$, 44.2% of total, 0.54% incidence), amplified ($n = 757$, 24.8% of total, 0.30% incidence), or missense hotspot mutant ($n = 943$, 31.0% of total, 0.38% incidence) and top-30 co-enriched tumour driver alterations found in the FMI pan-cancer cohort ($n = 249,570$). **b**, Enrichments of top-30 tumour driver co-alterations in the indicated *FGFR2* alteration categories in the FMI pan-cancer cohort. **c**, Odds ratios (OR) of top-30 tumour driver co-alterations in the indicated *FGFR2* alteration categories (E18-truncation, $n = 1,344$; E1-E18 amp, $n = 757$; missense hotspot mut, $n = 943$) versus *FGFR2* WT samples ($n = 224,711$) of the FMI pan-cancer cohort. Data are represented as $\log_2$-transformed OR ± 95% confidence interval (CI). Co-occurrence, OR > 1; mutual exclusivity, OR < 1. **d**, Frequencies (top panel) and distributions (bottom panel) per tumour type of the indicated *FGFR2* alteration categories in the FMI pan-cancer cohort. **e**, Enrichment of top tumour driver co-alterations in the indicated *FGFR2* alteration categories in the FMI-CHOL, OV, COAD/READ, ESCA/STAD, and LUAD/LUSC cohorts. *P* values were calculated with one-tailed proportion *z*-tests (**b**, **e**) or two-tailed Fisher's exact tests (**c**) and FDR multiple-testing corrections using the Benjamini-Hochberg method (**b**, **c**, **e**). \*$P < 0.05$, \*\*$P < 0.01$, \*\*\*$P < 0.001$, \*\*\*\*$P < 0.0001$. Sample sizes and statistical details for **b**, **c**, **e** are in Supplementary Table 2.

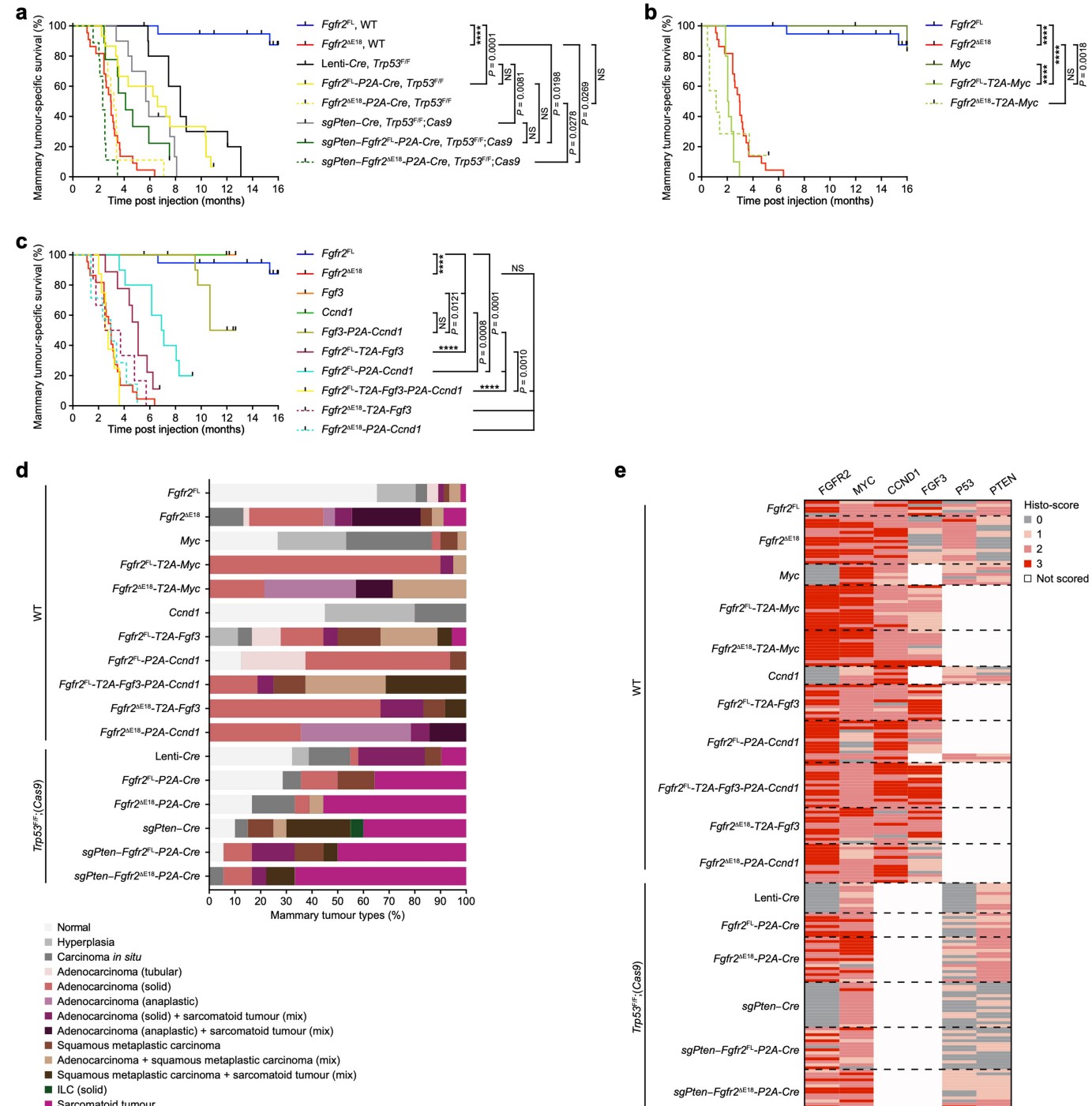

**Extended Data Fig. 11 | Somatic modelling of *Fgfr2* variants and co-occurring driver alterations. a-c**, Kaplan-Meier curves showing mammary tumour-specific survival of *Trp53^F/F* and *Trp53^F/F*;*Rosa26-Cas9* (**a**) and WT (**b**, **c**) female mice intraductally injected with lentiviruses encoding indicated variants. *Trp53^F/F*;(*Rosa26-Cas9*), Lenti-*Cre*, *sgPten-Cre*, *n* = 10; *Fgfr2^FL-P2A-Cre*, *n* = 15; *Fgfr2^ΔE18-P2A-Cre*, *sgPten-Fgfr2^FL-P2A-Cre*, *sgPten-Fgfr2^ΔE18-P2A-Cre*, *n* = 9 mice. WT, *Myc*, *Fgfr2^FL-T2A-Fgf3*, *n* = 9; *Fgfr2^FL-T2A-Myc*, *Fgf3*, *Ccnd1*, *Fgf3-P2A-Ccnd1*, *Fgfr2^FL-P2A-Ccnd1*, *n* = 10; *Fgfr2^ΔE18-T2A-Myc*, *Fgfr2^ΔE18-P2A-Ccnd1*, *n* = 7; *Fgfr2^FL-T2A-Fgf3-P2A-Ccnd1*, *n* = 8; *Fgfr2^ΔE18-T2A-Fgf3*, *n* = 6 mice. *Fgfr2^FL* and *Fgfr2^ΔE18* curves are duplicates from Extended Data Fig. 6d. *P* values were calculated with log rank (Mantel-Cox) tests. \*\*\*\**P* < 0.0001. **d**, Mammary tumour type classifications of somatic mouse models based on H&Es. WT, *Myc*, *n* = 30 of 9; *Fgfr2^FL-T2A-Myc*, *n* = 20 of 10; *Fgfr2^ΔE18-T2A-Myc*, *Fgfr2^ΔE18-P2A-Ccnd1*, *n* = 14 of 7; *Ccnd1*, *n* = 40 of 10; *Fgfr2^FL-T2A-Fgf3*, *n* = 18 of 9; *Fgfr2^FL-P2A-Ccnd1*, *Fgfr2^FL-T2A-Fgf3-P2A-Ccnd1*, *n* = 16 of 8; *Fgfr2^ΔE18-T2A-Fgf3*,

*n* = 12 injected MGs of 6 mice. *Trp53^F/F*;(*Rosa26-Cas9*), Lenti-*Cre*, *n* = 31 of 10; *Fgfr2^FL-P2A-Cre*, *n* = 14 of 7; *Fgfr2^ΔE18-P2A-Cre*, *sgPten-Fgfr2^FL-P2A-Cre*, *sgPten-Fgfr2^ΔE18-P2A-Cre*, *n* = 18 of 9; *sgPten-Cre*, *n* = 20 injected MGs of 10 mice. WT *Fgfr2^FL* and *Fgfr2^ΔE18* classifications are duplicates from Extended Data Fig. 8b. **e**, Histo-scoring of indicated IHC stains on mammary tumours from somatic mouse models. WT, *Fgfr2^FL*, *n* = 5 of 5; *Fgfr2^ΔE18*, *n* = 16 of 11; *Myc*, *n* = 7 of 3; *Fgfr2^FL-T2A-Myc*, *n* = 15 of 10; *Fgfr2^ΔE18-T2A-Myc*, *n* = 12 of 6; *Ccnd1*, *n* = 6 of 2; *Fgfr2^FL-T2A-Fgf3*, *n* = 12 of 8; *Fgfr2^FL-P2A-Ccnd1*, *n* = 14 of 9; *Fgfr2^FL-T2A-Fgf3-P2A-Ccnd1*, *n* = 15 of 8; *Fgfr2^ΔE18-T2A-Fgf3*, *n* = 12 of 7; *Fgfr2^ΔE18-P2A-Ccnd1*, *n* = 13 tumours of 7 mice. *Trp53^F/F*;(*Rosa26-Cas9*), Lenti-*Cre*, *n* = 10 of 8; *Fgfr2^FL-P2A-Cre*, *n* = 8 of 5; *Fgfr2^ΔE18-P2A-Cre*, *n* = 15 of 10; *sgPten-Cre*, *n* = 15 of 9; *sgPten-Fgfr2^FL-P2A-Cre*, *n* = 14 of 7; *sgPten-Fgfr2^ΔE18-P2A-Cre*, *n* = 14 tumours of 9 mice. In **d**, **e**, one tissue section per MG was stained and quantified for each of the indicated stains acquired in 4 independent randomized batches across all *Fgfr2* variants and genotypes.

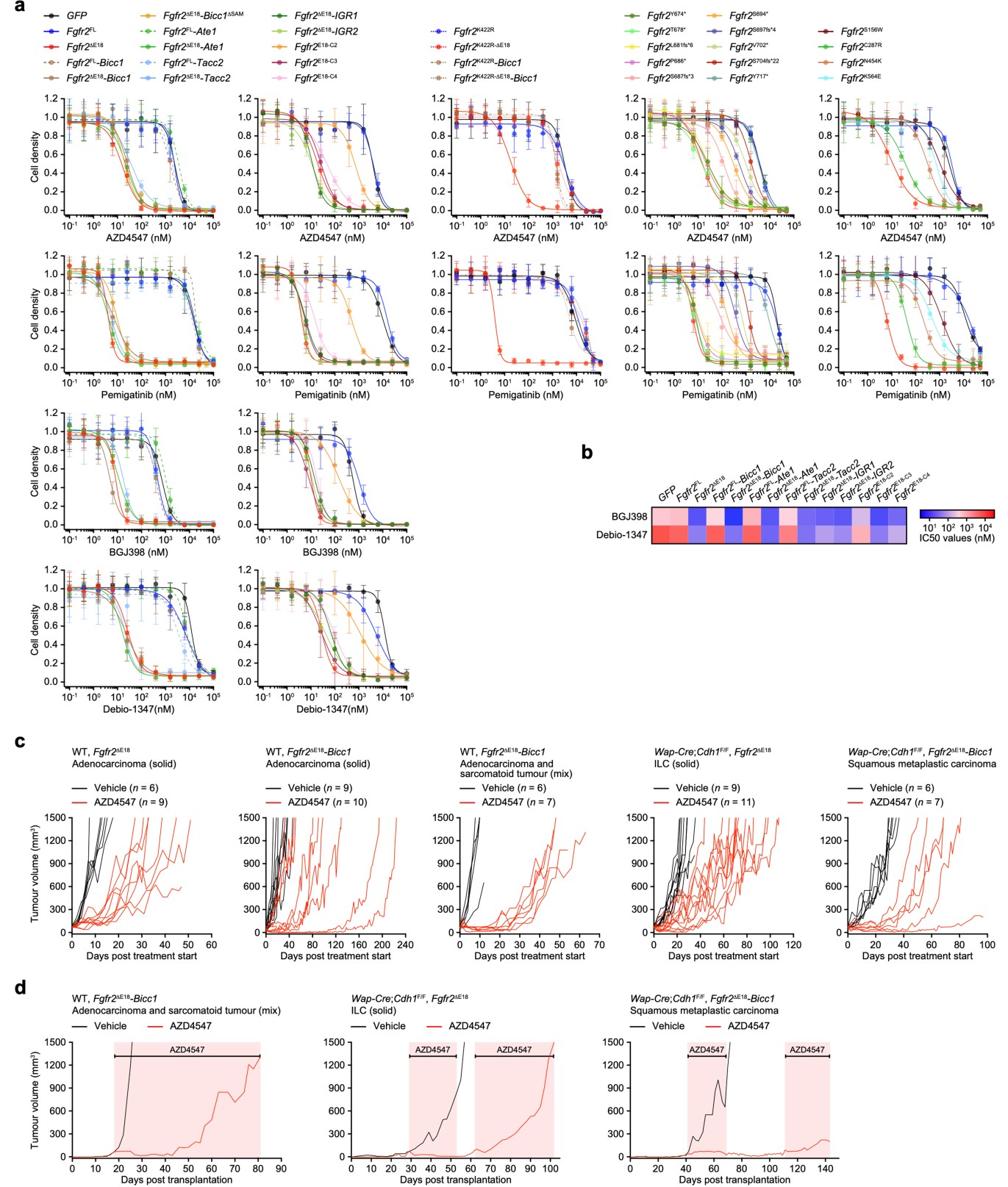

**Extended Data Fig. 12 | In vitro and in vivo sensitivity of *Fgfr2* variants to FGFRi. a**, Dose-response curves of 2D-grown NMuMG cells expressing *GFP* or indicated *Fgfr2* variants and treated with AZD4547, pemigatinib, BGJ398, or debio-1347 for 4 days. Data are represented as mean ± s.d. of *n* = 5 replica per group collected across 5 independent experiments. **b**, Half-maximum inhibitory concentration (IC$_{50}$) value quantifications of BGJ398 and debio-1347 dose-response curves in **a**. Data are represented as mean of 3 independent

experiments (*GFP*, *Fgfr2*$^{FL}$, *Fgfr2*$^{ΔE18}$) or 1 experiment (other *Fgfr2* variants). IC50 values for AZD4547 and pemigatinib are displayed in Fig. 4a. **c**, Individual growth curves of indicated tumour donors transplanted into the mammary fat pad of female syngeneic WT mice and treated daily orally with vehicle or 12.5 mg/kg AZD4547 using a previously established[55] intermittent dosing regimen. **d**, Selected tumour transplant growth curves of mice in **c**. Durations of AZD4547 treatments according to intermittent dosing regimen are indicated.

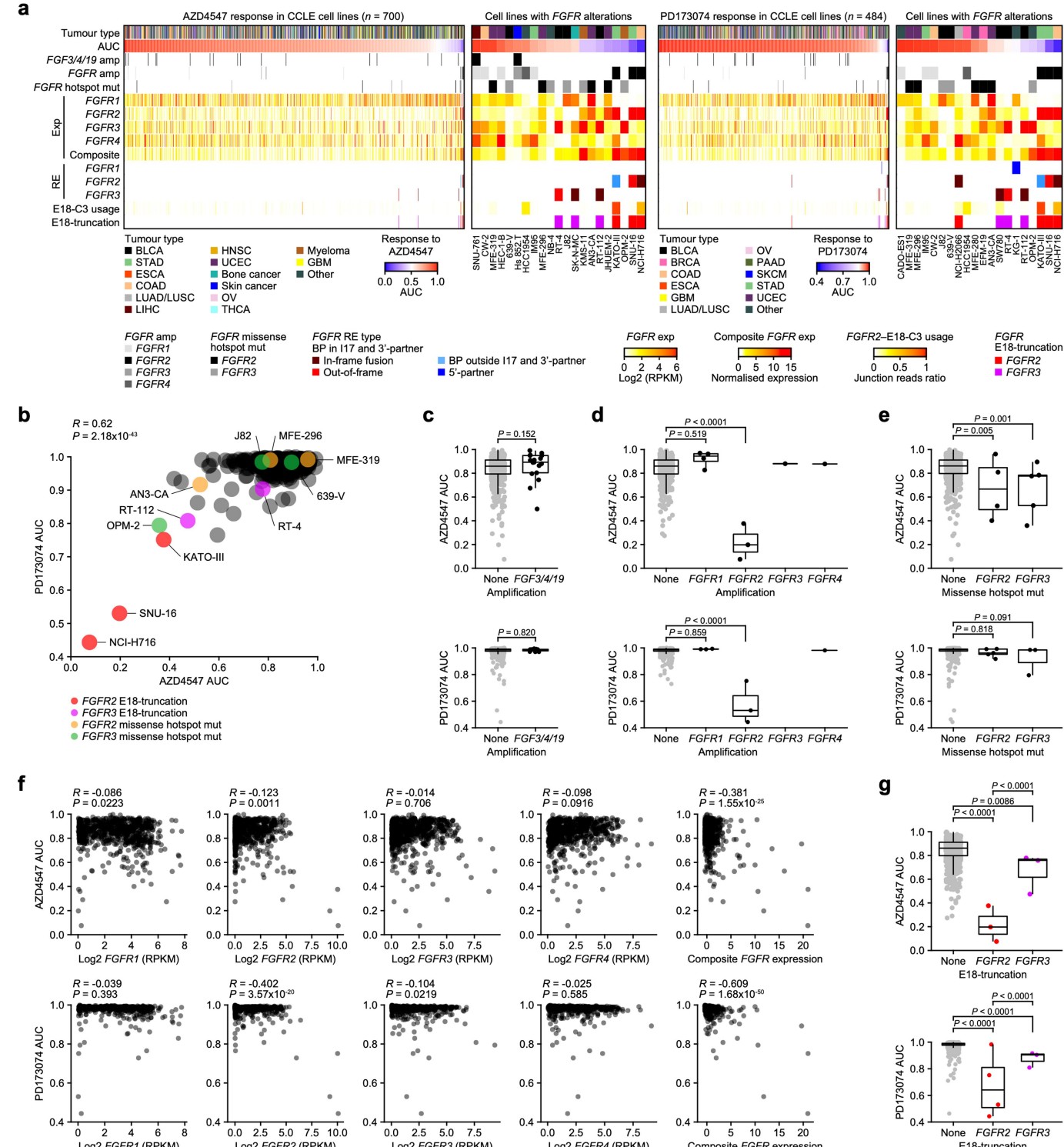

**Extended Data Fig. 13** | See next page for caption.

**Extended Data Fig. 13 | Correlation of *FGFR2* alterations to FGFRi sensitivity across CCLE. a**, The Broad Institute Cancer Cell Line Encyclopedia (CCLE) cell lines rank-ordered according to their AZD4547 or PD173074 FGFRi area under the sigmoid-fit concentration-response curve (AUC) values derived from the Cancer Therapeutics Response Portal (CTRP) v2 deposited in the PharmacoDB database (*n* = 700) and the Genomics of Drug Sensitivity in Cancer (GDSC) database (*n* = 484), respectively. Data on *FGF/FGFR* mutations, CN status, REs, and expression and *FGFR2*–E18-C3 usage was obtained from CCLE. RPKM, reads per kilobase of transcript per million mapped reads. **b**, Correlation of AZD4547 versus PD173074 AUC values across shared CCLE cell lines (*n* = 384). **c**, AZD4547 and PD173074 AUC values in CCLE cell lines with *FGF3/4/19* amp (AZD4547, *n* = 17; PD173074, *n* = 16) versus unaltered cell lines (AZD4547, *n* = 658; PD173074, *n* = 455). **d**, AZD4547 and PD173074 AUC values in CCLE cell lines with *FGFR1* amp (AZD4547, *n* = 4; PD173074, *n* = 3), *FGFR2* amp (AZD4547, *n* = 3; PD173074, *n* = 3), *FGFR3* amp (AZD4547, *n* = 1; PD173074, *n* = 0), or *FGFR4* amp (AZD4547, *n* = 1; PD173074, *n* = 1) versus unaltered cell lines (AZD4547, *n* = 666; PD173074, *n* = 464). **e**, AZD4547 and PD173074 AUC values in CCLE cell lines with *FGFR2* missense hotspot mut (AZD4547, *n* = 4; PD173074, *n* = 5) or *FGFR3* missense hotspot mut (AZD4547, *n* = 5; PD173074, *n* = 3) versus unaltered cell lines (AZD4547, *n* = 691; PD173074, *n* = 476). **f**, Correlations of *FGFR1*, *FGFR2*, *FGFR3*, *FGFR4*, or composite *FGFR* expression versus AZD4547 or PD173074 AUC values across CCLE cell lines. **g**, AZD4547 or PD173074 AUC values in CCLE cell lines expressing E18-truncated *FGFR2* (AZD4547, *n* = 3; PD173074, *n* = 4) or *FGFR3* (AZD4547, *n* = 3; PD173074, *n* = 3) versus no truncation (AZD4547, *n* = 694; PD173074, *n* = 477). Data in **c**–**e**, **g** are represented as median (centre line) ± IQR (25th to 75th percentile, box) and IQR ± 1.5 x IQR (whiskers) . *P* values were calculated with two-tailed *t*-transformations of Pearson's *R* correlation coefficients (**b**, **f**), two-tailed Wilcoxon rank-sum tests (**c**), or one-tailed one-way ANOVA and Tukey's multiple-testing corrections (**d**, **e**, **g**).

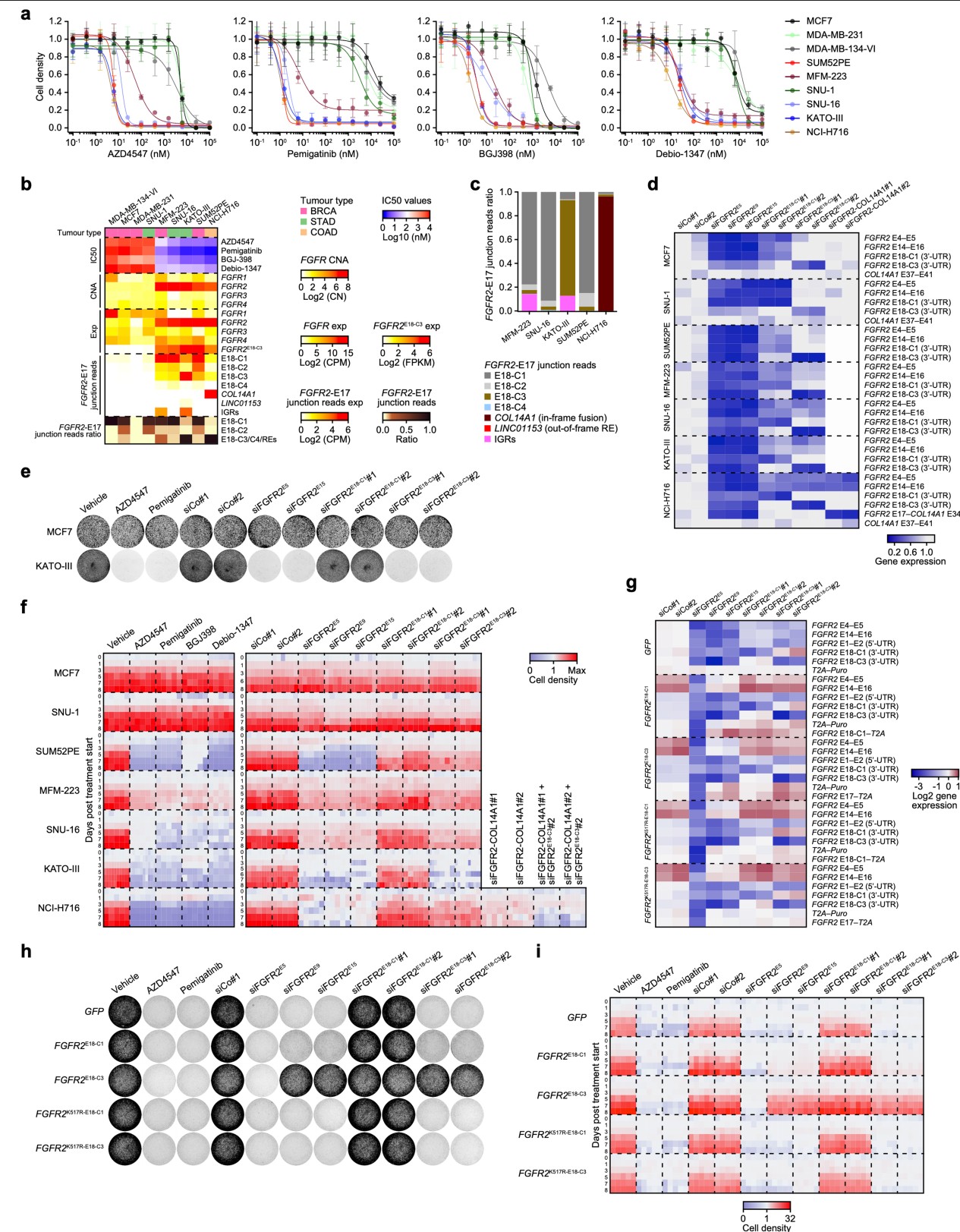

**Extended Data Fig. 14 |** See next page for caption.

**Extended Data Fig. 14 | Human cancer cell lines depend on *FGFR2^ΔE18* variants. a**, Dose-response curves of indicated human cancer cell lines treated with AZD4547, pemigatinib, BGJ398, or debio-1347 for 4 days. Data are represented as mean ± s.d. of *n* = 5 replica per group collected across 2 independent experiments. **b**, Human cancer cell lines rank-ordered according to IC$_{50}$ values for indicated FGFRi in (**a**). *FGFR1-4* CNA and expression is based on low-coverage WGS and RNA-seq profiles. *FGFR2^E18-C3* isoform expression, *FGFR2*-E17 junction reads expression, and *FGFR2* RE types were identified in RNA-seq profiles. FPKM, fragments per kilobase of transcript per million mapped reads. **c**, Distribution of *FGFR2*-E17 junction reads to canonical full-length E18-C1 versus noncanonical E18-C2/C3/C4, RE partners, and IGRs in indicated human cancer cell lines. **d**, Heatmap showing silencing of *FGFR2* variants in indicated human cancer cell lines using small interfering (si) RNAs. Cells were transfected with the following siRNAs: non-targeting siRNAs (siCo), siRNAs targeting shared exons among *FGFR2* isoforms (siFGFR2^E5,E9,E15), siRNAs specifically targeting canonical E18-C1 of *FGFR2^FL* (siFGFR2^E18-C1) or E18-C3 of truncated *FGFR2^E18-C3* (siFGFR2^E18-C3), or siRNAs specifically targeting the *FGFR2-COL14A1* fusion (siFGFR2-COL14A1). Silencing of specific *FGFR2* variants was detected with RT-qPCR using primers spanning indicated cDNA segments. Expression of each cDNA segment is normalized to *USF1* expression and cDNA segment expression in siCo condition of each cell line (average of siCo#1 and siCo#2). Data are represented as mean of *n* = 3 technical replica per group. Data represent 1 replica of 2 independent experiments. **e, f**, Representative images of 6-well plate wells at 8 days post treatment start (**e**) and cell density quantifications over 8 days (**f**) of 2D-grown indicated cell lines treated with vehicle or 100 nM AZD4547, pemigatinib, BGJ398, or debio-1347 or (co)-transfected with siCo, siFGFR2^E5, siFGFR2^E9, siFGFR2^E15, siFGFR2^E18-C1,

siFGFR2^E18-C3, and/or siFGFR2-COL14A1. Data in **f** represent *n* = 6 independent replica collected across 1 experiment (MCF7, siRNA treatments), *n* = 10 independent replica collected across 2 independent experiments (KATO-III vehicle, AZD4547, and siRNA treatments), or *n* = 5 independent replica collected across 2 independent experiments (other cell lines and/or treatment conditions). **g**, Heatmap showing silencing of *FGFR2* isoforms using indicated siRNAs in KATO-III cells expressing *GFP* or indicated *FGFR2* variants. Validation of overexpression of *FGFR2* variants using RT-qPCR is in Supplementary Table 4. *FGFR2^K517R* variants encode KD-dead FGFR2 variants. siFGFR2^E5 targets endogenous *FGFR2* transcripts and *FGFR2* transcripts derived from lentiviral constructs. Other siRNAs specifically target endogenous *FGFR2* transcripts. Silencing of specific *FGFR2* variants was detected with RT-qPCR using primers spanning indicated cDNA segments. E4–E5 and E14–E16 primers detect endogenous *FGFR2* transcripts and *FGFR2* transcripts derived from lentiviral constructs. E1–E2 (5′-UTR), E18-C1 (3′-UTR), and E18-C3 (3′-UTR) primers specifically detect endogenous *FGFR2* transcripts. E18-C1–T2A, E17–T2A, and T2A–Puro primers specifically detect *FGFR2* transcripts derived from lentiviral constructs. Log$_2$-transformed expression of each cDNA segment is normalized to *USF1* expression and cDNA segment expression in siCo-treated *GFP*-expressing cells (average of siCo#1 and siCo#2). Data are are represented as mean of *n* = 3 technical replica per group of 1 experiment. **h, i**, Representative images of 6-well plate wells at 8 days post treatment start (**h**) and cell density quantifications over 8 days (**i**) of 2D-grown KATO-III cells expressing *GFP* or indicated *FGFR2* variants and treated with vehicle, 100 nM AZD4547, or 100 nM pemigatinib or transfected with siCo, siFGFR2^E5, siFGFR2^E9, siFGFR2^E15, siFGFR2^E18-C1, or siFGFR2^E18-C3. Data in **i** represent *n* = 6 independent replica per group collected across 1 experiment.

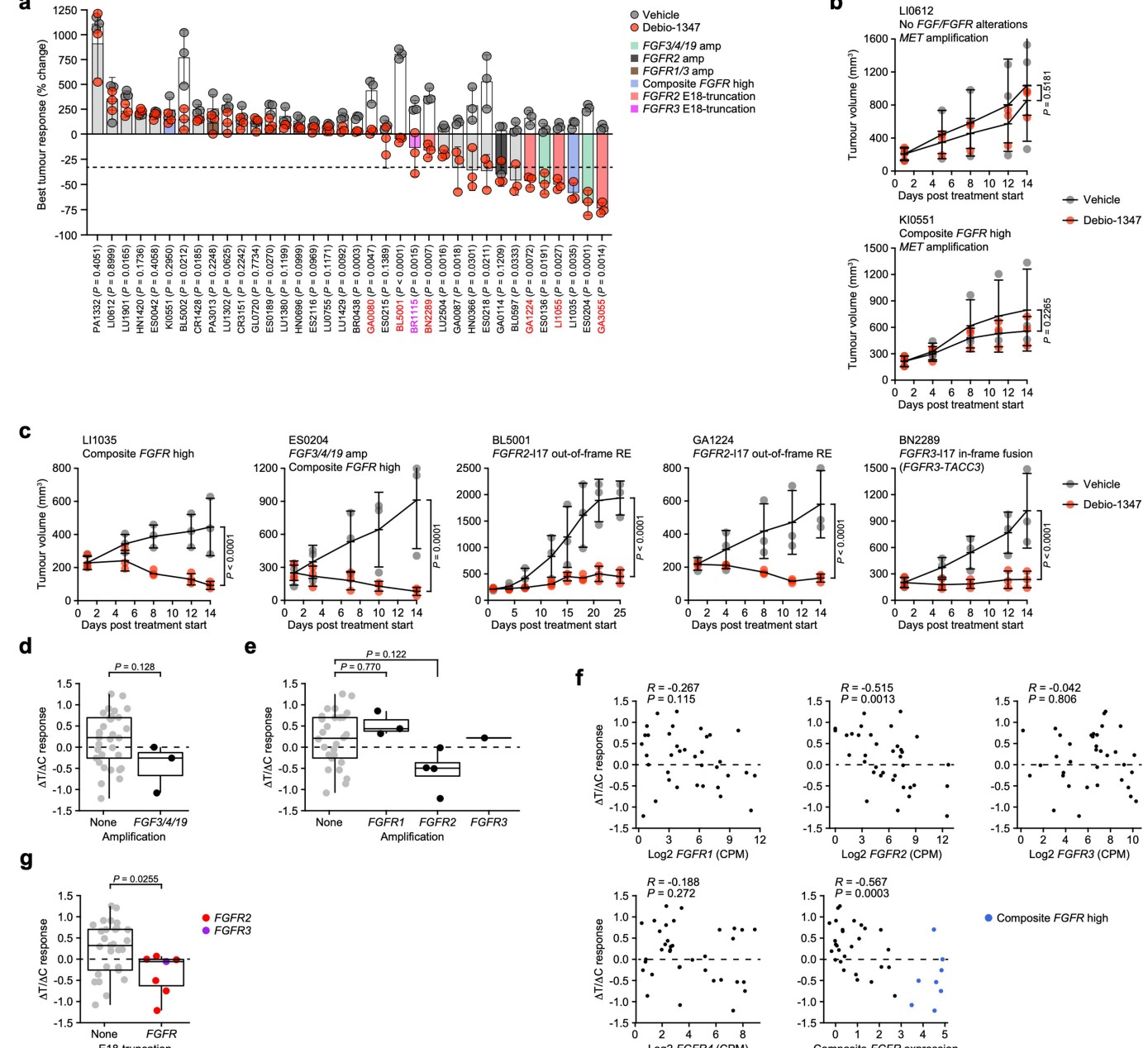

**Extended Data Fig. 15 | PDXs expressing *FGFR2^ΔE18* variants are sensitive to debio-1347. a**, Best percentage change from baseline tumour volume in patient-derived xenograft (PDX) models (*n* = 36) engrafted in female NOD-SCID (BL5001, BL5002) or BALB/c Nude (all other PDX models) mice and treated daily orally with vehicle or debio-1347 (*n* = 3 mice per PDX model and treatment group). Coloured bars indicate identified *FGF/FGFR2* alterations detailed in Fig. 4c. **b, c**, Growth curves of indicated PDXs models engrafted in NOD-SCID or BALB/c Nude mice and treated daily orally with vehicle or debio-1347 (ES0204, Li0612, LI1035, BN2289, 60 mg/kg; GA1224, KI0551, 80 mg/kg; BL5001, day 1-14, 40 mg/kg; day 15-25, 60 mg/kg; *n* = 3 mice per PDX model and treatment group). **d**, Debio-1347 *ΔT/ΔC* response ratios in PDXs with *FGF3/4/19* amp (*n* = 3) versus normal CN (*n* = 33). **e**, Debio-1347 *ΔT/ΔC* response ratios in PDXs with *FGFR1* amp (*n* = 3), *FGFR2* amp (*n* = 4), or *FGFR3* amp (*n* = 1) versus normal CN

(*n* = 28). **f**, Correlations of *FGFR1*, *FGFR2*, *FGFR3*, *FGFR4*, or composite *FGFR* expression versus debio-1347 *ΔT/ΔC* response ratios across PDXs. Composite *FGFR* expression was defined as high, if normalized expression > 3. **g**, Debio-1347 *ΔT/ΔC* response ratios in PDXs expressing E18-truncated *FGFR2* (*n* = 6) or *FGFR3* (*n* = 1) versus no truncation (*n* = 29). Data are represented as mean ± s.d. (**a, b**) or as median (centre line) ± IQR (25th to 75th percentile, box) and IQR ± 1.5 x IQR (whiskers) (**d, f**). *P* values were calculated with two-tailed unpaired Student's *t*-tests (**a**), one-tailed two-way ANOVA and FDR multiple-testing corrections using the two-stage step-up method from Benjamini, Krieger, and Yekutieli (**b, c**), two-tailed Wilcoxon rank-sum tests (**d, g**), one-tailed one-way ANOVA and Tukey's multiple-testing corrections (**e**), or two-tailed *t*-transformations of Pearson's *R* correlation coefficients (**f**).

# Reporting Summary

## Statistics

For all statistical analyses, confirm that the following items are present in the figure legend, table legend, main text, or Methods section.

| n/a | Confirmed | |
|---|---|---|
| ☐ | ☒ | The exact sample size (*n*) for each experimental group/condition, given as a discrete number and unit of measurement |
| ☐ | ☒ | A statement on whether measurements were taken from distinct samples or whether the same sample was measured repeatedly |
| ☐ | ☒ | The statistical test(s) used AND whether they are one- or two-sided<br>*Only common tests should be described solely by name; describe more complex techniques in the Methods section.* |
| ☒ | ☐ | A description of all covariates tested |
| ☐ | ☒ | A description of any assumptions or corrections, such as tests of normality and adjustment for multiple comparisons |
| ☐ | ☒ | A full description of the statistical parameters including central tendency (e.g. means) or other basic estimates (e.g. regression coefficient) AND variation (e.g. standard deviation) or associated estimates of uncertainty (e.g. confidence intervals) |
| ☐ | ☒ | For null hypothesis testing, the test statistic (e.g. *F*, *t*, *r*) with confidence intervals, effect sizes, degrees of freedom and *P* value noted<br>*Give P values as exact values whenever suitable.* |
| ☒ | ☐ | For Bayesian analysis, information on the choice of priors and Markov chain Monte Carlo settings |
| ☒ | ☐ | For hierarchical and complex designs, identification of the appropriate level for tests and full reporting of outcomes |
| ☐ | ☒ | Estimates of effect sizes (e.g. Cohen's *d*, Pearson's *r*), indicating how they were calculated |

*Our web collection on statistics for biologists contains articles on many of the points above.*

## Software and code

Policy information about availability of computer code

| | |
|---|---|
| Data collection | Sequencing data of Sleeping Beauty tumours was obtained from European Nucleotide Archive (ENA) with accession code PRJEB14134. WGS and RNA-seq data from HMF were downloaded from their google cloud computing platform under data sharing agreement DR-138. TCGAbiolinks (version 2.14.1; R package) was used to download TCGA raw RNA-seq data from GDC Data Portal (https://portal.gdc.cancer.gov/). CGP data were obtained from FMI via their standard data requesting process. For PDX models, genomic information and RNA-seq data was obtained from the CrownBio HuPrime data portal (https://www.crownbio.com/oncology/in-vivo-services/patient-derived-xenograft-pdx-tumor-models). Pharmacogenomic datasets of human cancer cell lines were obtained from CCLE (https://sites.broadinstitute.org/ccle/datasets), GDSC (https://www.cancerrxgene.org/), CTRP (https://portals.broadinstitute.org/ctrp.v2.1/), and PharmacoDB (https://www.pharmacodb.ca/) portals. FIGHT-202 oncogenomic data were obtained from Incyte via their standard data requesting process. Human reference genome for sequencing data analysis was obtained from the CTAT Genome Lib data resources (https://data.broadinstitute.org/Trinity/CTAT_RESOURCE_LIB/). Domain-domain interaction information of proteins was obtained from the 3did (https://3did.irbbarcelona.org) and the PPIDM (http://ppidm.loria.fr) databases. The SLIPPER Golden Standard Dataset of potentially (self-)interacting proteins was previously published by Liu, Z. et al. Proteome-wide prediction of self-interacting proteins based on multiple properties. Mol. Cell. Proteomics 12, 1689–700 (2013). |
| Data analysis | Analyses of in vitro and in vivo experiments were performed in Prism (version 9.3.1, GraphPad Software). All omics data analyses were performed in R (versions 3.6.3 - 4.1.2). Custom computer codes used to analyse the genomics and proteomics data in this study are available at https://doi.org/10.5281/zenodo.6630874 and https://doi.org/10.5281/zenodo.6630632, respectively.<br><br>Software used:<br>- Sample sizes calculation, G*Power software (version 3.1)<br>- IVIS measurements, Living Image Software (version 4.5.2, PerkinElmer)<br>- Digital processing of HE and IHC slides, CaseViewer software (version 2.2.1, 3DHISTECH)<br>- FACS analyses and quantifications thereof, BD FACSDiva Software (version 8.0.2, BD Biosciences) and FlowJo (version 10.7.1, BD Biosciences)<br>- Imaging of cell luminescence, fluorescence, or absorbance, Tecan i-control software (version 3.9.1, Tecan) |

- 3D colony formation quantifiction, GelCount colony counting platform (version 1.1.2, Oxford Optronix)
- Primer design for In-Fusion, SnapGene (version 5.2)
- Primer design for site-directed mutagenesis, QuikChange Primer Design (https://www.agilent.com/store/primerDesignProgram.jsp)
- Primer design for RT-qPCR, Primer-BLAST (https://www.ncbi.nlm.nih.gov/tools/primer-blast/)
- RT-qPCR recording, QuantStudio Real-Time PCR Software (version 1.7.2, Thermo Fisher Scientific)
- Western blot recording , Fusion FX7 Edge imaging system (version 18.05, Vilber)
- Western blot post-imaging processing, Photoshop 2022 (version 23.2.2, Adobe)
- Western blot band intensity quantification,  Fiji (version 1.0)
- LC-MS/MS operation, Tune (version 2.11) and Xcalibur Software (version 4.3.73.11, #OPTON-30965, both Thermo Fisher Scientific)
- HiSeq 2500 System operation, HiSeq Control Software (version 2.2.68, Illumina)
- NextSeq 500 System operation, NextSeq Control Software (version 2.0.2, Illumina)
- NovaSeq 6000 System operation, NovaSeq Control Software (version 1.7.5, Illumina)

Software/packages used for proteomics analyses:
- MS/MS spectra annotation, MaxQuant software (version 2.0.3.0) with default settings and the Swissprot M. musculus reference proteome
- Spectral library generation from MaxQuant output file, Spectronaut software (version 15.4.210913, Biognosys)
- Single sample gene set enrichment analysis (ssGSEA), via GenePattern platform (https://www.genepattern.org) using the ssGSEA module (version 10.0.11) and hallmark gene sets from MSigDB (version 7.0)
- Differential expression analysis on class 1 phosphosite intensity data, R package limma (version 3.52.1)
- (Single sample) phosphosite signature enrichment analysis ((ss)PTM-SEA), via GenePattern platform (https://www.genepattern.org) using a 7-AA sequence flanking the phosphosite as identifier and the murine kinase/pathway definitions of PTMsigDB (version 1.9.0)
- Robust kinase activity inference (RoKAI) tool (https://rokai.io), used with default settings and the Uniprot M. musculus reference proteome

Software/packages used for genomics analyses:
- Analysis of WGS data from HMF, published computer code  at https://github.com/hartwigmedical/pipeline5
- WGS read mapping, BWA-MEM (version 0.7.5a) and the human reference genome GRCh37
- WGS somatic SV calling, GRIDSS (version 1.8.0)
- WGS CNA and tumour purity estimation, PURPLE (version 2.43)
- WGS event annotation and derivate chromosome construction, LINX (version 1.9)
- Low-coverage WGS read mapping, BBWA-MEM (version 0.7.5a) and the human reference genome GRCh38
- Low-coverage WGS alignment analysis, QDNAseq (version 1.14.0) with 20,000 bp bin size
- TCGA raw RNA-seq data download from GDC Data Portal (https://portal.gdc.cancer.gov/), TCGAbiolinks (version 2.14.1; R package)
- RNA-seq read mapping, STAR (version 2.7.2) and STAR-Fusion (version 1.8.1) and the human reference genome GRCh38 gencode v32 CTAT
- RNA-seq gene and exon level expression quantification, featureCounts (version 1.6.2)
- RNA-seq read count normalisation, Trimmed Mean of M-value (TMM) method via edgeR (version 3.26.6)
- RNA-seq sashimi plots, Integrated Genomics Viewer (version 1.11.0, https://igv.org/app/).
- hybrid-capture RNA-seq read mapping, STAR (version 2.7.3a), RSEM (version 1.3.0) and LeafCutter (version 0.2.9)
- PDX-derived RNA-seq mapping, Disambiguate (version 2018.05.03-6) to filter mouse (mm10 gencode M23) from human (GRCh38 gencode v32 CTAT)-derived reads
- Genome coordinates conversion from GRCh37 to GRCh38, UCSC Lift Genome Annotations (https://genome.ucsc.edu/cgi-bin/hgLiftOver)
- Self-interacting capacity of FGFR2 RE partners, SLIPPER algorithm (Liu et al., 2013, Proteome-wide prediction of self-interacting proteins based on multiple properties)
- Domain enrichment analysis among FGFR2 RE partners, DAVID bioinformatic resources (https://david.ncifcrf.gov/)

For manuscripts utilizing custom algorithms or software that are central to the research but not yet described in published literature, software must be made available to editors and reviewers. We strongly encourage code deposition in a community repository (e.g. GitHub). See the Nature Portfolio guidelines for submitting code & software for further information.

# Data

Policy information about availability of data

All manuscripts must include a data availability statement. This statement should provide the following information, where applicable:
- Accession codes, unique identifiers, or web links for publicly available datasets
- A description of any restrictions on data availability
- For clinical datasets or third party data, please ensure that the statement adheres to our policy

All data generated or analysed during this study are included in this article and its Supplementary Information. Source data are provided with this paper.

- The low-coverage WGS and RNA-seq data of human cell lines generated in this study are available in the European Nucleotide Archive (ENA) under accession number PRJEB42514
- The mass spectrometry proteomics data and MaxQuant-generated text files generated in this study are available in the ProteomeXchange Consortium under accession numbers PXD031711 (Fgfr2 samples) and PXD032007 (KB1P(M) samples)
- Sequencing data of SB-tumours were previously published (Kas et al., 2017) and are available in ENA under accession number PRJEB14134
- WGS and RNA-seq data from HMF can be obtained through standardized procedures and request forms at https://www.hartwigmedicalfoundation.nl/en/
- CGP data can be obtained from FMI on reasonable request at https://www.foundationmedicine.com/service/genomic-data-solutions
- Data from TCGA, CCLE, CTRPv2, and GDSC are available through the respective data portals at https://portal.gdc.cancer.gov/, https://sites.broadinstitute.org/ccle/datasets, https://www.cancerrxgene.org/, https://portals.broadinstitute.org/ctrp.v2.1/, and https://www.pharmacodb.ca/
- Details on PDXs can be obtained from the CrownBio-HuPrime data portal at https://www.crownbio.com/oncology/in-vivo-services/patient-derived-xenograft-pdx-tumor-models.
- The FIGHT-202 study was previously published (Abou-Alfa et al., 2020). Information on Incyte's clinical trial data sharing policy and instructions for submitting clinical trial data requests are available at https://www.incyte.com/Portals/0/Assets/Compliance%20and%20Transparency/clinical-trial-data-sharing.pdf?ver=2020-05-21-132838-960
- Human reference genome (GRCh38 gencode v32 CTAT) used for RNA-seq data analysis is available in CTAT Genome Lib data resources at https://data.broadinstitute.org/Trinity/CTAT_RESOURCE_LIB
- SLIPPER list of self-interacting proteins was previously published (Liu et al., 2013, Proteome-wide prediction of self-interacting proteins based on multiple

properties)
- 3did and PPIDM domain-domain interaction information is available at https://3did.irbbarcelona.org and http://ppidm.loria.fr
- Reference human and mouse Swissprot proteome information is available in the UniProt database at https://www.uniprot.org

# Field-specific reporting

Please select the one below that is the best fit for your research. If you are not sure, read the appropriate sections before making your selection.

☒ Life sciences   ☐ Behavioural & social sciences   ☐ Ecological, evolutionary & environmental sciences

For a reference copy of the document with all sections, see nature.com/documents/nr-reporting-summary-flat.pdf

# Life sciences study design

All studies must disclose on these points even when the disclosure is negative.

| Sample size | Sample size determinations for the FIGHT-202 (NCT02924376) clinical trial were previously described (Abou-Alfa et al., 2020), and were large enough to measure the effect sizes. Sample sizes in the SB-transposon insertional mutagenesis screen and ex vivo analyses thereof were previously published (Kas et al., 2017), and were large enough to measure the effect sizes. Sample sizes for the remaining in vivo experiments and follow-up ex vivo experiments were determined using G*Power software (version 3.1) and were large enough to measure the effect sizes. For in vitro experiments, no sample size calculations were performed. Instead, in vitro experiments' sample sizes were based on previous experiences. For all in vitro experiments, large enough sample sizes were obtained to appropriately evaluate statistical differences between experimental groups and to ensure reproducibility. |
|---|---|
| Data exclusions | No data had to be excluded. |
| Replication | For in vitro global expression proteomics and global phosphoproteomics using IMAC enrichment, 4 independent NMuMG cell replica per Fgfr2 variant tested were collected and subjected to the two methodologies in parallel. An independent set of 3 replica per variant was collected for -Tyr-specific proteomics using p-Tyr-IP enrichment. Sample processing and LC-MS/MS measurements were performed across 3 randomized batches per methodology. One global phosphoproteomic sample had to be excluded because of low protein concentration (Fgfr2-dE18-Bicc1_4). Other in vitro experiments were repeated independently at least twice, and all attempts at replication were successful. Across these independent experiments, data on a total of at least 3 independent replica were collected. In vivo tumorigenesis and intervention studies were each performed as 'single' experiments, but in sufficient mice (= independent biological replica) to measure the effect sizes. For intraductal injections of lentiviruses, virus variants were injected in parallel in several batches across several days, rather than injection each virus in series. No obvious differences in tumour latencies were observed in between 'replica' of injection. For the intervention studies, tumour-bearing mice were continuously randomly allocated to treatment arms when tumours reached treatment-starting size. No obvious response differences were observed in between mice with faster versus slower growing tumours. All attempts at replication were successful. Ex vivo H&E + IHC stainings and analyses of mouse mammary tissues was performed in multiple independent batches. For each tissue sample, one H&E and/or IHC stain per marker was evaluated. All attempts at replication were successful. Ex vivo FACS analyses were done on pooled mammary gland samples from at least 4 different mice per time-point analysed. Typically, per batch of analysis 1-2 mice per Fgfr2 variant from one time-point were analysed. All attempts at replication were successful. Ex vivo proteomics was performed on individual mouse mammary tumours subjecting each sample to three (phospho)-proteomic methodologies in parallel (global expression proteomics, global phosphoproteomics using IMAC enrichment, and p-Tyr-specific proteomics using p-Tyr-IP enrichment). Sample processing and LC-MS/MS measurements were performed across several randomized batches. A few p-Tyr-IP samples hat to be excluded because of low protein concentrations (Fgfr2-FL_1, Fgfr2-FL-Ate1_1, Fgfr2-FL-Bicc1_2, Fgfr2-FL-Tacc2_2, Fgfr2-dE18-Ate1_2, Fgfr2-dE18-Tacc2_2, Fgfr2-E18-C4_1). |
| Randomization | Allocation of mice into lentivirus injection cohorts, tumour fragment transplantation cohorts, as well as into treatment arms was randomized. For DNA/RNA-seq and proteomics experiments, samples/replica were randomized during processing and data acquisition. For other experiments, no randomization strategy was applied. |
| Blinding | Animal care takers and animal pathologists were actively blinded towards mouse examinations and histopathological evaluations. For other experiments, no active blinding strategy was applied. Yet, experiments were performed by a multitude of researchers and technicians, the majority of whom were agnostic to the outcome of experiments. |

# Reporting for specific materials, systems and methods

We require information from authors about some types of materials, experimental systems and methods used in many studies. Here, indicate whether each material, system or method listed is relevant to your study. If you are not sure if a list item applies to your research, read the appropriate section before selecting a response.

## Materials & experimental systems

| n/a | Involved in the study |
|-----|----------------------|
| ☐ | ☒ Antibodies |
| ☐ | ☒ Eukaryotic cell lines |
| ☒ | ☐ Palaeontology and archaeology |
| ☐ | ☒ Animals and other organisms |
| ☐ | ☒ Human research participants |
| ☐ | ☒ Clinical data |
| ☒ | ☐ Dual use research of concern |

## Methods

| n/a | Involved in the study |
|-----|----------------------|
| ☒ | ☐ ChIP-seq |
| ☐ | ☒ Flow cytometry |
| ☒ | ☐ MRI-based neuroimaging |

# Antibodies

| | |
|---|---|
| Antibodies used | Mouse IgG1 monoclonal anti-β-Actin, Sigma-Aldrich, #A5441, lot#127M4866V, clone#AC-15<br>Rabbit monoclonal anti-AKT1, Cell Signaling Technology, #2938, lot#4, clone#C73H10<br>Rabbit monoclonal anti-p(S473)-AKT, Cell Signaling Technology, #4060, lot#25, clone#D9E<br>Rabbit monoclonal anti-E-cadherin, Cell Signaling Technology, #3195, lot#10, clone#24E10<br>Rabbit monoclonal anti-Cyclin D1, Abcam, #ab16663, lot#GR249365-2, clone#SP4<br>Rabbit polyclonal anti-EIF4B, Cell Signaling Technology, #3592, lot#3<br>Rabbit polyclonal anti-p(S422)-EIF4B, Cell Signaling Technology, #3591, lot#6<br>Rabbit monoclonal anti-EIF4EBP1, Cell Signaling Technology, #9644, lot#10, clone#53H11<br>Rabbit monoclonal anti-p(T37/T46)-EIF4EBP1, Cell Signaling Technology, #2855, lot#17, clone#236B4<br>Rat monoclonal anti-EpCAM, BV650-conjugated, BD Biosciences, #740559, lot#1187955, clone#G8.8<br>Rabbit monoclonal anti-ERK1/2, Cell Signaling Technology, #4695, lot#28, clone#137F5<br>Rabbit polyclonal anti-p(T202/Y204)-ERK1/2, Cell Signaling Technology, #9101, lot#30 and 31<br>Rabbit polyclonal anti-FGF3, LifeSpan BioSciences, #LS-B11923, lot#53099<br>Rabbit monoclonal anti-FGFR2, Cell Signaling Technology, #11835, lot#4 and 5, clone#D4H9<br>Rabbit polyclonal anti-p(Y653/Y654)-FGFR, Cell Signaling Technology, #3471, lot#8 and 12<br>Goat polyclonal anti-GFP, Abcam, #ab6673, lot#GR3371856-3<br>Rabbit monoclonal anti-C-MYC, Abcam, #ab32072, lot#GR189790-46, clone#Y69<br>Rabbit polyclonal anti-P53, Leica Biosystems, #NCL-L-p53-CM5p, lot#6070664<br>Rabbit monoclonal anti-PTEN, Cell Signaling Technology, #9559, lot#12, clone#138G6<br>Rabbit monoclonal anti-RPS6, Cell Signaling Technology, #2217, lot#10, clone#5G10<br>Rabbit polyclonal anti-p(S235/S236)-RPS6, Cell Signaling Technology, #2211, lot#23<br>Rabbit multi-monoclonal anti-p-Tyr mix, Cell Signaling Technology, #8954, lot#13<br>Donkey polyclonal anti-goat IgG (H+L), AF488-conjugated, Thermo Fisher Scientific, #A-11055, lot#2301114<br>Donkey polyclonal anti-rabbit IgG (H+L), AF647-conjugated, Thermo Fisher Scientific, #A32795, lot#WA308388<br>Goat polyclonal anti-mouse IgG (H+L), HRP-conjugated, Thermo Fisher Scientific, #G-21040, lot#1925065<br>Goat polyclonal anti-rabbit IgG (H+L), HRP-conjugated, Dako, #P0448, lot#20083037 |
| Validation | Primary antibodies used for IHC were rabbit monoclonal anti-E-cadherin (CST #3195), rabbit monoclonal anti-Cyclin D1 (Abcam #ab16663), rabbit polyclonal anti-FGF3 (LifeSpan #LS-B11923), rabbit monoclonal anti-FGFR2 (CST #11835), rabbit monoclonal anti-C-MYC (Abcam #ab32072), rabbit polyclonal anti-P53 (Leica Biosystems #NCL-L-p53-CM5p), and rabbit monoclonal anti-PTEN (CST #9559). The antibodies were independently validated by a certified pathologist by evaluation of IHC results in positive and negative biological control FFPE tissues to ensure specificity and sensitivity. In addition, negative technical controls were performed by omission of the primary antibody in extra sections for a randomly selected small subset of the samples.<br><br>The primary antibody BV650-conjugated rat monoclonal anti-EpCAM (BD Biosciences #740559) used for FACS was validated by BD Biosciences (https://www.bdbiosciences.com/en-ca/products/reagents/flow-cytometry-reagents/research-reagents/single-color-antibodies-ruo/purified-rat-anti-mouse-cd326.552370) for this application. The primary antibodies goat anti polyclonal anti-GFP (Abcam #ab6673) and rabbit monoclonal anti-FGFR2 (CST #11835) were validated for FACS using cells overexpressing EGFP or FGFR2 versus control cells negative for GFP or FGFR2. These controls were taken along for each FACS experiment on mouse derived mammary glands and in vitro-cultured NMuMG cells.<br><br>The primary control antibody used for Western blotting was mouse IgG1 monoclonal anti-β-Actin (Sigma-Aldrich #A5441) and has been validated by Sigma-Aldrich for this application (https://www.sigmaaldrich.com/NL/en/product/sigma/a5441).<br>All other primary antibodies used for Western blotting or p-Tyr IPs were derived from Cell Signaling Technology (CST) and were validated for specificity and sensitivity in the respective applications by CST according to their rigid Antibody Validation Principles (https://www.cellsignal.com/about-us/cst-antibody-validation-principles). Details on each antibody and its validation data are in the following links:<br>Rabbit monoclonal anti-AKT1 (CST #2938, https://www.cellsignal.com/products/primary-antibodies/akt1-c73h10-rabbit-mab/2938)<br>Rabbit monoclonal anti-p(S473)-AKT (CST #4060, https://www.cellsignal.com/products/primary-antibodies/phospho-akt-ser473-d9e-xp-rabbit-mab/4060)<br>Rabbit polyclonal anti-EIF4B (CST #3592, https://www.cellsignal.com/products/primary-antibodies/eif4b-antibody/3592)<br>Rabbit monoclonal anti-EIF4EBP1 (CST #9644, https://www.cellsignal.com/products/primary-antibodies/4e-bp1-53h11-rabbit-mab/9644)<br>Rabbit monoclonal anti-p(T37/T46)-EIF4EBP1 (CST #2855, https://www.cellsignal.com/products/primary-antibodies/phospho-4e-bp1-thr37-46-236b4-rabbit-mab/2855)<br>Rabbit monoclonal anti-ERK1/2 (CST #4695, https://www.cellsignal.com/products/primary-antibodies/p44-42-mapk-erk1-2-137f5-rabbit-mab/4695)<br>Rabbit polyclonal anti-p(T202/Y204)-ERK1/2 (CST #9101, https://www.cellsignal.com/products/primary-antibodies/phospho-p44-42- |

mapk-erk1-2-thr202-tyr204-antibody/9101)
Rabbit monoclonal anti-FGFR2 (CST #11835, https://www.cellsignal.com/products/primary-antibodies/fgf-receptor-2-d4h9-rabbit-mab/11835)
Rabbit polyclonal anti-p(Y653/Y654)-FGFR (CST #3471. https://www.cellsignal.com/products/primary-antibodies/phospho-fgf-receptor-tyr653-654-antibody/3471)
Rabbit monoclonal anti-RPS6 (CST #2217, https://www.cellsignal.com/products/primary-antibodies/s6-ribosomal-protein-5g10-rabbit-mab/2217)
Rabbit polyclonal anti-p(S235/S236)-RPS6 (CST #2211, https://www.cellsignal.com/products/primary-antibodies/phospho-s6-ribosomal-protein-ser235-236-antibody/2211)
Rabbit multi-monoclonal anti-p-Tyr mix (CST #8954, https://www.cellsignal.com/products/primary-antibodies/phospho-tyrosine-p-tyr-1000-multimab-rabbit-mab-mix/8954)

# Eukaryotic cell lines

Policy information about cell lines

| | |
|---|---|
| Cell line source(s) | HEK 293T cells (#CRL-3216, ATCC), MCF7 (#HTB-22, ATCC), MDA-MB-134-VI (#HTB-23, ATCC), MDA-MB-231 (#HTB-26, ATCC), NCI-H716 (#CCL-251, ATCC), NMuMG (#CRL-1636, ATCC), KATO-III (#HTB-103, ATCC), SNU-1 (#CRL-5971, ATCC), SNU-16 (#CRL-5974, ATCC), MFM-223 (#98050130, ECACC), and SUM52PE (#HUMANSUM-0003018, BioIVT). |
| Authentication | Cell lines were previously authenticated by providers. No re-authentication was performed for this study. |
| Mycoplasma contamination | Routine mycoplasma testing repeatedly confirmed all cell lines used to be negative for mycoplasma via the MycoAlert Mycoplasma Detection Kit (#LT07-218, Lonza). |
| Commonly misidentified lines (See ICLAC register) | No commonly misidentified cell lines were used in this study. |

# Animals and other organisms

Policy information about studies involving animals; ARRIVE guidelines recommended for reporting animal research

| | |
|---|---|
| Laboratory animals | Mice, all female. <br> - GEMMs and somatic mouse models: FVB/NRj background, 6-week-old. Strains: WT, Wap-Cre;Cdh1F/F, Wap-Cre;Cdh1F/F;Fgfr2-FL-IRES-Luc, Wap-Cre;Cdh1F/F;Fgfr2-dE18-IRES-Luc, Trp53F/F, Trp53F/F;Rosa26-Cas9, Rosa26-mT/mG. <br> - AZD4547 intervention: FVB/NRj WT mice, 8-week-old. <br> - PDXs: BALB/cAnNRj-Foxn1nu/nu or NOD.CB17-Prkdcscid/NCrHsd mice, 8-week-old. <br> The maximal permitted disease endpoints were not exceeded in any of the experiments. The mouse colony was housed in a certified animal facility with a 12-hour light/dark cycle in a temperature- and humidity-controlled room set to 21 °C and 55% relative humidity. Mice were kept in individually ventilated cages, and food and water were provided ad libitum. |
| Wild animals | No wild animals were used in the study. |
| Field-collected samples | No field collected samples were used in the study. |
| Ethics oversight | GEMMs and somatic mouse models: All animal experiments were approved by the Animal Ethics Committee of the Netherlands Cancer Institute and performed in accordance with institutional, national and European guidelines for Animal Care and Use. <br> PDXs: All the procedures related to animal handling, care, and the treatment in this intervention study were performed according to guidelines approved by the Institutional Animal Care and Use Committee (IACUC) of Crown Bioscience following the guidance of the Association for Assessment and Accreditation of Laboratory Animal Care (AAALAC). |

Note that full information on the approval of the study protocol must also be provided in the manuscript.

# Human research participants

Policy information about studies involving human research participants

| | |
|---|---|
| Population characteristics | The FIGHT-202 trial was previously published. Abou-Alfa, G. K. et al. Pemigatinib for previously treated, locally advanced or metastatic cholangiocarcinoma: a multicentre, open-label, phase 2 study. Lancet Oncol. 21, 671–684 (2020). Population characteristics are described in detail in Abou-Alfa et al., 2020, and its appendix. Briefly, the trial was done at 146 academic or community-based sites in the USA, Europe, the Middle East, and Asia. Age, median (range), 59 (26 to 78). Sex, male, 42%; female, 58%. Region, North America, 61%; Western Europe, 24%; Rest of world, 15%. Race, White, 71%; Asian, 15%; Black or African American, 6%; American Indian or Alaska Native, 1%; Other or data missing, 8%. Metastatic disease, 86%; Previous cancer surgery, 33%; Previous radiotherapy, 23%; Previous systemic therapies, 100%. |
| Recruitment | Details on patient recruitment were previously published (Abou-Alfa et al., 2020). Briefly, patients were identified during routine clinical practice. Eligible patients were aged 18 years or older and had a histological or cytological diagnosis of locally advanced or metastatic cholangiocarcinoma with documented disease progression following at least one previous systemic cancer therapy. Before assessment for eligibility, patients were pre-screened centrally for FGF/FGFR status using massively parallel DNA-sequencing (FoundationOne). Patients who already had an FGF/FGFR status report based on local assessment or an existing FoundationOne report were also included. Retrospective central confirmation of locally documented FGF/FGFR status with FoundationOne was required for cohort assignment. Based on the centrally confirmed results, patients were assigned to one of three cohorts: patients with FGFR2 fusions/REs, patients with other FGF/FGFR alterations, or patients with no FGF/FGFR alterations. No self-selection or other biases were observed. |

| Ethics oversight | The study protocol ( see Abou-Alfa et al., 2020, appendix p26) was approved by each institutional review board or independent ethics committee; the trial was performed in accordance with the Declaration of Helsinki. Patients gave written, informed consent for inclusion in the study. |
|---|---|

Note that full information on the approval of the study protocol must also be provided in the manuscript.

# Clinical data

Policy information about clinical studies

All manuscripts should comply with the ICMJE guidelines for publication of clinical research and a completed CONSORT checklist must be included with all submissions.

| Clinical trial registration | FIGHT-202 (NCT02924376) |
|---|---|
| Study protocol | The trial was previously published. Abou-Alfa, G. K. et al. Pemigatinib for previously treated, locally advanced or metastatic cholangiocarcinoma: a multicentre, open-label, phase 2 study. Lancet Oncol. 21, 671–684 (2020).<br><br>Study protocol is in the appendix (p26) of Abou-Alfa et al., 2020.<br>INCB 54828-202 |
| Data collection | Data collection is described in  Abou-Alfa et al., 2020 and its appendix. |
| Outcomes | Primary and secondary outcomes are described in Abou-Alfa et al., 2020.<br>The primary endpoint was the proportion of patients with FGFR2 fusions or rearrangements who achieved an objective response (best overall response of confirmed complete response or confirmed partial response), assessed by independent central review. Secondary endpoints were the proportion of patients with an objective response in patients with other FGF/FGFR alterations, in all patients with FGF/FGFR alterations, and in patients with no FGF/FGFR alterations, and duration of response, the proportion of patients with disease control, progression-free survival, overall survival, safety in all cohorts, and population pharmacokinetics (data to be reported separately). Progression-free survival was defined as the time from first dose to progressive disease or death, overall survival was defined as the time from first dose to death from any cause, duration of response was defined as the time from complete response or partial response to progressive disease or death, and disease control was defined as complete response, partial response, or stable disease. |

# Flow Cytometry

## Plots

Confirm that:

☒ The axis labels state the marker and fluorochrome used (e.g. CD4-FITC).

☒ The axis scales are clearly visible. Include numbers along axes only for bottom left plot of group (a 'group' is an analysis of identical markers).

☒ All plots are contour plots with outliers or pseudocolor plots.

☒ A numerical value for number of cells or percentage (with statistics) is provided.

## Methodology

| Sample preparation | Mammary glands of Rosa26-mT/mG female mice injected with Fgfr2-P2A-Cre lentiviruses were minced and digested with 4 mg/ml collagenase A (#11088793001, Roche) and 25 ug/ml DNase I (#DN25, Sigma Aldrich) in Dulbecco's Modified Eagle Medium/Nutrient Mixture F-12 (DMEM/F-12, #31331, Thermo Fisher Scientific) containing 100 IU/ml penicillin and streptomycin (Pen Strep, #15070, Thermo Fisher Scientific) for 1 hr at 37 °C. Digests were passed through a 70 um cell strainer prewetted with PBS containing 10% Foetal Bovine Serum (FBS, #S-FBS-EU-015, Serana) and 2 mM EDTA (FACS buffer). Single cells were stained with BV650-conjugated anti-EpCAM antibody (1:100, #740559, BD Biosciences) in FACS buffer, labelled with the LIVE/DEAD Fixable Violet Dead Cell Stain Kit (405 nm excitation, #L34964, Thermo Fisher Scientific), fixed with BD Phosflow Fix Buffer I (#557870, BD Biosciences), and permeabilised with BD Phosflow Perm Buffer III (#558050, BD Biosciences), each for 30 min at 4 °C. Cells were incubated with primary antibodies overnight and subsequently with secondary antibodies for 1 hr both in FACS buffer and at 4 °C.<br>Cultured NMuMG cells were collected with 2mM EDTA and passed through a 70 um cell strainer prewetted with FACS buffer. Single cells were labelled with the LIVE/DEAD Fixable Violet Dead Cell Stain Kit and fixed with BD Phosflow Fix Buffer I, and permeabilised with BD Phosflow Perm Buffer III, each for 30 min at 4 °C. Cells were incubated with primary antibodies overnight and subsequently with secondary antibodies for 1 hr both in FACS buffer and at 4 °C.<br><br>Antibodies used are the following:<br>Rat monoclonal anti-EpCAM, BV650-conjugated, BD Biosciences, #740559, , lot#1187955, clone#G8.8, 1:100<br>Rabbit monoclonal anti-FGFR2, Cell Signaling Technology, #11835, lot#4 and 5, clone#D4H9, 1:200<br>Goat polyclonal anti-GFP, Abcam, #ab6673, lot#GR3371856-3, 1:200<br>Donkey polyclonal anti-goat IgG (H+L), AF488-conjugated, Thermo Fisher Scientific, #A-11055, lot#2301114, 1:400<br>Donkey polyclonal anti-rabbit IgG (H+L), AF647-conjugated, Thermo Fisher Scientific, #A32795, lot#WA308388, 1:400 |
|---|---|
| Instrument | BD LSRFortessa Cell Analyzer (BD Biosciences) equipped with 405 nm (450/50, 670/30 pass filters), 488 nm (530/30 pass filters), and 638 nm (670/30 pass filters) lasers. |

| Software | Data were analysed with BD FACSDiva Software (version 8.0.2, BD Biosciences) and FlowJo (version 10.7.1, BD Biosciences). |
|---|---|
| Cell population abundance | FACS sorting was not performed, thus post-sorting purity of fractions was not assessed. |
| Gating strategy | NMuMG cells were gated for (i) FSC-A / SSC-A to select bulk of cells and exclude debris events, (ii) SSC-A / SSC-H to select single cells, (iii) FSC-A / 405-Live/Dead to select live cells, (iv) FSC-A / FGFR2, AF647 to gate and subsequently display FGFR2 intensity as histogram and measure FGFR2 MFI. See Supplementary Figure 1a.<br>Mammary gland-derived cells were gated for (i) FSC-A / SSC-A to select bulk of cells and exclude debris events, (ii) SSC-A / SSC-H to select single cells, (iii) FSC-A / 405-Live/Dead to select live cells, (iv) FSC-A / BV650-EpCAM to select EpCAM+ cells, (v) FSC-A / EGFP, AF488 to select EGFP- and EGFP+ cells, (vi) FSC-A / FGFR2, AF647 for gating (not shown) to subsequently display FGFR2 intensity as histogram and measure FGFR2 MFI of EGFP- and EGFP+ cells. See Supplementary Figure 1b. |

☒ Tick this box to confirm that a figure exemplifying the gating strategy is provided in the Supplementary Information.

