## [Peer Review File · Nature]

Manuscript Title: Truncated FGFR2 is a clinically actionable oncogene in multiple cancers.

Reviewer Comments & Author Rebuttals

Reviewer Reports on the Initial Version:

Referees' comments:

Referee #1 (Remarks to the Author):

Zingg et al present a robust and highly interesting manuscript on the oncogenicity of truncated FGFR2. The authors present compelling mouse modelling evidence that truncated FGFR2 is an oncogene, and combine with robust clinical data confirming that these truncated forms of FGFR2 are clinically relevant (if extremely rare) with evidence of benefit from a clinical trial. The data presented in this paper supports the implementation of clinical sequencing of patients to detect these variants.

The principal weakness of the manuscript, which is not highlighted sufficiently by the authors, is the prior published work demonstrating the oncogenicity of truncated FGFR2. This does reduce the novelty of the findings. C terminal truncations of the receptor in FGFR2 amplified cancers have been reported for 20 years, and prior work has confirmed that the c terminal translocations will transform cells (resulting in growth in soft agar), with biochemical work confirming that the c terminal truncations reduce receptor recycling, enhancing oncogenic signalling. The presented work does go very substantially beyond prior work, but this prior work must be clearly acknowledged and presented in the first paragraph of the main text. However, the work using GEMMs and mice transfected with constructs encoding *fgfr2* variants provides additional, important evidence that FGFR2 variants with exon 18 truncations are oncogenic.

Figure 1. The work with the GEMMs in Figure 1H and I is very convincing in demonstrating that FGFR2 variants with E18 truncations are oncogenic. This is powerful, additional evidence that FGFR2 truncations are oncogenic, substantially extending prior work.

Figure 2. The data presented in this figure suggests that although translocations in intron 17 of FGFR2 exist in a range of cancer types, these structural variants are very rare. This could limit the potential applications of this work. The authors should present the incidence of the non-fusion rearrangements in the text, and provide estimates of incidence by tumor type. Can the authors provide additional comments on incidence of C3 usage without an underlying rearrangement. The data presented that these variants have RNA support is strong.

Figure 3. It has long been an area of uncertainty over whether the FGFR2 fusions are oncogenic through simply through the fusion partner enhancing dimerisation, or through deletion of Exon 18. Prior studies published on the fusions suggested that the fusion partner was required not just deletion of exon 18.

In figure 3A it appears that there is growth of NMuMG cells that express the *Fgfr2FL-Bicc1* construct

in soft agar. This indicates that the authors cannot definitively say that the fusion proteins themselves are not oncogenic. In extended Figure 7 it appears that the Fgfr2FL-Bicc1 is expressed at lower levels than the corresponding Fgfr2ΔE18-Bicc1 construct, and this complicates assessment. I recommend that the authors either address this point, or substantially soften conclusions. Figure parts A-D are essentially similar data, and I do not think that all of these parts are needed.

Figures 3E-F figures 3E-F are a highly interesting demonstration that exon 18 deletion is required (and this reduces the concern in interpretation of figure 3A). As a control, to interpret this experiment the authors must demonstrate that the Fgfr2FL-Bicc1 (and other full length constructs) successfully transfect and express. Figure G is very nice data that demonstrates that tumours expressing this variant are sensitive to FGFRi.

Figure 4

Figure 4A. This data is not novel, there is substantial prior data demonstrating expression of truncated FGFR2 variants in amplified cancers, and that some of these cancers have an intron 17 rearrangement, that is amplified.

Figure 4B. I believe that KATO-III cells have amplified an intron 17 rearrangement, and thus express predominantly the C3 variant. As such this experiment cannot be interpreted as the authors suggest. It is probable that KATO-III cells express more of the Fgfr2ΔE18-C3 variant than Fgfr2ΔE18-C1. Therefore siRNA depletion of the Fgfr2ΔE18-C3 variant may cause the majority of the total Fgfr2 in the cell to be depleted. Conversely silencing of the C1 variant may have a relatively minor effect on total FGFR2. Therefore, this experiment cannot be used to support the relative oncogenicity of C1 vs C3 variants. Only a cell line with equal, or lower, protein expression of C3 could be used to support the hypothesis.

Extended data figure 13B appears to indicate that depletion of the Fgfr2ΔE18-C3 variant leads to a less dramatic decrease in cell density when compared to depletion of the Fgfr2ΔE18-C1 variant in other cell lines. How do the expression levels of these two variants compare in these cell lines at the protein level?

I recommend that unless the authors can address my concerns that figure parts 4A-C are moved to the appendix, and conclusions from the data substantially softened.

Figure 4D/E: Compelling evidence that patient derived models with c terminal truncations are sensitive to FGFR inhibitor.

Figure 5: This figure provides compelling evidence that c terminal truncations without fusion partners are clinically actionable.

It is important to point out that this study does illustrate the clear prior understanding that c terminal truncations without fusion partners would be oncogenic and targetable. In this registration study, the design of which clearly predates this manuscript by a number of years, patients with c terminal truncations without fusion partners were recruited into the same cohort as patients with FGFR2-fusions, reflecting the prior understanding that these patients would have a similar sensitivity to therapy.

Nonetheless, this data clearly adds strong support to the highly interesting pre-clinical data. The authors must confirm that the data in this figure has not been published before in prior publications on this trial, including in the very recently published manuscript in Cancer Discovery “Clinicogenomic Analysis of FGFR2-Rearranged Cholangiocarcinoma Identifies Correlates of Response and Mechanisms of Resistance to Pemigatinib”

Figure 5C: the required statistics have not been carried out.

Referee #2 (Remarks to the Author):

The authors previously reported that *Fgfr2* is the most frequently affected gene in mouse mammary tumors with conditional E-cadherin (*Cdh1*) inactivation and Sleeping Beauty (SB) insertion.

In this study, the authors re-discovered that preferential SB insertions within intron 17 of the mouse *Fgfr2* gene give rise to upregulation of C-terminally truncated *Fgfr2* because of exon 18 (E18) deficit through transcriptional truncation within the SB transposon. They demonstrated that E18-defective *Fgfr2* variants promote mammary tumorigenesis in conditional *Cdh1*^{+/-} or *Cdh1*^{-/-} mice. They found that E18-defective FGFR2 variants also occur in a variety of human cancers, such as biliary tract, breast, gastric, lung and ovarian cancers, owing to in-frame fusions, intragenic rearrangements and point mutations. They then demonstrated that mouse mammary tumors induced by E18-defective *Fgfr2* variants are sensitive to FGFR inhibitors irrespective of presence or absence of C-terminal fusions.

Results presented in this manuscript are of great interest and value for diverse groups of researchers.

In contrast, the following points were noticed as issues to be further addressed:

(1) Functional genomics of E18-defective *Fgfr2* variants. It remains unclear how E18-defective *Fgfr2* variants induce malignant phenotypes. Because E18 of *Fgfr2* consists of the C-terminal coding region and 3'-untranslated region (UTR), C-terminal deletion of *Fgfr2* protein and 3'-UTR deletion of *Fgfr2* mRNA might be involved in enhanced tumorigenesis. In addition to structural variations, nonsense point mutations cause C-terminal truncation of human FGFR2 protein (Extended Data Figure 3). The authors need to investigate the effects of (i) C-terminal nonsense mutations in E18 of mouse *Fgfr2* and/or (ii) 3'-UTR deletion from E18 of mouse *Fgfr2* to shed light on the precise mechanisms of malignant phenotypes driven by E18-defective *Fgfr2* variants.

(2) Functional proteomics of E18-defective *Fgfr2* variants. It also remains unclear how E18-defective *Fgfr2* variants elicit intracellular signaling dysregulation, although the authors showed western blot analyses of mouse tumors using anti-FGFR2 antibody and anti-Y653/Y654-phosphorylated FGFR antibody (Extended Data Figure 7b). They are advised to carry out phosphoproteome analyses on *Fgfr2* and its substrates in mouse tumors presented in Figure 2.

(3) Fusion partners of FGFR2. The authors illustrated several fusion partners of FGFR2 in Extended Data Figure 3g; however, entire FGFR2 “fusionome” remains to be displayed. To increase the value of this manuscript for readers, the authors need to list-up all binding partners of in-frame FGFR2 fusions depicted in Figure 2b and 2c.

(4) Clinical genomics of E18-defective FGFR2 variants. Some cases with FGFR2 C-terminal recombination are accompanied by FGFR2 gene amplification (Figure 2). E18-defective FGFR2 variants could be generated through breakage-fusion-bridge cycles during FGFR2 gene amplification, which might elicit intra-tumor or intra-patient heterogeneity and affect the response of cancer patients to FGFR2-targeted therapeutics. The authors are advised to investigate whether response rates of FGFR inhibitors differ between cases of FGFR2 C-terminal recombination with FGFR2 gene amplification and those without FGFR2 gene amplification.

Referee #3 (Remarks to the Author):

The increasing availability of tumor molecular profiling by next-generation sequencing provides the opportunity to tailor therapies to potential targets. The manuscript by Zingg et al. “Truncated FGFR2 is a clinically actionable oncogene in multiple cancers” describes the discovery and characterization of exon18-truncated FGFR2 as an oncogenic driver and a potential biomarker/target for targeted therapies with FGFR inhibitors in cancers. The authors performed a number of functional analyses using engineered expression constructs in 2d, 3d cultures, and pdx models to establish FGFR2-e18 truncation as a novel oncogenic aberration that gives proliferative advantage to cells in culture, and imparts sensitivity to FGFR inhibitors, similar to cancer cells harboring FGFR2 gene fusions. Overall, this study provided sufficient evidence in supporting the pathogenicity of E18-truncated-FGFR2 in cancer and demonstrated drug sensitivity of this type of FGFR2 aberrations towards FGFR inhibitors. However, while the discovery of 3' truncation events as potential driver using SB screen is a good starting point, the analysis and description of the FGFR2 aberrations in cancer cohorts needs much improvement. In addition, additional controls for the functional analyses and drug treatment experiments specified below may help make a more convincing case.

Major concerns:

1. The strength of the paper is the demonstration through a wide-ranging set of experiments that truncation of FGFR2 contributes to the oncogenicity of FGFR2, in cases where this aberration is found. The cataloging of cases and the recognition that this type of event is not usually tested for is an important contribution to precision oncology. The experiments are well done and clearly presented. They add significant further evidence that truncation of FGFR2 plays a role in oncogenesis and correctly cite Szybowska as demonstrating the critical role of p.S780 in the regulation of FGFR2. Szybowska also pointed out that truncation is an oncogenic mechanism, and identified clinical cases with truncation events. The authors should cite them in that context.

2. The manuscript, however, suffers from a central and repeated flaw. The authors oversimplify the complexity of FGFR2 activation, genetic aberrations of all types, and clinical responses to a small number of inhibitors in order to claim that truncation of FGFR2 plays a “central” role in FGFR2 oncogenicity. As well established by numerous groups and publications over several decades,

oncogenicity of receptor tyrosine kinases is a complex system, with amplification, expression, mutations (both missense and indels), gene fusions, alterations in ligand and interacting proteins all playing roles to different degrees in different cases. The authors' own transposon data demonstrate that expression plays an important role, as well as most in vitro experiments couple overexpression with a genetic aberration of a second type. The genetic evidence on FGFR2 is quite substantial and clearly indicate amplification, in-frame fusions, and mutations both in the kinase domain and extracellular domains play functional roles in the development of several human cancer types. This role in oncogenesis should not be confused with clinical response. For example, point mutations in tyrosine kinases are often quite unpredictable with regard to clinical response to an inhibitor. The point mutations in FGFR2 do not seem to respond clinically to small molecule inhibitors against the kinase domain as well as other aberrations. More work needs to be done to answer why. But no conclusions should be drawn as to ranking the relative role of this aberration vs. another. By occurrence alone, recurrent missense mutations could be judged to be "most" important, but again this is not a proper conclusion from the data. The paper needs to be revised with the recognition that all the aberration classes of FGFR2 play a role, perhaps interacting and synergistic in some cases. The use of clinical response data is very important in the selection of cases containing aberration classes most likely to respond, but should not be used to conclude whether any aberration class is more "central" to oncogenesis. The authors have repeatedly presented a skewed interpretation of the role of fusions in FGFR2 oncogenicity.

A) Their own data illustrates in-frame fusions (and frame undetermined, which may very well be complex yet in-frame) are more prevalent than intergenic breakpoints.

B) The fusions are recurrent by gene (a large set with some partner genes highly recurrent) and more importantly, the fusions are functionally recurrent in the sense that nearly all if not all 3' partner genes to date retain a known dimerization or oligomerization domain in the fusion protein. The number of genes is large (other FGFRs have fusions with the same properties) and increasing. The overall dataset highlights the decidedly nonrandom attributes of the 3' partner genes.

C) Breakpoints in intergenic space are enriched in amplification cases relative to in-frame fusions, highlighting that in-frame fusions in particular are not byproducts of amplification.

D) The discussion concerning measured clinical benefit relative to genetic aberration type is convoluted at best and somewhat misleading. First, there is no specific evidence presented to date concerning non-fusion RE's vs. any other aberration type for clinical response. The current consensus data does indicate fusions may respond better than missense mutations and amplification. The statement, "Notably, cholangiocarcinoma patients with FGFR2-BICC1 fusions versus other fusion and RE partners equally responded to FGFR-targeted therapy, despite the potent SAM oligomerization domain in BICC1." is misleading. The distinction of FGFR2-BICC1 fusion from other fusions with oligomerization domains does not lead to supportable conclusions concerning the relative role of fusions vs. any other category of aberration. Would the authors please elaborate on what measure of "potency" of oligomerization domain they are referring to?

E) The statement, "In line with these clinical observations, we uncovered truncation of the C-terminus to be central to FGFR2 oncogenicity." is not a proper statement. Clinical responses to an inhibitor and oncogenicity are quite distinct measures and concepts, they should not be linked in this way.

F) The remaining discussion should be modified with a more complete and less simplistic view of all FGFR2 aberrations and their roles in oncogenesis and the separate issue of potential clinical benefit. Perhaps a more accurate final statement would be "Instead of considering patients based on FGFR2

fusion, amplification, or mutation status alone, our data suggest that identifying cancers with SVs or mutations that result in expression of FGFR2 E18-truncating variants will be an important additional biomarker for FGFR-targeted therapeutics and may significantly expand the number of cancer patients who may benefit from such therapy.” Such a statement would both make the author’s point of the importance of detecting this category of mutation, but not distort the larger view of all FGFR2 aberrations in precision oncology.

3. Figure 2a: the linear map of FGFR2 breakpoints in pan-cancer WGS data- please use color code to highlight truncating mutations/ breaks considered activating. Also include a parallel track showing the kinase domain and other functional domains to facilitate easy interpretation of domains affected by the different aberrations. Here, it will also help to include a track of typical breakpoints involved in FGFR2 gene fusions.

4. Figure 2b: FGFR2 breakpoint locations are displayed in three categories- 5’ to I17, I17, and E18. Arguably, only breaks within I17, and E18 are proposed to be activating, while breaks or deleterious mutations within 5’ to I17 would presumably be deleterious (for example, the kinase domain within Exon10 to Exon17, and more upstream regions encoding the receptor etc). The authors could declutter the figures by focusing on only the functionally relevant aberrations- not showing everything that the data analysis identified. Also, the cases with breakpoints in I17/ E18 seem far fewer in Figure 2a as compared to 2b- both representing the same HMF cohort. Please explain and reconcile the numbers as appropriate.

5. In Figure 2c, the track of “amplifications” distinguishes cases with amplification of E1-E17 (excluding E18)- the presumed activating variant, from those with full gene amplification (E1 to E18). However, Figure 2b (HMF cohort), and Figure 2e (TCGA cohort) do not show the amplification of the E18 excluding E1-E17 variant. This seems odd as a bona fide drive event would be expected to be seen across these cohorts as well.

6. The prevalence of "functional" FGFR2-RE" in human cancer is not clearly estimated. For example, 86 out of 2,112 HMF-WGS cases carried breakpoints (4.1%) in the FGFR2 gene. These breakpoints distributed throughout the entire FGFR2, but only alterations that retained an intact tyrosine kinase domain are potential oncogenic events. Likewise, number of cases with FGFR2-REs in the FMI cohort (249,570 cases) varies in different places: Figure 2c, n=1,307, Extended Data Figure 3b, n=2,245. Again, how many of these events are predicted to be pathogenic and targetable by FGFR inhibitors are not explicitly described. It would be helpful to provide a total of likely actionable FGFR2-positive cases for each cohort investigated.

7. Further, in the cases with FGFR2 amplification (E1 to E17)- the associated REs are presumably incidental and noisy, resulting from bridge fusion bridge cycles, not balanced translocation events that characterize more typical, recurrent gene fusions. I would suggest the removal those amplification- associated REs from the data tracks to help focus in on more specific putative driver aberrations. Notably, a large subset of cases with amplification E1-E17 are mutually exclusive from other events (Figure 2c). It may be worthwhile reviewing the cases with both amplifications and REs- for allelic balance between the two events. For example, in a case with 20 copies of FGFR2, multiple non-recurrent REs with a few reads each likely represent noise, and may be disregarded.

Incidentally, in Figure 2c (FMI cohort) and Figure 2e (TCGA cohort), the mutual exclusivity of cases with mutations in E18 with all other structural aberrations is very striking and credible.

8. Figure 2d circus plot is so busy it fails to convey anything specific. Per the comments above, please consider removing the REs associated with FGFR2 amplifications (as those are presumed noisy byproducts of amplification process, not specific driver events per se).

9. Figure 2b, c, e- the gender track doesn't appear to serve any purpose since in the pan-cancer dataset the gender distribution is quite dispersed as may be expected. Unless the authors want to convey something specific through this track- it should be removed.

10. Incidentally, querying for FGFR2 mutations in cBioportal, a curated set of 185 non-redundant studies (with a total of 48045 cancer samples), only 6 frameshift mutations were noted in Exon 18, not notably clustered there as may be expected in a locus that imparts selective advantage when mutated (for example, contrast this with c-terminal mutations in Notch1). This number seems much lower than the number of mutant cases shown in the three cohorts presented here (Figure 2). Can the authors please explain the reason for this? Also, in the functional assays with the panel of E18-truncated FGFR2 variants (Figure 3), please also include some frameshift truncating mutation disrupting E18, to fully span the range of aberrations noted in tumor samples. This is important since, REs and gene fusions presumably stabilize the E18 excluded FGFR2 gene, and it is very curious if the truncated FGFR2 resulting from stopgain/ frameshift indels are stable as well. It will also be useful to include some hotspot activating mutation(s) such as N549H/D/K or N550H as positive controls for sensitivity to FGFR inhibitors as well.

11. Figure 3: Interestingly, FGFR2-FL constructs show no proliferative/ drug sensitivity effects. In view of this, how do the authors see tumors with FGFR2 amplification (E1 to E18)- which are essentially wild type FGFR2, only expressed at much higher levels- seen in subsets of tumors across all pan-cancer cohorts analyzed here (Figure 2). Can they model potential phenotype of FGFR2 amplification by overexpressing it under a strong promoter? Perhaps this merits analysis of the version of E18-c2 versus c3/c4 predominant in amplified FGFR2 E1-E18 cases.

12. In transgenic mouse models, fgfr2-FL expression was never detected by IHC in any of the tested genetic conditions (Extended Data Figure 2). Even the low-grade lesions observed in Wap-Cre;Cdh1F/+; Fgfr2FL and Wap-Cre;Cdh1F/F;Fgfr2FL mammary glands were FGFR2-negative. This raised the concern of the integrity of the fgfr-FL construct.

13. Figure 4a- all of the gene fusions/REs in various cancer cell lines are included in the data matrix; it would be more informative to show the fusions/RE breakpoints in a linear exon map showing kinase domain location so that we can correlate the cluster of breaks within I17/E18 affected by these events. It would be particularly interesting to see FGFR2-PLPP4, and FGFR2-ACADSB with their breakpoints indicated as 5' to I17, being sensitive to various TKIs and FGFR inhibitors tested. Since immediately upstream of I17 is the kinase domain on FGFR2 (up to exon10), presumably the breakpoint would not be too far upstream or there is some conceptual difficulty in making sense of the data.

14. Figure 5, in the analysis of FIGHT202 cohort as well, please provide the coordinates of the

breakpoints of FGFR2 in cases with “unknown frame”, “not in frame” variants on a linear map of FGFR2 gene- exon domain structure to allow assessment of the retained domains and whether those are recurrent positions, similar to the pan-cancer data cohorts (Figure 2).

15. AZD4547 is a pan-FGFR inhibitor that has been reported to target FGFR1, FGFR2 and FGFR3 kinase activity with IC50 values of 0.2, 2.5 and 1.8 nM, respectively (Gavine et al. Cancer Research 2012). Although cell lines expressing E18-truncating FGFR3-REs also exhibited considerable AZD4547 sensitivity, it appears that cells with FGFR2-REs are more sensitive to AZD4547 (Extended Data Figure 12a). Are there data of other FGFR inhibitors in the CCLE database and do they show the same trend?

16. Stop-gain mutations in FGFR2-E18 were detected in a fraction of tumors, but their impact in FGFR2 protein function, tumorigenicity, and drug sensitivity were not discussed or tested in this study. Could the authors expand on the occurrence and position of this class of mutations in the cohorts they analyzed?

Zingg *et al.*, “Truncated *FGFR2* is a clinically actionable oncogene in multiple cancers”

Point-by-point response to referees’ comments

Dear Referees,

We are very pleased that you found our manuscript on the role of on the role of truncated FGFR2 as clinically actionable oncogene “robust and highly interesting”. Guided by your very constructive and helpful comments, we were able to considerably improve our study, revealing how C-terminally truncated FGFR2 versus full-length FGFR2 drive cancer. In the present revision, we have addressed all issues raised by you, and our revised study presents the following key improvements.

- C-terminally truncated FGFR2 encoded by an *Fgfr2* allele devoid of its last exon (E18) functions as a *bona fide* cancer driver, which we now demonstrated in genetically engineered and somatic mouse models using early timepoint and endpoint analyses.
- Using *in vitro* and *in vivo* modelling, we classified the full spectrum of potentially E18-truncating *FGFR2* variants of unknown clinical significance, including non-canonical rearrangements and truncating mutations, as oncogenic and actionable *FGFR2* alterations.
- We conducted mass-spectrometry-based (phospho)-proteomic analyses of mouse cell lines and tumours expressing different *Fgfr2* full-length and E18-truncation variants. This revealed that, as opposed to full-length FGFR2, C-terminal truncation of FGFR2 strongly enhanced signalling through the MAPK and PI3K/AKT/mTOR pathways.
- To decipher whether the oncogenicity of *FGFR2* fusions is fusion partner dependent, we conducted bioinformatics analyses of oncogenomic data and functional *in vitro* and *in vivo* analyses of various *Fgfr2* fusions. The oncogenomic data suggested enrichment of dimerising fusion partners. However, our *in vitro* and *in vivo* modelling approaches pinpointed oncogenicity of E18-truncating *Fgfr2* fusions to depend mostly on FGFR2 truncation-induced kinase domain activity and to a much lesser extent on dimerisation capacity of fusion partners.
- We now show that full-length *Fgfr2* can act as a tumour driver in a distinct oncogenomic context, whereas E18-truncated *Fgfr2* is a potent single-driver. Analysis of co-occurring driver gene alterations in the *FGFR2* amplification and E18-truncation cohorts identified distinct sets of driver genes co-occurring with the *FGFR2* alteration groups. Functional evaluation of these findings using somatic *in vivo* modelling showed that full-length *Fgfr2* overexpression is only marginally tumorigenic in the absence of other driver alterations but displays marked oncogenicity in conjunction with co-driver alterations such as loss of *Trp53* and *Pten* or overexpression of *Myc* or *Fgf3-Ccnd1*.
- Throughout the revised manuscript, we now implemented a clear distinction between oncogenic potential versus clinical response rates of *FGFR2* alteration types.

We hope that with the novel data and mechanistic insights provided by our work, you will find our revised manuscript suitable for publication in *Nature*. Please find below our point-by-point replies to your comments.

Referee #1 (Remarks to the Author):

Zingg et al present a robust and highly interesting manuscript on the oncogenicity of truncated FGFR2. The authors present compelling mouse modelling evidence that truncated FGFR2 is an oncogene, and combine with robust clinical data confirming that these truncated forms of FGFR2 are clinically relevant (if extremely rare) with evidence of benefit from a clinical trial. The data presented in this paper supports the implementation of clinical sequencing of patients to detect these variants.

The principal weakness of the manuscript, which is not highlighted sufficiently by the authors, is the prior published work demonstrating the oncogenicity of truncated FGFR2. This does reduce the novelty of the findings. C terminal truncations of the receptor in FGFR2 amplified cancers have been reported for 20 years, and prior work has confirmed that the c terminal translocations will transform cells (resulting in growth in soft agar), with biochemical work confirming that the c terminal truncations reduce receptor recycling, enhancing oncogenic signalling. The presented work does go very substantially beyond prior work, but this prior work must be clearly acknowledged and presented in the first paragraph of the main text. However, the work using GEMMs and mice transfected with constructs encoding *fgfr2* variants provides additional, important evidence that FGFR2 variants with exon 18 truncations are oncogenic.

- We appreciate that referee #1 considers our manuscript “robust and highly interesting”, our experimental strategies “compelling”, and agrees that the presented work goes “substantially beyond prior work”. In this revised version of the manuscript, previously published *in vitro* work on truncated FGFR2 is now appropriately mentioned and cited in lines 56-57, “... namely fusions and amplifications^{1,16}, some of which produce truncated FGFR2 isoforms¹⁸⁻²³”, and 127, “previous work indicated *in vitro*-transforming capabilities of C-terminally truncated FGFR2 isoforms^{21-23,31-33}”.

Figure 1. The work with the GEMMs in Figure 1H and I is very convincing in demonstrating that FGFR2 variants with E18 truncations are oncogenic. This is powerful, additional evidence that FGFR2 truncations are oncogenic, substantially extending prior work.

- We appreciate that the referee acknowledges our extensive GEMM-based *in vivo* validations of the *SB*-transposon screen findings on *Fgfr2* truncation. We have now moved these data to Extended Data Figure 11, in support of the comprehensive somatic *in vivo* modelling efforts presented in Figure 2c-e and Extended Data Figure 10.

Figure 2. The data presented in this figure suggests that although translocations in intron 17 of FGFR2 exist in a range of cancer types, these structural variants are very rare. This could limit the potential applications of this work. The authors should present the incidence of the non-fusion rearrangements in the text, and provide estimates of incidence by tumor type. Can the authors provide additional comments on incidence of C3 usage without an underlying rearrangement? The data presented that these variants have RNA support is strong.

- We agree that the different truncation variants affecting *FGFR2* are relatively rare in certain cancer types. However, we would like to point out that the total incidences in intrahepatic cholangiocarcinoma, extrahepatic cholangiocarcinoma, and stomach cancer are about 9%, 3%, and 2% (Extended Data Fig. 21), respectively, which are considerable frequencies. We have now presented the pan-cancer frequencies and frequencies per tumour type of different E18-truncation alterations, especially also of C3 and other expressed variants, in Extended Data Figures 21 (FMI data) and 5e (TCGA data) and the corresponding figure legends. Relevant incidences are furthermore mentioned in the manuscript text in lines 88-92 (FMI data). We would like to highlight that, in the era of personalized precision oncology, actionable alterations that are observed at frequencies comparable to the presented *FGFR2* alteration incidences are considered clinically relevant and pursued for targeting. For instance, *ROS1* fusions occur in 0.9-2.6% of non-small-cell lung cancers, and *ROS1*-targeting inhibitors have been approved as first-line therapies (Gendarme et al. 2022), thus emphasising the clinical relevance of our findings.

Figure 3. It has long been an area of uncertainty over whether the FGFR2 fusions are oncogenic through simply through the fusion partner enhancing dimerisation, or through deletion of Exon 18. Prior studies published on the fusions suggested that the fusion partner was required not just deletion of exon 18.

In figure 3A it appears that there is growth of NMuMG cells that express the Fgfr2FL-Bicc1 construct in soft agar. This indicates that the authors cannot definitively say that the fusion proteins themselves are not oncogenic.

→ We observed baseline growth of NMuMG cells in our 3D soft-agar assay, even for the GFP- and full-length (FL) *Fgfr2*-expressing control cells. We also observed (statistically insignificant) increases in clonogenicity in cells expressing *Fgfr2*^{FL} fused to partners with dimerizing potential, such as *Fgfr2*^{FL}-*Bicc1* or *Fgfr2*^{FL}-*Tacc2* (Extended Data Fig. 8). Likewise, we saw elevated FGFR2 signalling and slightly decreased tumour latencies mediated by *Fgfr2*^{FL}-*Bicc1* and *Fgfr2*^{FL}-*Tacc2* as compared to *Fgfr2*^{FL} alone (Fig. 2b, c, Extended Data Figs. 7, 10). However, compared to these modest changes, E18-truncation of *Fgfr2* appeared to be the most relevant determinant significantly and strongly potentiating FGFR2 oncogenicity, as evidently observed *in vitro* in signal inductions and gains in clonogenicity and *in vivo* in clonal expansion of the mammary epithelium resulting in vastly decreased tumour latencies across *Fgfr2*^{ΔE18} variants and fusions (Fig. 2b-e, Extended Data Figs. 7-10). Notably, depletion of BICC1's SAM oligomerisation domain had no impact on the oncogenicity of *Fgfr2*^{ΔE18}-*Bicc1* (Fig. 2c, Extended Data Figs. 7, 9, 10a). Thus, dimerization of FGFR2^{FL} mediated by fusion partners can artificially enhance oncogenicity of full-length FGFR2 (these fusions are controls used in this manuscript and have not been observed in human cancers), but the oncogenicity of *Fgfr2*^{ΔE18} fusions (representing clinical *FGFR2* fusion oncogenes) does neither depend on the fusion partners nor their oligomerisation potential.

In extended Figure 7 it appears that the Fgfr2FL-Bicc1 is expressed at lower levels than the corresponding Fgfr2ΔE18-Bicc1 construct, and this complicates assessment. I recommend that the authors either address this point, or substantially soften conclusions. Figure parts A-D are essentially similar data, and I do not think that all of these parts are needed.

→ In all our *in vitro* lentiviral constructs, *Fgfr2* cDNA sequences were cloned in combination with a 3'-T2A-Puro sequence allowing puromycin selection of infected cells. For all *in vitro* experiments, we continuously grew cells with 1 μg/ml puromycin to ensure maintained expression of our *Fgfr2*-T2A-Puro variants. We carefully assessed expression of all *Fgfr2* constructs used, both on RNA and protein level. We measured expression of *Fgfr2* itself but especially also of the T2A-Puro sequence using RT-qPCR, allowing sensible comparison of mRNA expression across our *Fgfr2* constructs (Extended Data Fig. 6a). We also used FACS to quantify FGFR2 protein levels across all our *Fgfr2* constructs and cell line replicates (Extended Data Fig. 6b). Notably, the constructs were expressed at comparable RNA and protein levels, with some expression differences among cell line replicates. To compensate for differences observed for FGFR2 expression on Western blots (*Fgfr2*^{FL}-*Bicc1* showed lower FGFR2 expression, and so did *Fgfr2*^{ΔE18}) possibly arising due to the cell starvation conditions used or differences in between cell line replicates (see differences among replicates in FACS FGFR2 expression data), we normalised downstream phosphorylation signals to FGFR2 protein levels (Extended Data Fig. 7h, i).

We reorganised former Figure 3, with its *in vivo* data now being displayed in revised Figure 2. The revised *in vitro* data described above is displayed in Extended Data Figures 6-8.

Figures 3E-F are a highly interesting demonstration that exon 18 deletion is required (and this reduces the concern in interpretation of figure 3A). As a control, to interpret this experiment the authors must demonstrate that the Fgfr2FL-Bicc1 (and other full-length constructs) successfully transfect and express. Figure G is very nice data that demonstrates that tumours expressing this variant are sensitive to FGFRi.

→ We are happy to read that the referee acknowledges our somatic *in vivo* modelling efforts demonstrating the requirement of *Fgfr2* E18-truncation for the oncogenicity and the actionability of these truncation variants. To address the referee's concerns regarding equal transduction and FGFR2 expression efficiencies among *Fgfr2* variants used *in vivo*, we took advantage of *Fgfr2*-P2A-Cre lentiviral constructs and *mT/mG* female mice, where Cre-activity mediates *mT/mG* allele-switching resulting in EGFP-positive cells. FACS analyses of mammary glands one week post intraductal injection of *Fgfr2*-P2A-Cre lentiviruses revealed mammary epithelial cells to be switched at comparable frequencies and EGFP-traced cells to express FGFR2 comparably across *Fgfr2* variants tested (Extended Data Fig. 9). Notably, analysis of mammary glands three-week and six-week post injection showed clonal expansion of mammary epithelial cells transduced with *Fgfr2*^{ΔE18} variants. However, epithelial cells expressing *Fgfr2*^{FL} or *Fgfr2*^{FL}-*Bicc1* were gradually lost over time from the mammary epithelium, despite maintained FGFR2 expression. Furthermore, we used *Fgfr2*^{K422R} kinase-dead variants and

Fgfr2^{ΔE18}-*Bicc1*^{ΔSAM} to demonstrate that *in vivo* clonogenicity of *Fgfr2*^{ΔE18} and *Fgfr2*^{ΔE18}-*Bicc1* depends on a functional kinase domain of FGFR2 but not on the SAM oligomerisation domain of BICC1 (Extended Data Fig. 9).

Figure 4A. This data is not novel, there is substantial prior data demonstrating expression of truncated FGFR2 variants in amplified cancers, and that some of these cancers have an intron 17 rearrangement, that is amplified.

→ We agree that expression of truncation variants and/or fusions in some of the cell lines has been described before, which we acknowledge in the revised manuscript in lines 217-218, “Based on these correlations, we obtained several human cancer cell lines expressing amplified *FGFR2* variants^{18,19,22,40,41}”. However, our comprehensive analysis of CNA-seq and RNA-seq data from all cell lines enabled unambiguous identification of all *FGFR2* variants expressed, including *FGFR2* rearrangements (REs) as well as E18-splicing isoforms. Analysis of paired-end RNA-seq data also allowed quantification of *FGFR2* I17-spanning reads to canonical E18-C1 versus non-canonical E18 and/or RE partners. These data were also central to design specific siRNAs targeting individual *Fgfr2* variants, as used in former Figure 4b, c. All of these data are now displayed in Extended Data Figures 18 and 19.

Figure 4B. I believe that KATO-III cells have amplified an intron 17 rearrangement, and thus express predominantly the C3 variant. As such this experiment cannot be interpreted as the authors suggest. It is probable that KATO-III cells express more of the *Fgfr2*^{ΔE18}-C3 variant than *Fgfr2*^{ΔE18}-C1. Therefore, siRNA depletion of the *Fgfr2*^{ΔE18}-C3 variant may cause the majority of the total *Fgfr2* in the cell to be depleted. Conversely silencing of the C1 variant may have a relatively minor effect on total FGFR2. Therefore, this experiment cannot be used to support the relative oncogenicity of C1 vs C3 variants. Only a cell line with equal, or lower, protein expression of C3 could be used to support the hypothesis.

Extended data figure 13B appears to indicate that depletion of the *Fgfr2*^{ΔE18}-C3 variant leads to a less dramatic decrease in cell density when compared to depletion of the *Fgfr2*^{ΔE18}-C1 variant in other cell lines. How do the expression levels of these two variants compare in these cell lines at the protein level?

I recommend that unless the authors can address my concerns that figure parts 4A-C are moved to the appendix, and conclusions from the data substantially softened.

→ We appreciate the concerns raised by the referee. To quantify expression of the different *FGFR2* variants, we quantified all spanning reads from *FGFR2*-E17 in cell lines containing *FGFR2* amplifications (Extended Data Fig. 18b, c). Indeed, in KATO-III cells, *FGFR2*^{E18-C3} was among the dominant *FGFR2* species expressed. However, in several other cell lines with *FGFR2* amplifications (SNU-16, SUM52PE, MFM-223), the contribution of E18-truncated *FGFR2* to the complete *FGFR2* expression spectrum was not as dominant as in KATO-III (Extended Data Fig. 18b, c). Nevertheless, in all of these cell lines, silencing of E18-truncating *FGFR2* variants using variant-specific siRNAs compromised growth of the cells, while silencing of full-length *FGFR2*^{E18-C1} had minor impacts on cell growth (Extended Data Fig. 19a-c).

To functionally decipher dependence on different *FGFR2* isoforms in KATO-III cells, we performed rescue experiments using human full-length *FGFR2*^{E18-C1} and E18-truncated *FGFR2*^{E18-C3} cDNA constructs in KATO-III cells transfected with the different siRNAs (Extended Data Fig. 19d, e). *FGFR2*^{E18-C3} was sufficient to rescue the silencing effect of any endogenous *FGFR2* isoforms. In contrast, *FGFR2*^{E18-C1} proved insufficient to rescue *FGFR2*^{E18-C3}-silencing (Extended Data Fig. 19f, g). Thus, KATO-III cells ultimately depend on oncogenic, E18-truncated *FGFR2*^{E18-C3}.

Figure 4D/E: Compelling evidence that patient derived models with c terminal truncations are sensitive to FGFR inhibition.

→ We appreciate that the referee values the results from our preclinical intervention study using PDX models.

Figure 5: This figure provides compelling evidence that c terminal truncations without fusion partners are clinically actionable. It is important to point out that this study does illustrate the clear prior understanding that c terminal truncations without fusion partners would be oncogenic and targetable. In this registration study, the design of which clearly predates this manuscript by a number of years, patients with c terminal truncations without fusion partners were recruited into the same cohort as patients with FGFR2-fusions, reflecting the prior understanding that these patients would have a similar sensitivity to therapy.

Nonetheless, this data clearly adds strong support to the highly interesting pre-clinical data. The authors must confirm that the data in this figure has not been published before in prior publications on this trial, including in the very recently published manuscript in Cancer Discovery “Clinicogenomic Analysis of FGFR2-Rearranged Cholangiocarcinoma Identifies Correlates of Response and Mechanisms of Resistance to Pemigatinib”.

→ We appreciate that the referee acknowledges the value of the clinical data presented in Figure 5. We would like to emphasise that the FIGHT-202 clinical trial was designed without any direct evidence that C-terminal *FGFR2* truncations without fusion partners are clinically actionable. Given the rarity of *FGFR2* REs, the sponsor of FIGHT-202 chose to be inclusive with respect to all *FGFR2* REs identified by FoundationOne, regardless of classification. Since the FoundationOne assay can only identify the DNA breakpoint and not the product, the sponsor considered the possibility that these REs may have represented bridged fusions as well as truncations.

We confirm that the data in Figure 5 has not been published before in prior publications on the FIGHT-202 trial (Abou-Alfa et al. 2020; Silverman et al. 2021). The re-analysis of FIGHT-202 presented in this manuscript is novel because we are using different RE classification than was used in the original study. The original study used definitions from Foundation Medicine, which classify REs as fusions, even if they are predicted to be out-of-frame (if the partner gene is known). In contrast, we classified *FGFR2* fusions versus REs solely based on the predicted reading frame of the fusion partner. Importantly, in the original analysis, FGFR2-N/A REs were also not analysed separately from other *FGFR2* fusions and REs.

Figure 5C: the required statistics have not been carried out.

→ We now carried out the statistical analysis as requested (Fig. 5c). We would like to indicate that this is a post-hoc analysis and that the original study was not designed to make comparisons between the presented cohorts.

Referee #2 (Remarks to the Author):

The authors previously reported that *Fgfr2* is the most frequently affected gene in mouse mammary tumors with conditional E-cadherin (*Cdh1*) inactivation and Sleeping Beauty (SB) insertion.

In this study, the authors re-discovered that preferential SB insertions within intron 17 of the mouse *Fgfr2* gene give rise to upregulation of C-terminally truncated *Fgfr2* because of exon 18 (E18) deficit through transcriptional truncation within the SB transposon. They demonstrated that E18-defective *Fgfr2* variants promote mammary tumorigenesis in conditional *Cdh1*^{+/-} or *Cdh1*^{-/-} mice. They found that E18-defective FGFR2 variants also occur in a variety of human cancers, such as biliary tract, breast, gastric, lung and ovarian cancers, owing to in-frame fusions, intragenic rearrangements and point mutations. They then demonstrated that mouse mammary tumors induced by E18-defective *Fgfr2* variants are sensitive to FGFR inhibitors irrespective of presence or absence of C-terminal fusions.

Results presented in this manuscript are of great interest and value for diverse groups of researchers.

→ We appreciate that referee #2 considers our presented results “of great interest and value” to a broad spectrum of researchers.

In contrast, the following points were noticed as issues to be further addressed:

(1) Functional genomics of E18-defective *Fgfr2* variants. It remains unclear how E18-defective *Fgfr2* variants induce malignant phenotypes. Because E18 of *Fgfr2* consists of the C-terminal coding region and 3'-untranslated region (UTR), C-terminal deletion of *Fgfr2* protein and 3'-UTR deletion of *Fgfr2* mRNA might be involved in enhanced tumorigenesis. In addition to structural variations, nonsense point mutations cause C-terminal truncation of human FGFR2 protein (Extended Data Figure 3). The authors need to investigate the effects of (i) C-terminal nonsense mutations in E18 of mouse *Fgfr2* and/or (ii) 3'-UTR deletion from E18 of mouse *Fgfr2* to shed light on the precise mechanisms of malignant phenotypes driven by E18-defective *Fgfr2* variants.

→ Both in our GEMMs where we introduced *Fgfr2* variants into the *Coll1a1* locus and in our *in vitro* and *in vivo* models where we employed *Fgfr2* lentiviral constructs, we used *Fgfr2* cDNA sequences consisting only of the protein-coding sequence lacking the 3'-UTR. Since *Fgfr2*^{FL} constructs were virtually nontumorigenic, despite the lack of the 3'-UTR, we can rule out the 3'-UTR as a relevant suppressor of *Fgfr2* oncogenicity.

To gain insight into the tumour-suppressive nature of the C-terminus itself, we generated a series of *Fgfr2* variants where we gradually truncated the C-terminus. The design of the *Fgfr2* truncation variants was guided by the observed E18-truncating nonsense mutations observed in human cancers. In addition, we generated several representative frameshift mutations by including the resulting sequences coding for non-canonical C-termini (Extended Data Fig. 2k). *In vitro* clonogenicity and *in vivo* oncogenicity of FGFR2 increased with gradual shortening of the C-terminus, with a considerable decrease in tumour-onset observed when deleting the distal part of the C-terminus (*Fgfr2*^{V702*}) resembling the oncogenicity of the *Fgfr2*^{E18-C2} isoform, which encodes a C-terminus matching the proximal part of the canonical FGFR2 C-terminus (Fig. 2d, e, Extended Data Figs. 2k, 8, 10b, e-g). However, oncogenicity comparable to *Fgfr2*^{ΔE18} was only achieved when also considerable parts of the proximal C-terminus were deleted (*Fgfr2*^{P686*}, *Fgfr2*^{T678*}, *Fgfr2*^{Y674*}, *Fgfr2*^{L681fs*6}; Fig. 2e, Extended Data Figs. 8, 10e-g). These data coincide with the marked enrichment of both nonsense and frameshift mutations in the most proximal part of the C-terminus (Extended Data Fig. 2k). Based on our functional *in vitro* and *in vivo* classifications, we grouped nonsense and frameshift mutations observed in human cancers into proximal E18-truncating mutations (E768-Y783) causing E18-truncation phenotypes and distal mutations (P784-T821) causing no or only limited E18-truncation phenotypes (Fig. 2a).

(2) Functional proteomics of E18-defective *Fgfr2* variants. It also remains unclear how E18-defective *Fgfr2* variants elicit intracellular signaling dysregulation, although the authors showed western blot analyses of mouse tumors using anti-FGFR2 antibody and anti-Y653/Y654-phosphorylated FGFR antibody (Extended Data Figure 7b). They are advised to carry out phosphoproteome analyses on *Fgfr2* and its substrates in mouse tumors presented in Figure 2.

→ As advised by the referee, we carried out mass spectrometry-based expression proteomic, global phosphoproteomic (based on IMAC enrichment), and phospho-Tyr-specific proteomic (based on phospho-Tyr IP enrichment) analyses of mouse

mammary tumours driven by different *Fgfr2* variants. We took into consideration that the measured tumour-phosphoproteomes could be confounded by stromal signals as well as signals derived from additional driver mutations acquired during tumour development/progression, particularly in slow-growing tumours expressing low-oncogenic *Fgfr2* variants. Therefore, we ran in parallel phosphoproteomic analyses of serum-starved NMuMG cells expressing different *Fgfr2* variants. For the phosphoproteomic analyses presented in the revised manuscript, we teamed up with the lab of Prof. Connie Jimenez (OncoProteomics Laboratory, Amsterdam UMC), a long-standing expert in oncoproteomics.

The *in vitro* analyses revealed *Fgfr2*^{ΔE18} and *Fgfr2*^{ΔE18}-*Bicc1* to strongly induce FGFR2 downstream signalling compared to *Fgfr2*^{FL}, as evidenced by the phospho-enrichment of targets of the MAPK and AKT/mTOR signalling pathways. Interestingly, *Fgfr2*^{FL} showed no signalling activity in starvation conditions, whereas *Fgfr2*^{FL}-*Bicc1* activated MAPK and AKT/mTOR pathways, albeit to a much lesser extent than *Fgfr2*^{ΔE18} variants (Fig. 2b, Extended Data Fig. 7). We were also able to measure phosphoproteomic enrichments in mammary tumours driven by *Fgfr2*^{FL} and *Fgfr2*^{ΔE18} variants (Extended Data Fig. 13a, b). The phosphoproteome of *Fgfr2*-driven tumours was overall distinct from mouse mammary tumours induced by loss of *Brcal* and *Trp53* (Extended Data Fig. 13c). Of note, we observed phospho-enrichments of MAPK and AKT/mTOR signalling pathway targets in tumours driven by *Fgfr2*^{ΔE18} variants (Extended Data Fig. 13d, e), in agreement with the *in vitro* experiment.

(3) Fusion partners of FGFR2. The authors illustrated several fusion partners of FGFR2 in Extended Data Figure 3g; however, entire FGFR2 “fusionome” remains to be displayed. To increase the value of this manuscript for readers, the authors need to list up all binding partners of in-frame FGFR2 fusions depicted in Figure 2b and 2c.

→ As suggested by the referee, we listed all the fusion partners identified in the different datasets in Extended Data Tables 1-5 accompanying the revised manuscript.

(4) Clinical genomics of E18-defective FGFR2 variants. Some cases with FGFR2 C-terminal recombination are accompanied by FGFR2 gene amplification (Figure 2). E18-defective FGFR2 variants could be generated through breakage-fusion-bridge cycles during FGFR2 gene amplification, which might elicit intra-tumor or intra-patient heterogeneity and affect the response of cancer patients to FGFR2-targeted therapeutics. The authors are advised to investigate whether response rates of FGFR inhibitors differ between cases of FGFR2 C-terminal recombination with FGFR2 gene amplification and those without FGFR2 gene amplification.

→ Indeed, a fraction of the FIGHT-202 patients with FGFR2 fusions/REs also harboured *FGFR2* amplifications ($n = 5$). Taking the small number of patients in cohort ‘*FGFR2* in-frame fusion + amp’ into consideration, the Kaplan-Meier curves did not show a significant response difference between patient cohorts with and without *FGFR2* amplification (Fig. 5c). Although the number of patients is small, these data suggested that, among cholangiocarcinoma patients with *FGFR2* fusions, *FGFR2* gene amplification does not have a marked impact on response to FGFR2 inhibitors.

Referee #3 (Remarks to the Author):

The increasing availability of tumor molecular profiling by next-generation sequencing provides the opportunity to tailor therapies to potential targets. The manuscript by Zingg et al. “Truncated FGFR2 is a clinically actionable oncogene in multiple cancers” describes the discovery and characterization of exon18-truncated FGFR2 as an oncogenic driver and a potential biomarker/target for targeted therapies with FGFR inhibitors in cancers. The authors performed a number of functional analyses using engineered expression constructs in 2d, 3d cultures, and pdx models to establish FGFR2-e18 truncation as a novel oncogenic aberration that gives proliferative advantage to cells in culture, and imparts sensitivity to FGFR inhibitors, similar to cancer cells harboring FGFR2 gene fusions.

Overall, this study provided sufficient evidence in supporting the pathogenicity of E18-truncated-FGFR2 in cancer and demonstrated drug sensitivity of this type of FGFR2 aberrations towards FGFR inhibitors. However, while the discovery of 3' truncation events as potential driver using SB screen is a good starting point, the analysis and description of the FGFR2 aberrations in cancer cohorts needs much improvement. In addition, additional controls for the functional analyses and drug treatment experiments specified below may help make a more convincing case.

- We are happy to read that referee #3 considers the study to provide “sufficient evidence” to support the major conclusions of this manuscript, namely that *FGFR2* E18-truncation is an oncogenic driver alteration and that patients harbouring these alterations are sensitive to FGFR2-targeting therapeutics. We clarified the descriptions of different *FGFR2* aberrations in the main text and the figure legends. We also included the suggested and appropriate controls for the functional experiments, as delineated below.

Major concerns:

1. The strength of the paper is the demonstration through a wide-ranging set of experiments that truncation of FGFR2 contributes to the oncogenicity of FGFR2, in cases where this aberration is found. The cataloging of cases and the recognition that this type of event is not usually tested for is an important contribution to precision oncology. The experiments are well done and clearly presented. They add significant further evidence that truncation of FGFR2 plays a role in oncogenesis and correctly cite Szybowska as demonstrating the critical role of p.S780 in the regulation of FGFR2. Szybowska also pointed out that truncation is an oncogenic mechanism, and identified clinical cases with truncation events. The authors should cite them in that context.

- We appreciate that the referee acknowledges the relevance of our findings for the field of precision oncology and that the experiments are performed well and add significant evidence to support the above-mentioned major conclusions of the manuscript. In the revised manuscript, we now cite (Szybowska et al. 2019) in the appropriate context, which is in lines 99-101, “... and protein-truncating mutations significantly enriched in the coding sequence (CDS) of *FGFR2*-E18 (Fig. 2a, Extended Data Fig. 2i-k), in line with described E18 nonsense mutations³⁰”.

2. The manuscript, however, suffers from a central and repeated flaw. The authors oversimplify the complexity of FGFR2 activation, genetic aberrations of all types, and clinical responses to a small number of inhibitors in order to claim that truncation of FGFR2 plays a “central” role in FGFR2 oncogenicity. As well established by numerous groups and publications over several decades, oncogenicity of receptor tyrosine kinases is a complex system, with amplification, expression, mutations (both missense and indels), gene fusions, alterations in ligand and interacting proteins all playing roles to different degrees in different cases. The authors’ own transposon data demonstrate that expression plays an important role, as well as most in vitro experiments couple overexpression with a genetic aberration of a second type. The genetic evidence on FGFR2 is quite substantial and clearly indicate amplification, in-frame fusions, and mutations both in the kinase domain and extracellular domains play functional roles in the development of several human cancer types. This role in oncogenesis should not be confused with clinical response. For example, point mutations in tyrosine kinases are often quite unpredictable with regard to clinical response to an inhibitor. The point mutations in FGFR2 do not seem to respond clinically to small molecule inhibitors against the kinase domain as well as other aberrations. More work needs to be done to answer why. But no conclusions should be drawn as to ranking the relative role of this aberration vs. another. By occurrence alone, recurrent missense mutations could be judged to be “most” important, but again this is not a proper conclusion from the data. The paper needs to be revised with the recognition that all the aberration classes of FGFR2 play a role, perhaps interacting and synergistic in some cases. The use of clinical response data is very important in the selection of cases containing aberration classes most likely to respond, but should not be used to conclude

whether any aberration class is more “central” to oncogenesis. The authors have repeatedly presented a skewed interpretation of the role of fusions in *FGFR2* oncogenicity.

→ We appreciate that the referee highlights these points. We agree that oncogenicity of a given alteration does not necessarily translate into targeted inhibitor sensitivity. Therefore, we implemented a clear distinction between oncogenic potential versus clinical response rates of *FGFR2* alteration types in our revised manuscript. We separated tumour formation and *FGFR* inhibitor intervention data into separate Figures and accompanying Extended Data Figures (Figs. 2, 3, Extended Data Figs. 6-15 versus Figs. 4, 5, Extended Data Figs. 16-20). We also rewrote the main text and especially the discussion paragraph of the revised manuscript, distinctly discussing the oncogenicity versus inhibitor sensitivity data (lines 262-277).

We also agree that *FGFR2* E18-truncations should not be viewed as “superior” or “more central” to oncogenesis than all other *FGFR2* alterations. Nevertheless, our *in vitro* and *in vivo* functional classification data clearly demonstrate that E18-truncated *Fgfr2* alone is a very strong tumour driver, whereas amplification-mimicking overexpression of *Fgfr2*^{FL} displayed no enhanced clonogenicity *in vitro* and only marginal oncogenicity *in vivo*. Moreover, fusion partners did not further enhance oncogenicity of *Fgfr2*^{ΔE18}.

We agree with the referee that, dependent on the tissue of origin and/or the mutational context, any *FGFR2* alteration (or combinations of synergizing *FGFR2* alterations) might contribute to tumorigenesis. Indeed, our extended oncogenomic analyses within this revision indicates that large cohorts of cancers harbour full-length *FGFR2* amplifications or missense hotspot mutations in a mutually exclusive manner to E18-truncation (Fig. 3a, Extended Data Fig. 14a). Moreover, *FGFR2* E18-truncations, amplifications, and hotspot mutations were differentially recurrent dependent on the tumour type (Extended Data Fig. 14d). Importantly, analysis of co-occurring driver genes in the three *FGFR2* alteration cohorts revealed distinct sets of driver genes that co-occurred or were mutually exclusive with the three *FGFR2* alteration groups (Fig. 3a-c, Extended Data Fig. 14). This information was used for extensive *in vivo* tumour modelling demonstrating that *Fgfr2*^{FL} alone is marginally oncogenic but can act as a potent tumour driver in specific oncogenomic contexts (Fig. 3d-f, Extended Data Fig. 15), as implied by the referee. These novel findings are an important addition to the revised manuscript, and more details are delineated below.

A) Their own data illustrates in-frame fusions (and frame undetermined, which may very well be complex yet in-frame) are more prevalent than intergenic breakpoints.

→ We agree with the referee that in the datasets we analysed, in-frame fusions were more prevalent than non-canonical REs, particularly REs with intergenic regions (Fig. 2a, Extended Data Fig. 2b, c, h, l). Transcripts terminating in-frame in coding genes are likely to be less susceptible to nonsense-mediated decay, which might generate a selection bias towards REs generating in-frame fusions. It also remains possible that specific fusion partners might further enhance oncogenicity of E18-truncated *FGFR2*, which might depend on the tissue of origin and/or the mutational context in a particular tumour. However, our *in vivo* data show that production of C-terminally truncated *FGFR2* is the dominant oncogenic trigger.

B) The fusions are recurrent by gene (a large set with some partner genes highly recurrent) and more importantly, the fusions are functionally recurrent in the sense that nearly all if not all 3' partner genes to date retain a known dimerization or oligomerization domain in the fusion protein. The number of genes is large (other *FGFRs* have fusions with the same properties) and increasing. The overall dataset highlights the decidedly nonrandom attributes of the 3' partner genes.

→ We agree that some of the fusion partners showed recurrence, such as *BICC1*. To properly quantify dimerising capacity and recurrence of fusion partners, we first classified all unique proteins encoded by fusion partner genes identified in the Foundation Medicine cohort ($n = 337$) into self-interacting versus non-self-interacting using domain-domain interaction information from the SLIPPER database (Liu et al. 2013). About 18% of all fusion partners had self-interacting potential, which was a significant enrichment over the self-interaction rate across the human proteome (Extended Data Fig. 2e). Using the SLIPPER algorithm, we likewise showed an enrichment of proteins with high dimerising capacity among the fusion partners versus the proteome (Extended Data Fig. 2f), and also specific oligomerisation/dimerization domains were enriched in fusion partners versus the proteome (Extended Data Fig. 2g). We then rank-ordered self-interacting versus non-interacting fusion partners according to their recurrence. Indeed, among the most recurrent fusion partners were dimerising candidates, such as *BICC1* and *TACC2*. Together, these made up 57.2% of all *FGFR2* E18-truncating REs identified. However, 19.8% of REs also made use of non-dimerising fusion partners, and 23% of all REs made use of non-coding

intergenic regions (Extended Data Fig. 2h, Extended Data Table 2). In summary, the oncogenomic data indicate a preferential selection for dimerising fusion partners, as suggested by the referee.

To directly assess whether the SAM oligomerisation domain of BICC1 is required for FGFR2-BICC1 oncogenicity, we tested *Fgfr2^{ΔE18}-Bicc1* versus *Fgfr2^{ΔE18}-Bicc1^{ΔSAM}* in our *in vitro* and *in vivo* assays. Depletion of the SAM domain had no impact on *Fgfr2^{ΔE18}-Bicc1* oncogenicity, which was as high as observed for *Fgfr2^{ΔE18}* devoid of any dimerization partner (Fig. 2c, Extended Data Figs. 7h, i, 8, 9, 10a). Nevertheless, as mentioned above, this does not exclude the possibility that in specific cellular or mutational contexts, fusion partners may contribute to oncogenicity of *FGFR2* fusions.

C) Breakpoints in intergenic space are enriched in amplification cases relative to in-frame fusions, highlighting that in-frame fusions in particular are not byproducts of amplification.

→ We agree with the referee that REs and in particular in-frame fusions do not necessarily have to be products of gene amplification. We adapted the text to further clarify that amplifications can be among the many ways by which REs can occur. In any case, we quantified relative enrichments of *FGFR2* RE types (in-frame fusions versus other types of REs) in samples with *FGFR2* amplifications versus without *FGFR2* amplifications. Indeed, REs with intergenic space were more frequent in cases with *FGFR2* amplifications, while in-frame fusions were more frequent in cases without amplified *FGFR2* (Extended Data Fig. 2b).

D) The discussion concerning measured clinical benefit relative to genetic aberration type is convoluted at best and somewhat misleading. First, there is no specific evidence presented to date concerning non-fusion RE's vs. any other aberration type for clinical response. The current consensus data does indicate fusions may respond better than missense mutations and amplification. The statement, "Notably, cholangiocarcinoma patients with *FGFR2*-*BICC1* fusions versus other fusion and RE partners equally responded to *FGFR*-targeted therapy, despite the potent SAM oligomerization domain in *BICC1*." is misleading. The distinction of *FGFR2*-*BICC1* fusion from other fusions with oligomerization domains does not lead to supportable conclusions concerning the relative role of fusions vs. any other category of aberration. Would the authors please elaborate on what measure of "potency" of oligomerization domain they are referring to?

→ We agree with the referee that this discussion paragraph was rather convoluted and needed improvement. The revised discussion paragraph (lines 262-277) now emphasises that distinct mutational contexts can be relevant for differential response rates among *FGFR2* alteration categories.

E) The statement, "In line with these clinical observations, we uncovered truncation of the C-terminus to be central to *FGFR2* oncogenicity." is not a proper statement. Clinical responses to an inhibitor and oncogenicity are quite distinct measures and concepts, they should not be linked in this way.

→ We agree with the referee and removed the above-mentioned sentence from the manuscript. We carefully revised the manuscript to ensure adequate discrimination between *FGFR2* oncogenicity and inhibitor sensitivity.

F) The remaining discussion should be modified with a more complete and less simplistic view of all *FGFR2* aberrations and their roles in oncogenesis and the separate issue of potential clinical benefit. Perhaps a more accurate final statement would be "Instead of considering patients based on *FGFR2* fusion, amplification, or mutation status alone, our data suggest that identifying cancers with SVs or mutations that result in expression of *FGFR2* E18-truncating variants will be an important additional biomarker for *FGFR*-targeted therapeutics and may significantly expand the number of cancer patients who may benefit from such therapy." Such a statement would both make the author's point of the importance of detecting this category of mutation, but not distort the larger view of all *FGFR2* aberrations in precision oncology.

→ As the reviewer suggests, we adapted the discussion towards a more complete view of *FGFR2* aberration types, and put these into perspective to our identified E18-truncation alterations.

3. Figure 2a: the linear map of FGFR2 breakpoints in pan-cancer WGS data- please use color code to highlight truncating mutations/ breaks considered activating. Also include a parallel track showing the kinase domain and other functional domains to facilitate easy interpretation of domains affected by the different aberrations. Here, it will also help to include a track of typical breakpoints involved in FGFR2 gene fusions.

→ In the revised panel, which is now displayed in Figure 1f, we added a track for protein domain information. The intention of the whole-genome sequencing (WGS) analysis was to identify enrichments of breakpoints in an unbiased manner. Therefore, we did not include additional tracks on previously published data to the panel. Instead, we referred to previously identified *FGFR2* fusion breakpoints in the revised manuscript text. Specifically, we included the following sentence in lines 79-80: "... revealed a significant enrichment of RE breakpoints in I17 (Fig. 1f, Extended Data Fig. 1f), coinciding with reported *FGFR2* fusion breakpoints²⁴⁻²⁷". We also did not include color-coding for activating REs, as this classification was only introduced based on the functional data in Figure 2 and associated Extended Data Figures 6-13.

4. Figure 2b: FGFR2 breakpoint locations are displayed in three categories- 5' to I17, I17, and E18. Arguably, only breaks within I17, and E18 are proposed to be activating, while breaks or deleterious mutations within 5' to I17 would presumably be deleterious (for example, the kinase domain within Exon10 to Exon17, and more upstream regions encoding the receptor etc). The authors could declutter the figures by focusing on only the functionally relevant aberrations- not showing everything that the data analysis identified. Also, the cases with breakpoints in I17/ E18 seem far fewer in Figure 2a as compared to 2b- both representing the same HMF cohort. Please explain and reconcile the numbers as appropriate.

→ The purpose of former Figure 2b was to present an unbiased overview for all the *FGFR2* REs identified in the WGS dataset from the Hartwig Medical Foundation. This panel has now been moved to Extended Data Figure 1h. We rearranged the panel to allow the reader to quickly grasp the distinction between I17/E18 REs and the remaining 5'-to-I17 REs. The linear BP map (Fig. 1f) shows an unbiased enrichment of all the *FGFR2* BPs discovered in the HMF cohort ($n = 196$ BPs in 86 tumour samples). A few of these *FGFR2* BPs were excluded because they resulted in unresolved types of REs, as described in detail in the methods paragraph 'Analysis of WGS data from HMF', lines 717-750. Several samples also contained multiple BPs in *FGFR2*. This led to a reduced number of unique cases harbouring one or multiple *FGFR2* REs ($n = 55$ tumours), as depicted in Extended Data Figure 1g, h.

5. In Figure 2c, the track of "amplifications" distinguishes cases with amplification of E1-E17 (excluding E18)- the presumed activating variant, from those with full gene amplification (E1 to E18). However, Figure 2b (HMF cohort), and Figure 2e (TCGA cohort) do not show the amplification of the E18 excluding E1-E17 variant. This seems odd as a bona fide drive event would be expected to be seen across these cohorts as well.

→ We thank the referee for spotting the lack of *FGFR2* E1-E17 amplification variant-calling in the HMF and TCGA datasets. We now distinguished between full-length *FGFR2* amplifications and E1-E17 partial amplifications also in the HMF and TCGA datasets (Extended Data Figs. 1g, h, 5c, d). Methodological details were included in the associated methods paragraphs 'Analysis of WGS data from HMF' (lines 717-750) and 'Analysis of RNA-seq data from TCGA' (lines 765-794).

6. The prevalence of "functional" FGFR2-RE" in human cancer is not clearly estimated. For example, 86 out of 2,112 HMF-WGS cases carried breakpoints (4.1%) in the FGFR2 gene. These breakpoints distributed throughout the entire FGFR2, but only alterations that retained an intact tyrosine kinase domain are potential oncogenic events. Likewise, number of cases with FGFR2-REs in the FMI cohort (249,570 cases) varies in different places: Figure 2c, $n=1,307$, Extended Data Figure 3b, $n=2,245$. Again, how many of these events are predicted to be pathogenic and targetable by FGFR inhibitors are not explicitly described. It would be helpful to provide a total of likely actionable FGFR2-positive cases for each cohort investigated.

→ We apologize that some of the numbers were not clearly explained. We improved this throughout the revised manuscript and figures. We also included frequency plots of all E18-truncating alteration types found in the FMI cohort (Extended Data Fig. 2l) and TCGA cohort (Extended Data Fig. 5e) in the revised manuscript. Moreover, we generated a frequency

plot displaying FGFR2 E18-truncations, amplifications, and missense hotspot mutations found pan-cancer as well as in specific cancer types (Extended Data Fig. 14d).

7. Further, in the cases with FGFR2 amplification (E1 to E17)- the associated REs are presumably incidental and noisy, resulting from bridge fusion bridge cycles, not balanced translocation events that characterize more typical, recurrent gene fusions. I would suggest the removal those amplification- associated REs from the data tracks to help focus in on more specific putative driver aberrations. Notably, a large subset of cases with amplification E1-E17 are mutually exclusive from other events (Figure 2c). It may be worthwhile reviewing the cases with both amplifications and REs- for allelic balance between the two events. For example, in a case with 20 copies of FGFR2, multiple non-recurrent REs with a few reads each likely represent noise, and may be disregarded. Incidentally, in Figure 2c (FMI cohort) and Figure 2e (TCGA cohort), the mutual exclusivity of cases with mutations in E18 with all other structural aberrations is very striking and credible.

→ We appreciate this valid suggestion by the referee. We computationally evaluated allelic complexities for tumour samples with *FGFR2* REs. To this end, we used ploidy and expression levels of individual REs derived from the HMF-WGS and TCGA RNA-seq data, respectively. For several cases with *FGFR2* full length or E1-E17 partial amplifications, we indeed observed multiple *FGFR2* REs. If we observed an I17/E18 RE ploidy level above 15% in case of HMF-WGS data (Extended Data Fig. 1g) or an E17-spanning read frequency above 15% in case of TCGA RNA-seq data (Extended Data Fig. 5c), we called these samples as *FGFR2* I17/E18 REs. Based on these cut-offs, a TCGA sample previously called *FGFR2* amplified and I17-rearranged, was now reclassified as *FGFR2* amplified only (TCGA-AQ-A04L; Extended Data Fig. 5c). Across the FMI dataset, already in the original manuscript similar cut-offs were used to distinguish between samples with imbalanced *FGFR2* E1-E17 amplifications accompanied by noisy REs and cases containing considerably enriched I17/E18 REs that were simultaneously amplified. Only considerably enriched I17/E18 REs called in *FGFR2* full-length or E1-E17 partial amplification cases were displayed, while low-level REs were excluded. Thus, no samples were removed or reclassified in the revised Figure 2a versus the former Figure 2c. The oncoplot (Fig. 2a) and all accompanying data panels (Extended Data Fig. 2) show per sample only the I17/E18 RE type called with the highest confidence.

8. Figure 2d circos plot is so busy it fails to convey anything specific. Per the comments above, please consider removing the REs associated with FGFR2 amplifications (as those are presumed noisy byproducts of amplification process, not specific driver events per se).

→ As the circos plot in former Figure 2d was indeed too busy to convey a specific message, we replaced it with a linear map of chromosome 10 depicting all intrachromosomal *FGFR2* REs identified. The thicknesses of the BP-connecting arcs are now proportional to the recurrence of the corresponding RE partners (Extended Data Fig. 2d).

9. Figure 2b, c, e- the gender track doesn't appear to serve any purpose since in the pan-cancer dataset the gender distribution is quite dispersed as may be expected. Unless the authors want to convey something specific through this track- it should be removed.

→ We removed the gender tracks throughout the figures.

10. Incidentally, querying for FGFR2 mutations in cBioportal, a curated set of 185 non-redundant studies (with a total of 48045 cancer samples), only 6 frameshift mutations were noted in Exon 18, not notably clustered there as may be expected in a locus that imparts selective advantage when mutated (for example, contrast this with c-terminal mutations in Notch1). This number seems much lower than the number of mutant cases shown in the three cohorts presented here (Figure 2). Can the authors please explain the reason for this?

→ We cross-compared frequencies of frameshift, nonsense, as well as the sum of truncating mutations affecting *FGFR2* called in cBioportal with the frequencies in the TCGA (part of cBioportal) and FMI datasets (see graphical representation below).

Indeed, the frequency of called mutations is higher in FMI than in cBioportal and TCGA. This difference is most likely due to the fact that FMI performs very deep targeted sequencing at 500X to 1,500X, with high coverage of exons and I17 of *FGFR2* using hybrid-capture probes. This results in an average read depth that is superior to the average read depth used in studies reported in cBioportal (e.g., TCGA whole-exome sequencing is at 100X), which likely explains the increased discovery rate of *FGFR2* mutations in FMI versus cBioportal datasets. With 249,570 samples, the FMI cohort is also the largest oncogenomics dataset to date, and five times larger than the total number of nonredundant samples in cBioportal. The large sample size and high sequencing depth of the FMI dataset allowed robust identification of a truncating mutation cluster in the proximal part of the *FGFR2* C-terminus, as displayed in Extended Data Figure 2i-k.

Also, in the functional assays with the panel of E18-truncated *FGFR2* variants (Figure 3), please also include some frameshift truncating mutation disrupting E18, to fully span the range of aberrations noted in tumor samples. This is important since, REs and gene fusions presumably stabilize the E18 excluded *FGFR2* gene, and it is very curious if the truncated *FGFR2* resulting from stopgain/ frameshift indels are stable as well. It will also be useful to include some hotspot activating mutation(s) such as N549H/D/K or N550H as positive controls for sensitivity to FGFR inhibitors as well.

→ As suggested by the referee and guided by E18-enrichments of truncating mutations in the FMI cohort, we generated 10 *Fgfr2* variants representing the most recurrent nonsense and frameshift truncating-mutations (Extended Data Fig. 2k). Moreover, we also generated four missense mutant *Fgfr2* variants (*Fgfr2*^{S156W}, *Fgfr2*^{C287R}, *Fgfr2*^{N454K}, *Fgfr2*^{K564E}) representing the most recurrent missense hotspots called in the FMI cohort (*FGFR2*^{S252W}, *FGFR2*^{C382R}, *FGFR2*^{N549K}, and *FGFR2*^{K659E}; Extended Data Fig. 2a), and in agreement with previously described *FGFR2* hotspot mutations (cBioportal and Nakamura *et al.*, 2021). We functionally analysed these variants in our *in vitro* and *in vivo* tumour modelling assays. *In vitro* clonogenicity and *in vivo* oncogenicity of *FGFR2* increased with gradual shortening of the C-terminus, with a considerable decrease in tumour-onset observed when deleting the distal part of the C-terminus (*Fgfr2*^{V702*}) resembling the oncogenicity of the *Fgfr2*^{E18-C2} isoform, which also encodes a C-terminus matching the proximal part of the canonical *FGFR2* C-terminus (Fig. 2d, e, Extended Data Figs. 2k, 8, 10b, e-g). However, full *Fgfr2*^{ΔE18} oncogenicity was only achieved when also considerable parts of the proximal C-terminus were deleted (*Fgfr2*^{P686*}, *Fgfr2*^{T678*}, *Fgfr2*^{Y674*}, *Fgfr2*^{L681fs*6}; Fig. 2e, Extended Data Figs. 8, 10e-g), coinciding with the marked enrichment of both nonsense and frameshift mutations in the most proximal part of the C-terminus (Extended Data Fig. 2k). Among the *Fgfr2* hotspot mutations tested, only *Fgfr2*^{C287R} was comparably oncogenic to *Fgfr2*^{ΔE18} (Extended Data Figs. 8, 10h, i). We also subjected all truncation and hotspot *Fgfr2* variants to FGFRi-sensitivity assays. Proximal C-terminal *FGFR2* truncation variants showed FGFRi sensitivity comparable to *Fgfr2*^{ΔE18}, while, except for *Fgfr2*^{C287R}, *Fgfr2* variants with hotspot mutations were less sensitive to FGFRi (Fig. 4a, Extended Data Fig. 16a).

11. Figure 3: Interestingly, *FGFR2*-FL constructs show no proliferative/ drug sensitivity effects. In view of this, how do the authors see tumors with *FGFR2* amplification (E1 to E18)- which are essentially wild type *FGFR2*, only expressed at much higher levels- seen in subsets of tumors across all pan-cancer cohorts analyzed here (Figure 2).

→ It is possible that in a fraction of tumours full-length *FGFR2* amplifications represent passenger rather than driver events. The amplification might be driven by another cancer driver gene residing in the amplified region and/or the amplicon might not translate into a considerable *FGFR2* upregulation. Given the strong oncogenicity of E18-truncated *FGFR2*, it is also possible that tumours with full-length *FGFR2* amplifications contain one or few copies of E18-truncated *FGFR2*. These might escape identification by standard-depth sequencing due to a very low variant allele frequency, yet be responsible for the oncogenic potential of the *FGFR2* amplicon. In support of this hypothesis, we depleted different *FGFR2* isoforms in cancers cell lines with high-level *FGFR2* amplifications, and only silencing of E18-truncated *FGFR2* transcripts suppressed cell growth, independently of the prevalence of E18-truncated isoforms (Extended Data Figs. 18c, 19). Finally, in certain

cancers harbouring full-length *FGFR2* amplifications, the co-occurring mutational landscape and the tissue of origin might contribute to the oncogenic capacity and/or the drug sensitivity of amplified *FGFR2*.

To examine the possibility that oncogenic activity of *FGFR2* amplification only becomes manifest in the context of specific co-occurring mutations in other cancer driver genes, we performed an in-depth analysis of FMI samples harbouring E18-truncations versus samples with full-length *FGFR2* amplifications versus samples with *FGFR2* missense hotspot mutations. Interestingly, these three categories were largely mutually exclusive and differentially recurrent across cancer types (Fig. 3a, Extended Data Fig. 14a, d). Next, we analysed co-occurring driver mutations in the three *FGFR2* alteration cohorts and found striking differences. Across cancers, the three *FGFR2* alteration groups showed co-occurrences and mutual exclusivities with distinct sets of driver genes (Extended Data Fig. 14a-c). In breast cancer specifically, samples with *FGFR2* amplifications contained significantly more frequently *TP53* hotspot mutations, *MYC* amplifications, *PTEN* loss-of-function alterations, and *CCND1* and *FGF3/4/19* co-amplifications than *FGFR2* E18-truncation breast cancers (Fig. 3a, b). *FGFR2* amplifications showed significant co-occurrence with *TP53*, *MYC*, and *PTEN* alterations, which we did not observe in *FGFR2* E18-truncation samples (Fig. 3c).

These data suggested that *FGFR2* amplifications might depend on a distinct driver gene context. To functionally test this, we modelled the observations from the human oncogenomic data *in vivo* using intraductal injection of lentiviruses and compound mutant mice. Specifically, we combined lentiviral *Fgfr2^{FL}* overexpression with (i) Cre-mediated deletion of *Trp53*, (ii) Cre-mediated deletion of *Trp53* and Cas9-mediated disruption of *Pten*, (iii) lentiviral *Myc* overexpression, (iv) lentiviral *Fgf3* overexpression, (v) lentiviral *Ccnd1* overexpression, (vi) lentiviral overexpression of *Fgf3* and *Ccnd1*. We also tested the according *Fgfr2^{ΔE18}* controls. This extensive *in vivo* modelling showed that oncogenic competence of full-length FGFR2 is context-specific and only becomes manifest in conjunction with either *Trp53* deletion, *Trp53* and *Pten* loss, *Myc* overexpression, or *Ccnd1* and/or *Fgf3* co-overexpression. Notably, in conjunction with some of these driver combinations, *Fgfr2^{FL}* became as oncogenic as the *Fgfr2^{ΔE18}* single driver alteration. The co-driver genes tested, however, had little effect on *Fgfr2^{ΔE18}* oncogenicity (Fig. 3d-f, Extended Data Fig. 15). Thus, the oncogenic competence of *FGFR2* amplifications likely depends on a specific set of co-driver alterations, while *FGFR2* E18-truncations appear to act more context-independently.

Can they model potential phenotype of FGFR2 amplification by overexpressing it under a strong promoter? Perhaps this merits analysis of the version of E18-c2 versus c3/c4 predominant in amplified FGFR2 E1-E18 cases.

→ We would like to point out that all mouse *Fgfr2* and human *FGFR2* variants are expressed from a strong SFFV promoter in our lentiviral constructs, which we used both *in vitro* and *in vivo*. *In vitro*, we observed a consistent 1,000-fold overexpression of all the *FGFR2* constructs, independent of full-length or E18-truncation variants (Extended Data Figs. 6a, 19d). Thus, we appropriately model high-level amplification of full-length *FGFR2*.

12. In transgenic mouse models, *fgfr2*-FL expression was never detected by IHC in any of the tested genetic conditions (Extended Data Figure 2). Even the low-grade lesions observed in *Wap-Cre;Cdh1F/+; Fgfr2^{FL}* and *Wap-Cre;Cdh1F/F;Fgfr2^{FL}* mammary glands were FGFR2-negative. This raised the concern of the integrity of the *fgfr*-FL construct.

→ This is a valid point of concern raised by the referee. To test the integrity of the *Fgfr2-IRES-Luc* alleles, we isolated mouse mammary epithelial cells (MMECs) from *Fgfr2^{FL}-IRES-Luc* and *Fgfr2^{ΔE18}-IRES-Luc* mice. *Ex vivo* activation of the alleles using Adeno-Cre led to comparable FGFR2 expression and luciferase activity in MMECs from both mouse strains (Extended Data Fig. 11a-e). Importantly, we also observed increased luciferase activity in mammary glands of *Wap-Cre;Cdh1^{F/F};Fgfr2^{FL}-IRES-Luc* mice over *Wap-Cre;Cdh1^{F/F}* control animals (background activity) at one month of age (Extended Data Fig. 11g), which is when *Wap-Cre* becomes active. At the first time point of measurement, *Wap-Cre;Cdh1^{F/F};Fgfr2^{ΔE18}-IRES-Luc* mice showed highly increased luciferase activity as compared to *Wap-Cre;Cdh1^{F/F};Fgfr2^{FL}-IRES-Luc* mice (Extended Data Fig. 11g). This was because induction of *Fgfr2^{ΔE18}* instantly promoted malignant expansion of the mammary epithelium, which became evident by the rapid emergence of palpable tumours shortly after one month of age (Extended Data Fig. 11h). Thus, both the *Fgfr2^{FL}-IRES-Luc* and the *Fgfr2^{ΔE18}-IRES-Luc* constructs were active at first. However, in contrast to the *Fgfr2^{ΔE18}* allele, *Fgfr2^{FL}* was not tumorigenic and therefore did not promote clonal expansion of recombined cells. Recombined cells were likely lost over time during mammary epithelium regeneration via neutral competition and/or the CAG promoter driving *Fgfr2* expression was silenced, as observed previously in genetically engineered mouse models using CAG-driven expression of non-tumorigenic transgenes

(Cornelissen et al. 2019). The FGFR2-negative mammary lesions observed in *Wap-Cre;Cdh1^{FL};Fgfr2^{FL}-IRES-Luc* and *Wap-Cre;Cdh1^{FL};Fgfr2^{FL}-IRES-Luc* mice thus likely represented spontaneous tumour events, which are known to occur in the FVB genetic background (Wakefield et al. 2003).

13. Figure 4a- all of the gene fusions/REs in various cancer cell lines are included in the data matrix; it would be more informative to show the fusions/RE breakpoints in a linear exon map showing kinase domain location so that we can correlate the cluster of breaks within I17/E18 affected by these events. It would be particularly interesting to see FGFR2-PLPP4, and FGFR2-ACADSB with their breakpoints indicated as 5' to I17, being sensitive to various TKIs and FGFR inhibitors tested. Since immediately upstream of I17 is the kinase domain on FGFR2 (up to exon10), presumably the breakpoint would not be too far upstream or there is some conceptual difficulty in making sense of the data.

→ FGFR2 breakpoints resulting in *FGFR2-PLPP4* and *FGFR2-ACADSB* were indeed 5' to I17, with the *FGFR2-ACADSB* BP being in *FGFR2-I5*, thus affecting the extracellular ligand-binding domain of FGFR2. To maintain consistency throughout the revised manuscript, we decided to focus only on E18-truncating *FGFR2* REs with BPs in the I17/E18 hotspot region. The E18-truncated *FGFR2* REs identified including E17-spanning read ratios and E18-C3 usage are displayed in Extended Data Figure 18b, c.

14. Figure 5, in the analysis of FIGHT202 cohort as well, please provide the coordinates of the breakpoints of FGFR2 in cases with “unknown frame”, “not in frame” variants on a linear map of FGFR2 gene- exon domain structure to allow assessment of the retained domains and whether those are recurrent positions, similar to the pan-cancer data cohorts (Figure 2).

→ In the FIGHT-202 trial, only patients with fusion/RE BPs in the I17/E18 hotspot region were included. Therefore, we think that in this case a linear exon map will provide limited information. We updated the manuscript text and the legend corresponding to Figure 5 to clearly describe the preselection process of patients.

15. AZD4547 is a pan-FGFR inhibitor that has been reported to target FGFR1, FGFR2 and FGFR3 kinase activity with IC₅₀ values of 0.2, 2.5 and 1.8 nM, respectively (Gavine et al. Cancer Research 2012). Although cell lines expressing E18-truncating FGFR3-REs also exhibited considerable AZD4547 sensitivity, it appears that cells with FGFR2-REs are more sensitive to AZD4547 (Extended Data Figure 12a). Are there data of other FGFR inhibitors in the CCLE database and do they show the same trend?

→ Unfortunately, the CCLE study did not include an additional FGFR inhibitor. However, the GDSC study from the Wellcome Sanger Institute contains response data for the FGFR inhibitor PD173074 (Iorio et al. 2016). Cell lines tested in the two datasets overlapped considerably. In the revised manuscript, we incorporated the PD173074 response values in our analyses, which are now displayed in Extended Data Figure 17. These analyses confirmed the referee's assertion. Cell lines containing E18-truncating *FGFR2* REs were indeed significantly more sensitive to FGFRi than cell lines containing E18-truncating *FGFR3* REs (Extended Data Fig. 17b, g).

16. Stop-gain mutations in FGFR2-E18 were detected in a fraction of tumors, but their impact in FGFR2 protein function, tumorigenicity, and drug sensitivity were not discussed or tested in this study. Could the authors expand on the occurrence and position of this class of mutations in the cohorts they analyzed?

→ We appreciate this valid suggestion. As described above in detail, we generated and functionally analysed a set of 10 *Fgfr2* variants representing E18-nonsense mutations observed in the FMI cohort (Extended Data Fig. 2k). Most proximal E18-truncating mutations were as oncogenic as *Fgfr2^{ΔE18}* and also showed comparable sensitivity to FGFRi (Figs. 2e, 4a, Extended Data Figs. 8, 10e-g, 16a).

References

- Abou-Alfa, Ghassan K., Vaibhav Sahai, Antoine Hollebecque, Gina Vaccaro, Davide Melisi, Raed Al-Rajabi, Andrew S. Paulson, Mitesh J. Borad, David Gallinson, Adrian G. Murphy, Do-Youn Oh, Efrat Dotan, Daniel V Catenacci, Eric Van Cutsem, Tao Ji, Christine F. Lihou, Huiling Zhen, Luis Féliz, and Arndt Vogel. 2020. “Pemigatinib for Previously Treated, Locally Advanced or Metastatic Cholangiocarcinoma: A Multicentre, Open-Label, Phase 2 Study.” *The Lancet Oncology* 21(5):671–84.
- Cornelissen, Lisette M., Linda Henneman, Anne Paulien Drenth, Eva Schut, Roebi de Bruijn, Sjoerd Klarenbeek, Wilbert Zwart, and Jos Jonkers. 2019. “Exogenous ER α Expression in the Mammary Epithelium Decreases Over Time and Does Not Contribute to P53-Deficient Mammary Tumor Formation in Mice.” *Journal of Mammary Gland Biology and Neoplasia* 24(4):305–21.
- Gendarme, Sébastien, Olivier Bylicki, Christos Chouaid, and Florian Guisier. 2022. “ROS-1 Fusions in Non-Small-Cell Lung Cancer: Evidence to Date.” *Current Oncology* 29(2):641–58.
- Iorio, Francesco, Theo A. Knijnenburg, Daniel J. Vis, Graham R. Bignell, Michael P. Menden, Michael Schubert, Nanne Aben, Emanuel Gonçalves, Syd Barthorpe, Howard Lightfoot, Thomas Cokelaer, Patricia Greninger, Ewald van Dyk, Han Chang, Heshani de Silva, Holger Heyn, Xianming Deng, Regina K. Egan, Qingsong Liu, Tatiana Mironenko, Xeni Mitropoulos, Laura Richardson, Jinhua Wang, Tinghu Zhang, Sebastian Moran, Sergi Sayols, Maryam Soleimani, David Tamborero, Nuria Lopez-Bigas, Petra Ross-Macdonald, Manel Esteller, Nathanael S. Gray, Daniel A. Haber, Michael R. Stratton, Cyril H. Benes, Lodewyk F. A. Wessels, Julio Saez-Rodriguez, Ultan McDermott, and Mathew J. Garnett. 2016. “A Landscape of Pharmacogenomic Interactions in Cancer.” *Cell* 166(3):740–54.
- Liu, Zhongyang, Feifei Guo, Jiyang Zhang, Jian Wang, Liang Lu, Dong Li, and Fuchu He. 2013. “Proteome-Wide Prediction of Self-Interacting Proteins Based on Multiple Properties.” *Molecular & Cellular Proteomics : MCP* 12(6):1689–1700.
- Nakamura, Ikuko Takeda, Shinji Kohsaka, Masachika Ikegami, Hiroshi Ikeuchi, Toshihide Ueno, Kunhua Li, Tyler S. Beyett, Takafumi Koyama, Toshio Shimizu, Noboru Yamamoto, Fumiyuki Takahashi, Kazuhisa Takahashi, Michael J. Eck, and Hiroyuki Mano. 2021. “Comprehensive Functional Evaluation of Variants of Fibroblast Growth Factor Receptor Genes in Cancer.” *NPJ Precision Oncology* 5(1):66.
- Silverman, Ian M., Antoine Hollebecque, Luc Friboulet, Sherry Owens, Robert C. Newton, Huiling Zhen, Luis Féliz, Camilla Zecchetto, Davide Melisi, and Timothy C. Burn. 2021. “Clinicogenomic Analysis of FGFR2 -Rearranged Cholangiocarcinoma Identifies Correlates of Response and Mechanisms of Resistance to Pemigatinib.” *Cancer Discovery* 11(2):326–39.
- Szybowska, Patrycja, Michal Kostas, Jørgen Wesche, Antoni Wiedlocha, and Ellen Margrethe Haugsten. 2019. “Cancer Mutations in FGFR2 Prevent a Negative Feedback Loop Mediated by the ERK1/2 Pathway.” *Cells* 8(6):518.
- Wakefield, Lalage M., Gudmundur Thordarson, Ana I. Nieto, G. Shyamala, Jose J. Galvez, Miriam R. Anver, and Robert D. Cardiff. 2003. “Spontaneous Pituitary Abnormalities and Mammary Hyperplasia in FVB/NCr Mice: Implications for Mouse Modeling.” *Comparative Medicine* 53(4):424–32.

Reviewer Reports on the First Revision:

Referees' comments:

Referee #2 (Remarks to the Author):

I feel that the points raised in the previous round of review have been satisfactorily addressed.

Referee #3 (Remarks to the Author):

The revised manuscript by Zingg et al., "Truncated FGFR2 is a clinically actionable oncogene in multiple cancers" provides very substantive response to reviewer comments and overall appears suitable for acceptance given both the depth of analysis and its translational implications in a pan-cancer precision oncology. I do have the following comments/ concerns:

1. The response to Reviewer 3, #12, the question about FGFR2 IHC is addressed by orthogonal assays including luc assay/western blot; not directly by supportive IHC data. This is intriguing, since functional data is highly supportive of the mutant construct's activity, it seems IHC should be feasible. Authors, please clarify.
2. In the global phosphoproteomic analysis of NMuMG cells expressing Fgfr2 variants (Figure 2b), it's curious why any changes in FGFR2 phosphorylation are not included? It seems if available, that could provide orthogonal evidence of the (relative) activity of the various FGFR2 aberrations. This merits a look in and an explanation.
3. The rebuttal document should highlight "new" figures/ panels, and the revised manuscript should highlight "revised/ new" texts for easy navigation of the changes. Many of the questions and their responses refer to different figure/panel numbers, making it difficult and time consuming to track the changes.

Author Rebuttals to First Revision:

Re-submission of revised manuscript 2021-01-00621E, “Truncated *FGFR2* is a clinically actionable oncogene in multiple cancers”

Point-by-point response to referees’ comments

Referee #2 (Remarks to the Author):

I feel that the points raised in the previous round of review have been satisfactorily addressed.

→ We are happy to read that referee #2 considers all issues raised in the previous round of revision to be satisfactorily addressed.

Referee #3 (Remarks to the Author):

The revised manuscript by Zingg et al., “Truncated FGFR2 is a clinically actionable oncogene in multiple cancers” provides very substantive response to reviewer comments and overall appears suitable for acceptance given both the depth of analysis and its translational implications in a pan-cancer precision oncology. I do have the following comments/ concerns:

1. The response to Reviewer 3, #12, the question about FGFR2 IHC is addressed by orthogonal assays including luc assay/western blot; not directly by supportive IHC data. This is intriguing, since functional data is highly supportive of the mutant construct’s activity, it seems IHC should be feasible. Authors, please clarify.

→ We agree that, in principle, FGFR2 positive cells could already be detected at early timepoints in mammary glands of *Fgfr2* GEMMs using IHC. However, we previously observed that *Wap-Cre*-mediated switching does not occur simultaneously in 100% of mammary epithelial cells. Therefore, it would be difficult to distinguish mammary epithelial cells ectopically expressing FGFR2 from the engineered alleles versus endogenous FGFR2 expression in the mammary epithelium and stroma. Hence, we instead favoured the orthogonal *ex vivo* approaches where switching of the *Fgfr2* alleles occurred in virtually all mammary epithelial cells exposed to AdenoCre, allowing clean signal quantifications versus baseline FGFR2 expression in mock-treated cells.

2. In the global phosphoproteomic analysis of NMuMG cells expressing *Fgfr2* variants (Figure 2b), it’s curious why any changes in FGFR2 phosphorylation are not included? It seems if available, that could provide orthogonal evidence of the (relative) activity of the various FGFR2 aberrations. This merits a look in and an explanation.

→ Phosphosites in FGFR2 detected via IMAC enrichment or phospho-Tyr IP in both NMuMG cells expressing *Fgfr2* variants as well as *Fgfr2* variant-driven tumours are fully disclosed in Extended Data Figs. 5c, 9b (7d and 13b in the previously submitted 2021-01-00621D version of the manuscript). The referee must have missed these data when reviewing our resubmission. Due to space limitations, we are unfortunately unable to include these data into the main figures.

3. The rebuttal document should highlight “new” figures/ panels, and the revised manuscript should highlight “revised/ new” texts for easy navigation of the changes. Many of the questions and their responses refer to different figure/panel numbers, making it difficult and time consuming to track the changes.

→ We apologise that we did not resubmit a ‘tracked changes’ version of the manuscript.